# Causal Effect Identification in the Presence of Latent Confounding with a Single Agnostic Proxy

## Abstract

We consider the problem of identifying the causal effect of a treatment on an outcome in the presence of latent confounding. Many existing works utilize proxies of the latent confounder to adjust for it indirectly, typically requiring multiple proxies. Within the framework of latent variable linear non-Gaussian acyclic model (lvLiNGAM), we propose a causal effect identification procedure requiring only a single proxy. Moreover, this proxy can be agnostic, which means that: first, it can have an arbitrary causal relationship with the treatment/outcome; second, this causal relationship is not required to be known a priori. The complexity of the agnostic proxy precludes identifying the causal effect via a simple analytical formula. Consequently, our procedure is designed to first derive several candidate solutions from cross-cumulants and then isolate the valid solution by examining certain independence relationships. We present and prove a series of new theoretical results, which collectively establish the soundness of our procedure: given the observational population distribution, it correctly identifies the true causal effect when identifiable, and correctly reports unidentifiability otherwise. Also, we conduct experiments to validate the correctness of our theoretical results.

## 1 Introduction

Ascertaining the causal effect of a treatment on an outcome is a crucial challenge in various fields such as social sciences (Sobel, 2000; Ogburn et al., 2024), public health (Hernán & Robins, 2006; Boon et al., 2021), and agriculture (Wuepper & Finger, 2023). The gold standard is conducting randomized experiments, but this is usually prohibitively expensive and even infeasible due to ethical or legal concerns. Consequently, researchers increasingly focus on the problem of whether and how the causal effect can be uniquely determined from the observational (non-experimental) population distribution, commonly known as causal effect identification. The backdoor formula (Pearl et al., 2016) is a widely-used identification strategy, but it is unsuitable for scenarios involving latent confounding that introduces spurious correlations. To tackle latent confounding, many works assume access to proxies of the latent confounder, which are often its observed children other than the treatment and the outcome. Intuitively, proxies serve as imperfect measurements of the latent confounder, enabling us to adjust for it indirectly.

Most early studies that employ proxies to handle latent confounding focus on causal effect identification in non-parametric structural causal models (SCMs) (Miao et al., 2018; Shi et al., 2020; Mastouri et al., 2021; Xu et al., 2021; Shpitser et al., 2023; Cui et al., 2024). They typically require multiple proxies—a requirement that is often difficult or even impossible to meet in practice—unless certain stringent conditions are satisfied. For instance, if the latent confounder $L$ has only one proxy $Z$, Kuroki & Pearl (2014) impose a stringent requirement that $Z$ and $L$ are both finite discrete variables and $P(Z|L)$ is known a priori. Recently, some works investigate parametric SCMs, particularly linear SCMs (Kivva et al., 2023; Tramontano et al., 2024; Kummerfeld et al., 2024; Xie et al., 2024a) which are the simplest parametric SCMs. Although a single proxy still cannot guarantee identifiability in all linear SCMs (Kivva et al., 2023), this is achievable in latent variable linear non-Gaussian acyclic models (lvLiNGAMs) where all exogenous noises are non-Gaussian. Within the framework of lvLiNGAM, Kivva et al. (2023) propose an identification procedure requiring only a single proxy, but this proxy must have no causal relationship with the treatment or the outcome, we refer to such

a proxy an isolated proxy in the following; the identification procedures of Tramontano et al. (2025) can work with a single proxy that is either isolated or a parent of the treatment. Notably, both Kivva et al. (2023) and Tramontano et al. (2025) assume that the underlying causal graph is known a priori.

In this paper, within the framework of lvLiNGAM, we propose a novel causal effect identification procedure. Our procedure can operate with a single agnostic proxy, where "agnostic" encompasses two distinct properties: flexibility and uncertainty. Flexibility means that the proxy can have an arbitrary causal relationship with the treatment/outcome. Uncertainty means that this causal relationship is not required to be known a priori.

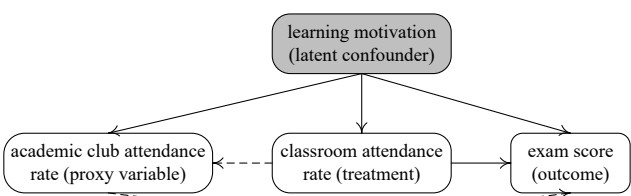

Figure 1: Academic club attendance rate is an agnostic proxy of learning motivation. The dashed lines represent potential causal relationships whose existence is uncertain.

Agnostic proxies are common in real-world scenarios. Consider the causal structure shown in Fig. 1, it is unknown whether academic club scheduling conflicts with classroom sessions, so we cannot determine whether classroom attendance rate impacts academic club attendance rate; it is also unknown whether club curriculum and exam scope overlaps, so we cannot determine whether academic club attendance rate affects exam score. Therefore, academic club attendance rate is an agnostic proxy of learning motivation. The complexity of the agnostic proxy precludes identifying the causal effect via a simple analytical formula. Consequently, our identification procedure first generates several candidate solutions by solving a quadratic equation $a\lambda^2 + b\lambda + c = 0$ where $a, b, c$ are all polynomial functions of cross-cumulants, and then isolates the valid solution from these candidates by examining certain independence relationships. We present and prove a series of new theoretical results, providing a formal guarantee for the soundness of our identification procedure: given the observational population distribution, it correctly identifies the true causal effect when identifiable, and correctly reports unidentifiability otherwise. Also, we create a plug-in estimation method based on our identification procedure and show that its error converges to zero as the sample size increases, empirically validating the correctness of our theoretical results.

## 2 PRELIMINARY

### § latent variable Linear Non-Gaussian Acyclic Model (lvLiNGAM)

Throughout this paper, we focus on causal effect identification within the framework of lvLiNGAM. Given a lvLiNGAM, its causal graph $\mathcal{G} = (\mathbf{V}, \mathbf{E})$ is a directed acyclic graph (DAG) where $\mathbf{V} = \{V_i\}_{i=1}^n$ consists of both latent and observed variables. Each $V_i \in \mathbf{V}$ follows

$$V_i = \sum_{V_j \in \mathbf{V}} a_{V_i V_j} V_j + \epsilon_{V_i}, \tag{1}$$

where $\epsilon_{V_i}$ refers to an exogenous noise. All exogenous noises have non-Gaussian distributions and are mutually independent. Without loss of generality, we assume all exogenous noises have a mean of zero. $a_{V_i V_j} \neq 0$ iff $V_j$ is a parent of $V_i$.

**Assumption 1.** *Given a polynomial in $\{a_{V_i V_j} | V_i, V_j \in \mathbf{V}\}$, it equals 0 only if it is identically 0.*

**Remark 1.** This assumption holds generically, since if a polynomial is not identically 0, the zeros set of the polynomial has 0 Lebesgue measure in the parameter space.

Eq. (1) can also be written as

$$V_i = \sum_{V_j \in \mathbf{V}} m_{V_i V_j} \epsilon_{V_j}, \tag{2}$$

where $M = (I - A)^{-1} = I + \sum_{i=1}^l A^i$ with $M$ being the matrix whose $(i, j)$-entry is $m_{V_i V_j}$, $A$ being the matrix whose $(i, j)$-entry is $a_{V_i V_j}$, and $l$ being the length of the longest directed path in $\mathcal{G}$. In particular, $m_{V_i V_j} = 0$ iff $V_i \neq V_j$ and $V_j$ is not an ancestor of $V_i$, $m_{V_i V_j} = a_{V_i V_j}$ iff $V_j$ is a parent of $V_i$ and there is no other directed path from $V_j$ to $V_i$. The causal effect of $V_j$ on $V_i$ is exactly $m_{V_i V_j}$.

### § Cross-cumulant

As mentioned above, our identification procedure first derives candidate solutions from cross-cumulants. Cross-cumulants are widely-used in both signal processing (Thi & Jutten, 1995; Belkin et al., 2013; Voss et al., 2013; Ge & Zou, 2016) and causal inference (Cai et al., 2023; Qiao et al., 2024; Chen et al., 2024; Li et al., 2025). We present the definition of cross-cumulant along with some properties as follows.

**Definition 1.** *Given random variables $V_1, ..., V_n$, the cross-cumulant $\text{Cum}(V_1, ..., V_n)$ is defined as*

$$\text{Cum}(V_1, ..., V_n) = \sum_{\pi} (-1)^{|\pi|-1} (|\pi| - 1)! \prod_{B \in \pi} \mathbb{E}\left[\prod_{i \in B} V_i\right], \tag{3}$$

*where $\pi$ is enumerated over all partitions of $\{1, ..., n\}$.*

**Example 1.** *Suppose $V_1, V_2, V_3, V_4$ all have a mean of zero,*

$$\text{Cum}(V_1, V_2, V_3, V_4) = \mathbb{E}[V_1 V_2 V_3 V_4] - \mathbb{E}[V_1 V_2]\mathbb{E}[V_3 V_4] - \mathbb{E}[V_1 V_3]\mathbb{E}[V_2 V_4] - \mathbb{E}[V_1 V_4]\mathbb{E}[V_2 V_3]. \tag{4}$$

Cross-cumulants are known to manifest the following properties:

1. (Exchangeability) $\text{Cum}(V_1, ..., V_i, ..., V_j, ..., V_n) = \text{Cum}(V_1, ..., V_j, ..., V_i, ..., V_n)$.

2. (Linearity) If $k$ is a constant, then $\text{Cum}(kV_1, ..., V_n) = k\text{Cum}(V_1, ..., V_n)$; if $V_1'$ is a random variable, then $\text{Cum}(V_1 + V_1', ..., V_n) = \text{Cum}(V_1, ..., V_n) + \text{Cum}(V_1', ..., V_n)$.

3. (Independence) If $X_1, ..., X_n$ are independent of $V_1, ..., V_n$, then $\text{Cum}(V_1 + X_1, ..., V_n + X_n) = \text{Cum}(V_1, ..., V_n) + \text{Cum}(X_1, ..., X_n)$.

For ease of exposition, we denote $\underbrace{\text{Cum}(X, \ldots)}_{i \text{ times}}$ by $C_i(X)$ and $\text{Cum}(\underbrace{X, \ldots}_{i \text{ times}}, \underbrace{Y, \ldots}_{j \text{ times}})$ by $C_{i,j}(X, Y)$.

**Assumption 2.** *For every $V \in \mathbf{V}$, $|C_4(\epsilon_V)| < \infty$ and $C_4(\epsilon_V) \neq 0$.*

**Remark 2.** First, $|C_4(\epsilon_V)| < \infty$ is a technical assumption that primarily serves to exclude those pathological distributions characterized by extremely heavy tails (e.g., Cauchy distribution or Student's t-distributions with $\nu \leq 4$). Second, note that $\epsilon_V$ is a non-Gaussian variable, $C_4(\epsilon_V) \neq 0$ simply specifies this non-Gaussianity in terms of a non-zero fourth-order cumulant. Intuitively, this assumption implies that the distribution has tail behavior that is not identical to a Gaussian distribution. Finally, Asmp. 2 logically implies that $|C_2(\epsilon_V)| < \infty$ and $C_2(\epsilon_V) \neq 0$.

## § Independence

After deriving candidate solutions from cross-cumulants, our identification procedure isolates the valid solution from candidates by examining certain independence relationships. Darmois-Skitovitch Theorem (Kagan et al., 1973), which establishes the connection between independence and non-Gaussianity, constitutes a theoretical cornerstone.

**Darmois-Skitovitch Theorem.** *Suppose two random variables $V_1$ and $V_2$ are both linear combinations of independent random variables $\{e_i\}_i$:*

$$V_1 = \sum_i \alpha_i e_i, \quad V_2 = \sum_i \beta_i e_i. \tag{5}$$

*If $V_1 \perp\!\!\!\perp V_2$, then for every non-Gaussian $e_j$, $\alpha_j \beta_j = 0$, that is, at least one of $V_1$ and $V_2$ does not contain $e_j$.*

## 3 MAIN RESULTS

We focus on such a problem: There are a treatment variable $T$ and an outcome variable $O$ where $T$ is a parent of $O$, these two variables have a common parent $L$ which is a latent variable, and $L$ has another observed child $Z$ serving as its agnostic proxy. Particularly, $Z$ can have an arbitrary causal relationship with $(T, O)$ and this causal relationship is not known a priori. Our goal is to identify the causal effect of $T$ on $O$ (i.e. $m_{OT}$) from the joint distribution of $(T, O, Z)$.

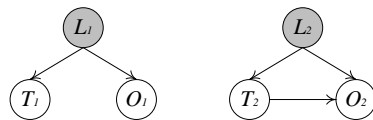

Figure 2: Two causal structures showing different properties on cross-cumulants.

### 3.1 A Motivating Example

We first show by a simple example that cross-cumulants can be used to generate candidate solutions and independence relationships can be used to isolate the valid solution, with particular emphasis on the former. Specifically, consider the two causal structures shown as Fig. 2. It is easy to verify that

$$C_{3,1}(T_1, O_1)C_{1,3}(T_1, O_1) = C_{2,2}^2(T_1, O_1), \quad C_{3,1}(T_2, O_2)C_{1,3}(T_2, O_2) \neq C_{2,2}^2(T_2, O_2). \quad (6)$$

Clearly, if removing the causal edge $T_2 \to O_2$, that is, subtracting $m_{O_2 T_2} T_2$ from $O_2$, the two causal structures shown as Fig. 2 are equivalent to each other. That is, one root of the equation in $\lambda$:

$$C_{3,1}(T_2, O_2 - \lambda T_2)C_{1,3}(T_2, O_2 - \lambda T_2) = C_{2,2}^2(T_2, O_2 - \lambda T_2), \quad (7)$$

is exactly $m_{O_2 T_2}$. In fact, Eq. (7) can be simplified as a quadratic equation (see Eq. (18) in App. A), two roots of which are $m_{O_2 L_2}/m_{T_2 L_2}$ and $m_{O_2 T_2}$ (see Lem. 2 in App. A). Therefore, cross-cumulants can be used to generate two candidate solutions.

Moreover, $O_2 - \lambda T_2$ contains different exogenous noises depending on the value of $\lambda$: it contains $\epsilon_{T_2}$ but not $\epsilon_{L_2}$ when $\lambda = m_{O_2 L_2}/m_{T_2 L_2}$, whereas it contains $\epsilon_{L_2}$ but not $\epsilon_{T_2}$ when $\lambda = m_{O_2 T_2}$. According to the Darmois-Skitovitch Theorem, these different underlying noise structures imply that the independence relationship between $O_2 - \lambda T_2$ and other specific variables will vary according to the value of $\lambda$. Therefore, we can distinguish between $m_{O_2 T_2}$ and $m_{O_2 L_2}/m_{T_2 L_2}$ by examining certain independence relationships.

For ease of exposition, we define cross-cumulant constraint and independence constraint as follows.

**Definition 2.** *Given two random variables $V_1, V_2$, we say $(V_1, V_2)$ satisfies the cross-cumulant constraint if $C_{3,1}(V_1, V_2)C_{1,3}(V_1, V_2) = C_{2,2}^2(V_1, V_2)$.*

**Definition 3.** *Given three random variables $V_1, V_2, V_3$, we say $(V_1, V_2, V_3)$ satisfies the independence constraint if $\mathrm{Cov}(V_1, V_3)\mathrm{Cov}(V_2, V_3) \neq 0$ and $\mathrm{Cov}(V_2, V_3)V_1 - \mathrm{Cov}(V_1, V_3)V_2 \perp\!\!\!\perp V_3$.*

**Remark 3.** The independence constraint is an adaptation of the GIN condition (Xie et al., 2020), which is widely-used in causal inference (Chen et al., 2022; Xie et al., 2022; Jin et al., 2024; Xie et al., 2024a). While the GIN condition in previous literature typically operates on original variables (such as $T$ and $O$ in Fig. 3) to infer their causal relations with each other, the independence constraint in this paper usually involves transformations of the original variables (such as $O - \lambda T$ in Thm. 2) to identify the valid causal effect from candidate solutions.

### 3.2 Detailed Analysis

#### § Flexibility

The first property of the agnostic proxy is flexibility, that is, $Z$ can have an arbitrary causal relationship with $(T, O)$, so the underlying causal structure may be any one of those depicted in Fig. 3. In the following, we investigate whether and how, given any causal structure in Fig. 3, the causal effect of interest can be uniquely determined from the joint distribution of $(T, O, Z)$.

In Fig. 3(a), $Z$ has no causal relationship with either $T$ or $O$, so $Z$ is an isolated proxy. In this case, $m_{OT}$ can be identified via an analytical formula (Kivva et al., 2023). Specifically, Let

$$g(X, Y) := \mathrm{sgn}(\mathrm{Cov}(X, Y))\sqrt{\frac{C_{3,1}(X, Y)}{C_{1,3}(X, Y)}}, \quad (8)$$

$$f(T, O, Z) := \frac{\mathrm{Cov}(T, O) - g(T, Z)\mathrm{Cov}(O, Z)}{\mathrm{Var}(T) - g(T, Z)\mathrm{Cov}(T, Z)}, \quad (9)$$

where $\mathrm{sgn}(\cdot)$ is the sign function. $m_{OT}$ is exactly $f(T, O, Z)$. Formally,

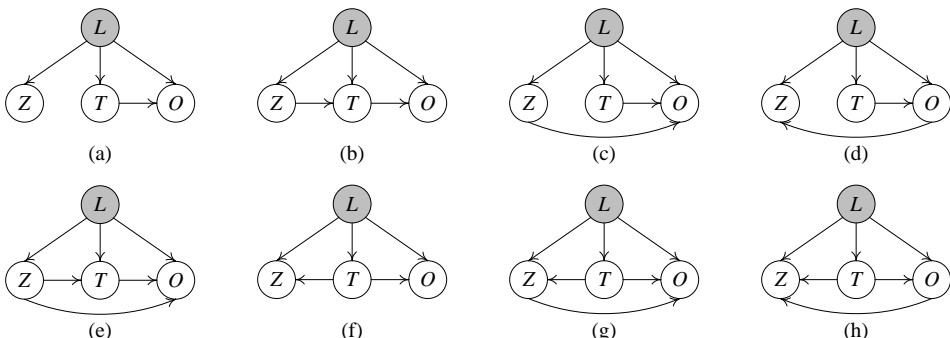

Figure 3: All possible causal structures in Sec. 3.

**Theorem 1.** *In Fig. 3(a), $m_{OT} = f(T, O, Z)$.*

In Fig. 3(b), following the discussion in Sec. 3.1, $(T, O - \lambda T)$ satisfy the cross-cumulant constraint iff $\lambda = m_{OT}$ or $m_{OL}/m_{TL}$. Furthermore, $(Z, T, O - \lambda T)$ satisfies the independence constraint iff $\lambda = m_{OT}$. Formally,

**Theorem 2.** *In Fig. 3(b), $m_{OT} = \lambda$ iff $(T, O - \lambda T)$ satisfies the cross-cumulant constraint and $(Z, T, O - \lambda T)$ satisfies the independence constraint.*

In Fig. 3(c), following the discussion in Sec. 3.1, $(T, O - \lambda T)$ satisfy the cross-cumulant constraint iff $\lambda = m_{OT}$ or $m_{OL}/m_{TL}$. Furthermore, $(O - \lambda T, Z, T)$ satisfies the independence constraint iff $\lambda = m_{OT}$. Formally,

**Theorem 3.** *In Fig. 3(c), $m_{OT} = \lambda$ iff $(T, O - \lambda T)$ satisfies the cross-cumulant constraint and $(O - \lambda T, Z, T)$ satisfies the independence constraint.*

In Fig. 3(d), $m_{ZO}$ can be identified in the same way that $m_{OT}$ in Fig 3(b) is identified. Next, note that $Z$ becomes an isolated proxy after subtracting $m_{ZO}O$ from $Z$, so $m_{OT}$ can be identified via Equation (9). Formally,

**Theorem 4.** *In Fig. 3(d), $m_{OT} = f(T, O, Z - \lambda O)$ where $(O, Z - \lambda O)$ satisfies the cross-cumulant constraint and $(T, O, Z - \lambda O)$ satisfies the independence constraint.*

In Fig. 3(e), following the discussion in Sec. 3.1, $(Z, T - \lambda_1 Z)$ satisfies the cross-cumulant constraint iff $\lambda_1 = m_{TZ}$ or $m_{TL}/m_{ZL}$ and $(Z, O - \lambda_2 Z)$ satisfies the cross-cumulant constraint iff $\lambda_2 = m_{OZ}$ or $m_{OL}/m_{ZL}$. Furthermore, $(T - \lambda_1 Z, O - \lambda_2 Z, Z)$ satisfies the independence constraint iff $(\lambda_1, \lambda_2) = (m_{TZ}, m_{OZ})$ or $(m_{TL}/m_{ZL}, m_{OL}/m_{ZL})$. In both cases, after subtracting $\lambda_1 Z, \lambda_2 Z$ from $T, O$ respectively, $Z$ becomes an isolated proxy, so $m_{OT}$ can be identified via Equation (9). Formally,

**Theorem 5.** *In Fig. 3(e), $m_{OT} = f(T - \lambda_1 Z, O - \lambda_2 Z, Z)$ where both $(Z, T - \lambda_1 Z)$ and $(Z, O - \lambda_2 Z)$ satisfy the cross-cumulants constraint and $(T - \lambda_1 Z, O - \lambda_2 Z, Z)$ satisfies the independence constraint.*

In Fig. 3(f)-(h), $m_{OT}$ is unidentifiable. We construct two SCMs $\mathcal{M}$ and $\tilde{\mathcal{M}}$ that share the same causal structure shown as Fig. 3(f):

$$\mathcal{M} := \begin{cases} L = \epsilon_L, \\ T = a_{TL}L + \epsilon_T, \\ O = a_{OT}T + a_{OL}L + \epsilon_O, \\ Z = a_{ZT}T + a_{ZL}L + \epsilon_Z, \end{cases} , \quad \tilde{\mathcal{M}} := \begin{cases} \tilde{L} = \epsilon_T, \\ \tilde{T} = \tilde{L} + a_{TL}\epsilon_L, \\ \tilde{O} = \frac{a_{OT}a_{TL}+a_{OL}}{a_{TL}}\tilde{T} - \frac{a_{OL}}{a_{TL}}\tilde{L} + \epsilon_O, \\ \tilde{Z} = \frac{a_{ZT}a_{TL}+a_{ZL}}{a_{TL}}\tilde{T} - \frac{a_{ZL}}{a_{TL}}\tilde{L} + \epsilon_Z, \end{cases} . \quad (10)$$

It is easy to verify that the joint distribution of $(T, O, Z)$ is identical to that of $(\tilde{T}, \tilde{O}, \tilde{Z})$. However, $m_{OT} = a_{OT}$ while $m_{\tilde{O}\tilde{T}} = \frac{a_{OT}a_{TL}+a_{OL}}{a_{TL}}$. Therefore, $m_{OT}$ in Fig. 3(f) is unidentifiable. Analogously, we can also construct such two SCMs for Fig. 3(g) and Fig. 3(h). For Fig. 3(g), we add $a_{OZ}Z$ to $O$, add $a_{OZ}\tilde{Z}$ to $\tilde{O}$, and keep all other assignments in Eq. (10) fixed. For Fig. 3(h), we add $a_{ZO}O$ to $Z$, add $a_{ZO}\tilde{O}$ to $\tilde{Z}$, and keep all other assignments in Eq. (10) fixed.

**Theorem 6.** *In Fig. 3(f)-(h), $m_{OT}$ is unidentifiable.*

---

**Algorithm 1:** Identification procedure

---

**Input:** Joint distribution of $(T, O, Z)$

**Output:** $m_{OT}$

1 **if** $(T, O, Z)$ *satisfies the independence constraint* **then**
2 $\quad$ | $\quad$ **return** $f(T, O, Z)$
3 **else if** *there exists one and only one* $\lambda \neq 0$ *s.t. the constraints in Thm. 2 are satisfied* **then**
4 $\quad$ | $\quad$ **return** $\lambda$
5 **else if** *there exists* $\lambda \neq 0$ *s.t. the constraints in Thm. 3 are satisfied* **then**
6 $\quad$ | $\quad$ **return** $\lambda$
7 **else if** *there exists* $\lambda \neq 0$ *s.t. the constraints in Thm. 4 are satisfied* **then**
8 $\quad$ | $\quad$ **return** $f(T, O, Z - \lambda O)$
9 **else if** *there exist* $\lambda_1 \neq 0, \lambda_2 \neq 0$ *s.t. the constraints in Thm. 5 are satisfied* **then**
10 $\quad$ | $\quad$ **return** $f(T - \lambda_1 Z, O - \lambda_2 Z, Z)$
11 **else**
12 $\quad$ | $\quad$ **raise** Error("$m_{OT}$ is unidentifiable.")

---

§ **Uncertainty**

The second property of the agnostic proxy is uncertainty, that is, the causal relationship of $Z$ with $(T, O)$ is not known a priori, so it is unknown which causal structure in Fig. 3 corresponds to the ground truth. In the following, we investigate how to determine the underlying causal structure from the joint distribution of $(T, O, Z)$.

First, since it is only in Fig. 3(a) that $Z$ has no causal relationship with $(T, O)$, there is an independence relationship that holds only in Fig. 3(a). Specifically,

**Corollary 1.** *The ground truth is Fig. 3(a) iff* $(T, O, Z)$ *satisfies the independence constraint.*

Fortunately, we find that Thms. 2-5 which are developed to identify causal effects can also be used to identify causal structures actually. Specifically,

**Corollary 2.** *If the underlying causal structure is not Fig. 3(a), it is Fig. 3(b) iff there exists one and only one* $\lambda \neq 0$ *s.t. the constraints in Thm. 2 are satisfied.*

**Corollary 3.** *If the underlying causal structure is not one of Fig. 3(a)-(b), it is Fig. 3(c) iff there exists* $\lambda \neq 0$ *s.t. the constraints in Thm. 3 are satisfied.*

**Corollary 4.** *If the underlying causal structure is not one of Fig. 3(a)-(d), it is Fig. 3(d) iff there exists* $\lambda \neq 0$ *s.t. the constraints in Thm. 4 are satisfied.*

**Corollary 5.** *If the underlying causal structure is not one of Fig. 3(a)-(c), it is Fig. 3(d) iff there exist* $\lambda_1 \neq 0$ *and* $\lambda_2 \neq 0$ *s.t. the constraints in Thm. 5 are satisfied.*

Finally, it is unnecessary to distinguish between Fig. 3(f)-(h), as $m_{OT}$ is unidentifiable in each case.

§ **Identification procedure**

We summarize our identification procedure in Alg. 1. Based on Thms. 1-6 and Cors. 1-5, our identification procedure is sound. Specifically,

**Theorem 7.** *If Asmps. 1 and 2 hold, Alg. 1 correctly identifies the true causal effect when identifiable, and correctly reports unidentifiability otherwise.*

Particularly, if the underlying causal structure is known a priori, the identification procedure can be substantially simplified: without the need for if-else logic, it can directly identifies the causal effect of interest or reports unidentifiability based on one of Thms. 1-6.

## 4 EXTENSIONS

We now extend the theoretical results in Sec. 3 in two distinct directions.

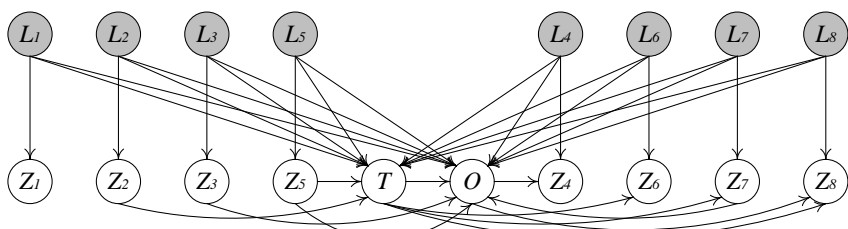

Figure 4: A typical example investigated in Sec. 4.1. The causal relationships of $Z_1, Z_2, ..., Z_8$ with $(T, O)$ follow Fig. 3(a), Fig. 3(b), ..., Fig. 3(h).

## 4.1 MULTIPLE LATENT CONFOUNDERS

In Sec. 3, we focus on the setting with a single latent confounder. In this section, we investigate a more complex setting with multiple latent confounders: $T$ is still a parent of $O$, but $T$ and $O$ have multiple common parents $\mathbf{L} = \{L_1, ..., L_m\}$ that are all latent variables and also root variables. Each $L_i$ has an observed child $Z_i$ which is not a child of any other $L_j \neq L_i$ serving as its agnostic proxy and there exists no causal edge between any two $Z_i, Z_j \in \mathbf{Z} = \{Z_i, ..., Z_m\}$. A typical example is shown as Fig. 4. Our goal is to identify the causal effect of $T$ on $O$ (i.e. $m_{OT}$) from the joint distribution of $(T, O, Z_1, ..., Z_m)$.

For this more complex setting, we present and prove a series of theoretical results (App. C.1), which collectively establish the soundness of our proposed identification procedure (App. C.2). Also, we empirically validate the correctness of our theoretical results(App. C.3). Intuitively, although multiple latent confounders exist between $T$ and $O$, there is only a single latent confounder $L_i$ between each $Z_i$ and $(T, O)$. Following the theoretical results in Sec. 3, the existence, direction, and strength of the causal relationship between $Z_i$ and $(T, O)$ can be determined. With this knowledge, the causal relationship between $Z_i$ and $(T, O)$ can be removed, thereby rendering $Z_i$ an isolated proxy. With isolated proxies, $m_{OT}$ can be identified via an analytical formula similar to Equation (9) (Kivva et al., 2023).

## 4.2 GENERALIZED AGNOSTIC PROXY

While $Z$ is always a child of $L$ in Sec. 3, we let $Z$ be a generalized agnostic proxy in this section: $T$ is still a parent of $O$ and $L$ is still a common parent of both $T$ and $O$, but $Z$ can have an arbitrary causal relationship with $(L, T, O)$ and this causal relationship is not known a priori. That is, the underlying causal structure may be any one of those depicted in Fig. 5 (note that (m)-(t) in Fig. 5 are exactly (a)-(h) in Fig. 3) and it is unknown which one corresponds to the ground truth. Our goal is to identify the causal effect of $T$ on $O$ (i.e. $m_{OT}$) from the joint distribution of $(T, O, Z)$.

For this more complex setting, we present and prove a series of theoretical results (App. D.1), which collectively establish the soundness of our identification procedure (App. D.2). Also, we empirically validate the correctness of our theoretical results(App. D.3). In brief, our identification procedure first determines whether the underlying causal graph is one of Fig. 5(m)-(t) by examining certain independence relationships. If it is, the rest of the problem can be solved by Alg. 1; if it is not, our identification procedure reports unidentifiability because the causal effect of interest is unidentifiable without prior knowledge about which specific one of Fig. 5(a)-(l) the underlying causal graph is.

## 5 RELATED WORK

There is a rich literature on causal effect identification in the presence of latent confounding. Besides instrumental variables (Angrist et al., 1996; Wei et al., 2021; Hartford et al., 2021; Wu et al., 2022; Xie et al., 2024b), one of the most frequently used strategies to handle latent confounding is through proxies. Readers can refer to Tchetgen Tchetgen et al. (2024) for a general introduction. An early study (Kuroki & Pearl, 2014) investigates two different settings. First, in non-parametric SCMs, the causal effect of the treatment $T$ on the outcome $O$ can be identified if (1) the latent confounder $L$ has an isolated proxy $Z$ where $L$ and $Z$ are both discrete finite variables, and $P(Z|L)$ is known a priori; or (2) $L$ has two proxies $Z$ and $W$ where $L, Z, W$ are all discrete finite variables, and some other

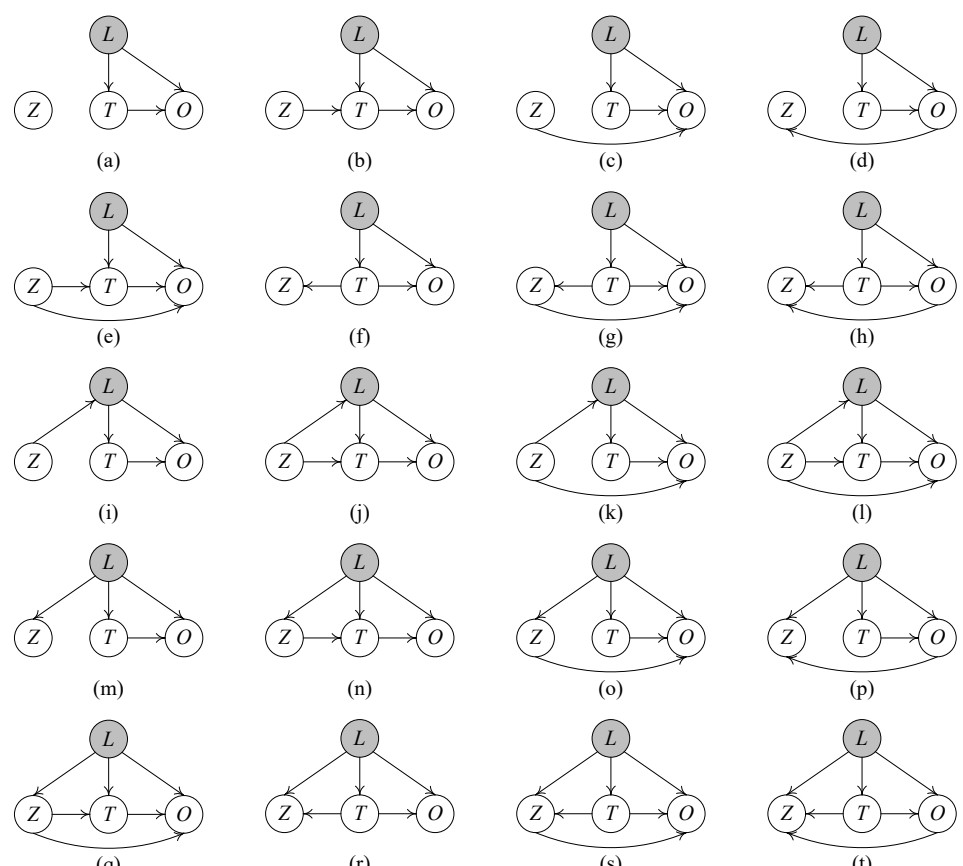

Figure 5: All possible causal structures considered in Sec. 4.2.

conditions, e.g., $W \perp\!\!\!\perp (T, O, Z)|L$ and $(Z, W) \perp\!\!\!\perp O|\{L, T\}$, are satisfied. Second, in linear SCM, the causal effect can be identified if $L$ has two proxies $Z$ and $W$ where $Z \perp\!\!\!\perp O|\{L, T\}$ and $W \perp\!\!\!\perp (T, Z)|L$.

Following this seminal work, many subsequent studies delve deeper into causal effect identification in non-parametric SCMs (Miao et al., 2018; Shi et al., 2020; Mastouri et al., 2021; Xu et al., 2021; Shpitser et al., 2023; Cui et al., 2024). It is known that without additional conditions such as $P(Z|L)$ is known a priori, the causal effect is not identifiable with a single proxy (Pearl, 2010; Kuroki & Pearl, 2014; Xu & Gretton, 2023), so all of these studies require multiple proxies. On the other hand, some recent works explore the sufficient conditions for single-proxy identifiability. For instance, Xu & Gretton (2023) rely on two main conditions: (1) the outcome is deterministically generated given the treatment and the latent confounder, and (2) the single proxy is an isolated proxy. Park et al. (2024) require that (1) $T$ is binary and the potential outcomes $O(1)$ and $O(0)$ serve as the latent confounders, and (2) the single proxy is a negative control outcome (NCE)[1].

Recently, many works turn to investigate parametric SCMs, particularly linear SCMs (Kivva et al., 2023; Tramontano et al., 2024; Kummerfeld et al., 2024; Xie et al., 2024a). The major contributions of Kummerfeld et al. (2024); Xie et al. (2024a) lie in data-driven methods for selecting valid proxies from all observed variables, they still require multiple proxies. Within the framework of lvLiNGAM, Kivva et al. (2023) propose a moment-based[2] identification procedure requiring only a single isolated proxy while the cumulant-based identification procedure of Tramontano et al. (2025) can work with a single proxy that is either isolated or a parent of the treatment. Notably, they both require the underlying causal graph to be known a priori. Following previous studies (Salehkaleybar et al., 2020; Adams et al., 2021; Yang et al., 2022), Tramontano et al. (2024) regard lvLiNGAM as an overcomplete independent components analysis (OICA) model. Under this framework, their identification procedure essentially boils down to identifying an OICA model, which is known to

---

[1] $Z$ is a NCE only if it has no causal relationship with $T$ and $O$ is not a cause of $Z$.

[2] Moment and cumulant are two informationally equivalent characterizations of a probability distribution.

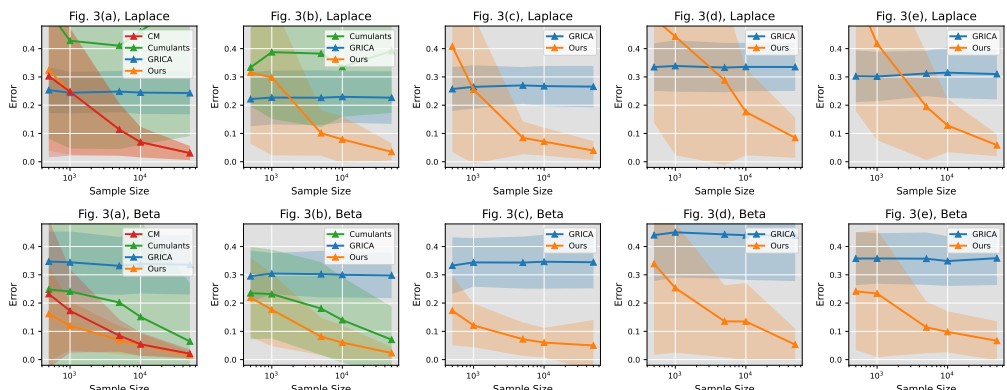

Figure 6: Error in estimating the causal effect w.r.t. sample size.

be an intractable problem. They design an estimation method based on graphical reconstruction independent components analysis (GRICA). Although allowing for an arbitrary causal relationship between the proxy and the treatment/outcome, it requires the underlying causal graph to be known a priori and even lacks consistency guarantees.

## 6  EXPERIMENTS

To empirically validate the correctness of our theoretical results that establish the soundness of Alg. 1, we construct a plug-in estimation method by substituting sample-level cross-cumulants and independence tests into Alg. 1 and then conduct experiments, where our primary focus is on whether the estimation error converges to zero as the sample size increases. We present the key experimental results below, with more details deferred to the App. E.

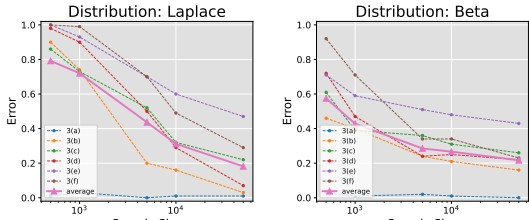

Figure 7: Error in determining the causal structure w.r.t. sample size.

First, we evaluate the ability of our estimation method to handle the uncertainty of the agnostic proxy, that is, given a finite sample generated by any causal structure in Fig. 3 (which is unknown to our estimation method), whether our estimation method can correctly determine the underlying causal structure. For each causal structure, we consider sample sizes of 0.5k, 1k, 5k, 10k, 50k. For each sample size, we synthesize 100 datasets. Each exogenous noise is generated from Laplace distribution with $\mu = 0, b = 1$ and Beta distribution with $\alpha = 1/3, \beta = 2/3$ respectively. Each direct causal strength is sampled from a uniform distribution over $[-1.0, -0.5] \cup [0.5, 1.0]$. The experimental results are summarized in Fig. 7. Taking the curve for 3(a) as an example, the error is defined as the proportion of datasets generated according to Fig. 3(a) that are incorrectly determined as originating from a different causal structure by our estimation method. In addition, since Fig. 3(f)-(h) are grouped into the same class (i.e. the unidentifiable class), we use only Fig. 3(f) as their representative. Obviously, the errors of our estimation method gradually approach zero with increasing sample size.

Second, we evaluate the ability of our estimation method to handle the flexibility of the agnostic proxy, that is, given a finite sample generated by any causal structure in Fig. 3 (which is known to our estimation method), whether our method can accurately estimate the causal effect of interest. We use CM (Kivva et al., 2023), cumulant (Tramontano et al., 2025), and GRICA (Tramontano et al., 2024) as baselines, all of which require the underlying causal structure to be known a priori. The experimental results are summarized in Fig. 6 where the error is given by |Estimated Value-True Value| / max(|True Value|,|Estimated Value|). Particularly, CM can only work with isolated proxies, so we report its performance on only Fig. 3(a). For the same reason, we report the performance of cumulants only on Fig. 3(a) and (b). Moreover, we apply a more advanced version of cumulant, cumulant with minimization, to Fig. 3 (b). Because cumulant relies on non-zero odd-order cumulants (Tramontano et al., 2025), it performs poorly on symmetric distributions such as the Laplace distribution. As sample size increases, the errors of our estimation method gradually ap-

proaches zero, but those of GRICA remain consistently at a substantial level because GRICA lacks consistency guarantees (Tramontano et al., 2024).

Finally, we apply our estimation method to a mouse obesity dataset (Wang et al., 2006). This dataset comprises mouse weight (as the outcome), 17 gene expressions that may influence mouse weight, and 5 potential instrumental variables. Following Miao et al. (2023); Xie et al. (2024a), we assume the data generation adheres to a linear causal model and there is one latent confounder. Regarding each gene expression as the treatment, we use other gene expressions as agnostic proxies. Besides, we do not require 5 potential instrumental variables as auxiliary information. The majority of our findings align with those presented by Miao et al. (2023); Xie et al. (2024a). For instance, the gene expressions Gstm2, Sirpa, and 2010002N04Rik exhibit positive effects on body weight, whereas the gene expression Dscam, Igfbp2, Irx3 show negative impacts. In particular, the negative impacts of Igfbp2 and Irx3 are also supported by Wheatcroft et al. (2007); Schneeberger (2018).

## 7 CONCLUSION

This paper investigates the problem of causal effect identification in the presence of latent confounding. With only a single agnostic proxy whose causal relationship with both the treatment/outcome can be arbitrary and is not required to be known a priori, our identification procedure first derives candidate solutions from cross-cumulants and then isolates the valid solution by examining certain independence relationships. We prove the soundness of our identification procedure and also empirically validate the soundness. Our work not only broadens the applicability of causal effect identification but also may benefit research in natural and social sciences.

**Limitation.** Although our identification procedure is sound, the small-sample performance of the corresponding plug-in estimation method is suboptimal as the variance of high-order cross-cumulants estimators is high with a small sample size. Besides, we primarily focus on addressing latent confounding, whereas not considering selection bias (Wang et al., 2023; Li et al., 2024a;b).

## ETHICS STATEMENT

The research conducted in the paper conform with the ICLR Code of Ethics.

## REPRODUCIBILITY STATEMENT

Our code is available at https://anonymous.4open.science/r/gH81VfT903h4l6Qz0k7

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

TABLE OF CONTENTS

## A  IMPORTANT LEMMAS

We denote the ancestors and descendants of a variable $V$ by $\mathrm{An}(V)$ and $\mathrm{De}(V)$ respectively. Also, we let $\overline{\mathrm{An}}(V) = \mathrm{An}(V) \cup \{V\}$ and $\overline{\mathrm{De}}(V) = \mathrm{De}(V) \cup \{V\}$. With Asmp. 1, we have

1. If $V_1 \in \overline{\mathrm{An}}(V_2)$, then $m_{V_2 V_1} \neq 0$.

2. If $\{V_1, V_2\} \subseteq \overline{\mathrm{An}}(V_3) \cap \overline{\mathrm{An}}(V_4)$ and there exist two non-intersecting directed paths from $\{V_1, V_2\}$ to $\{V_3, V_4\}$, then $m_{V_3 V_1} / m_{V_3 V_2} \neq m_{V_4 V_1} / m_{V_4 V_2}$.

**§ Cross-Cumulants**

Lem. 1 gives a cross-cumulant constraint satisfied in Fig. 8(a), whereas Lem. 2 gives a different cross-cumulant constraint satisfied in Fig. 8(b).

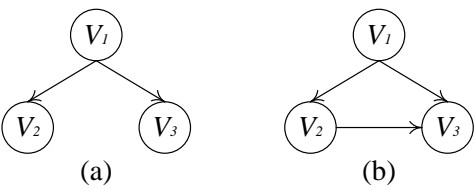

Figure 8: Causal structures appearing in Lems. 1 and 2.

**Lemma 1.** *Suppose three random variables $V_1, V_2, V_3$ are causally sufficient, if the causal structure follows Fig. 8(a), then $(V_2, V_3)$ satisfies the cross-cmulants constraint.*

*Proof.* In Fig. 8(a), $V_2$ and $V_3$ can be expressed as

$$V_2 = m_{V_2 V_1} \epsilon_{V_1} + \epsilon_{V_2}, \quad V_3 = m_{V_3 V_1} \epsilon_{V_1} + \epsilon_{V_3}. \tag{11}$$

Based on the independence property of cross-cumulants, we have

$$C_{3,1}(V_2, V_3) = m_{V_2V_1}^3 m_{V_3V_1} C_4(\epsilon_{V_1}), \tag{12}$$

$$C_{1,3}(V_2, V_3) = m_{V_2V_1} m_{V_3V_1}^3 C_4(\epsilon_{V_1}), \tag{13}$$

$$C_{2,2}(V_2, V_3) = m_{V_2V_1}^2 m_{V_3V_1}^2 C_4(\epsilon_{V_1}). \tag{14}$$

Therefore, $C_{3,1}(V_2, V_3)C_{1,3}(V_2, V_3) = C_{2,2}^2(V_2, V_3)$. □

**Lemma 2.** *Suppose three random variables $V_1, V_2, V_3$ are causally sufficient, if the causal structure follows Fig. 8(b), then $(V_2, V_3 - \lambda V_2)$ satisfies the cross-cmulants constraint iff $\lambda = m_{V_3V_2}$ or $\frac{m_{V_3V_1}}{m_{V_2V_1}}$.*

*Proof.* Based on the exchangeability and linearity of cross-cumulants, we have

$$C_{3,1}(V_2, V_3 - \lambda V_2) = C_{3,1}(V_2, V_3) - \lambda C_4(V_2), \tag{15}$$

$$C_{1,3}(V_2, V_3 - \lambda V_2) = C_{1,3}(V_2, V_3) - 3\lambda C_{2,2}(V_2, V_3) + 3\lambda^2 C_{3,1}(V_2, V_3) - \lambda^3 C_4(V_2), \tag{16}$$

$$C_{2,2}(V_2, V_3 - \lambda V_2) = C_{2,2}(V_2, V_3) - 2\lambda C_{3,1}(V_2, V_3) + \lambda^2 C_4(V_2). \tag{17}$$

Let $C_{3,1}(V_2, V_3 - \lambda V_2)C_{1,3}(V_2, V_3 - \lambda V_2) = C_{2,2}(V_2, V_3 - \lambda V_2)^2$, we have

$$(C_{3,1}^2(V_2, V_3) - C_{2,2}(V_2, V_3)C_4(V_2))\lambda^2 + (C_{1,3}(V_2, V_3)C_4(V_2) - C_{3,1}(V_2, V_3)C_{2,2}(V_2, V_3))\lambda$$

$$+ C_{2,2}^2(V_2, V_3) - C_{3,1}(V_2, V_3)C_{1,3}(V_2, V_3) = 0. \tag{18}$$

In Fig. 8(b), $V_2$ and $V_3$ can be expressed as

$$V_2 = m_{V_2V_1}\epsilon_{V_1} + \epsilon_{V_2}, \quad V_3 = m_{V_3V_1}\epsilon_{V_1} + m_{V_3V_2}\epsilon_{V_2} + \epsilon_{V_3}. \tag{19}$$

Note that

$$C_{3,1}(V_2, V_3) = m_{V_2V_1}^3 m_{V_3V_1} C_4(\epsilon_{V_1}) + m_{V_3V_2} C_4(\epsilon_{V_2}), \tag{20}$$

$$C_{1,3}(V_2, V_3) = m_{V_2V_1} m_{V_3V_1}^3 C_4(\epsilon_{V_1}) + m_{V_3V_2}^3 C_4(\epsilon_{V_2}), \tag{21}$$

$$C_{2,2}(V_2, V_3) = m_{V_2V_1}^2 m_{V_3V_1}^2 C_4(\epsilon_{V_1}) + m_{V_3V_2}^2 C_4(\epsilon_{V_2}), \tag{22}$$

and $C_4(\epsilon_{V_1})C_4(\epsilon_{V_2}) \neq 0$ based on Asmp. 2, we can write Eq. (18) as

$$(2m_{V_2V_1}^2 m_{V_3V_1} m_{V_3V_2} - m_{V_2V_1} m_{V_3V_1}^2 - m_{V_2V_1}^3 m_{V_3V_2}^2)\lambda^2$$

$$+(m_{V_3V_1}^3 + m_{V_2V_1}^3 m_{V_3V_2}^3 - m_{V_2V_1} m_{V_3V_1}^2 m_{V_3V_2} - m_{V_2V_1}^2 m_{V_3V_1} m_{V_3V_2}^2)\lambda$$

$$+(2m_{V_2V_1} m_{V_3V_1}^2 m_{V_3V_2} - m_{V_2V_1}^2 m_{V_3V_1} m_{V_3V_2}^3 - m_{V_3V_1}^3 m_{V_3V_2})$$

$$=0. \tag{23}$$

With Asmp. 1, $2m_{V_2V_1}^2 m_{V_3V_1} m_{V_3V_2} - m_{V_2V_1} m_{V_3V_1}^2 - m_{V_2V_1}^3 m_{V_3V_2}^2 \neq 0$, so Eq. (23) is a quadratic equation in $\lambda$. Trivially, its two roots are $m_{V_3V_2}$ and $\frac{m_{V_3V_1}}{m_{V_2V_1}}$. □

## § Independence

In the following, Lem. 3 gives a sufficient condition for the independence constraint to be satisfied, whereas Lems. 4 and 5 give two sufficient conditions for the independence constraint to be violated.

**Lemma 3.** *Suppose three random variables $V_1, V_2, V_3$ can be expressed as*

$$V_1 = c_1 e + e_1, \quad V_2 = c_2 e + e_2, \quad V_3 = c_3 e + e_3, \tag{24}$$

*where $e, (e_1, e_2), e_3$ are mutually independent, $c_1 c_2 c_3 \neq 0$, and $\mathrm{Var}(e) \neq 0$, then $(V_1, V_2, V_3)$ satisfies the independence constraint.*

*Proof.*

$$\mathrm{Cov}(V_1, V_3)\mathrm{Cov}(V_2, V_3) = c_1 c_2 c_3^2 (\mathrm{Var}(e))^2 \neq 0, \tag{25}$$

$$\mathrm{Cov}(V_2, V_3)V_1 - \mathrm{Cov}(V_1, V_3)V_2 = c_2 c_3 \mathrm{Var}(e)(c_1 e + e_1) - c_1 c_3 \mathrm{Var}(e)(c_2 e + e_2)$$

$$= c_2 c_3 \mathrm{Var}(e)e_1 - c_1 c_3 \mathrm{Var}(e)e_2 \perp\!\!\!\perp V_3. \tag{26}$$

Therefore, $(V_1, V_2, V_3)$ satisfies the independence constraint. □

**Lemma 4.** *Given three variables $V_1, V_2, V_3$, if $\exists V \in \mathbf{V}$ s.t. only one of $m_{V_1 V}$ and $m_{V_2 V}$ is non-zero and $m_{V_3 V}$ is non-zero, then $(V_1, V_2, V_3)$ does not satisfy the independence constraint.*

*Proof.* If $\mathrm{Cov}(V_1, V_3)\mathrm{Cov}(V_2, V_3) = 0$, $(V_1, V_2, V_3)$ does not satisfy the independence constraint. Otherwise, both $\mathrm{Cov}(V_2, V_3)V_1 - \mathrm{Cov}(V_1, V_3)V_2$ and $V_3$ contains $\epsilon_V$, so $\mathrm{Cov}(V_2, V_3)V_1 - \mathrm{Cov}(V_1, V_3)V_2 \not\perp\!\!\!\perp V_3$ based on Darmois-Skitovitch Theorem, that is, $(V_1, V_2, V_3)$ also does not satisfy the independence constraint. $\square$

**Lemma 5.** *Given three variables $V_1, V_2, V_3$, if $(V_1, V_2, V_3)$ satisfies the independence constraint and $m_{V_3 V_i} m_{V_3 V_j} \neq 0$, then $m_{V_1 V_i} m_{V_2 V_j} = m_{V_1 V_j} m_{V_2 V_i}$.*

*Proof.* We prove this lemma by contradiction. Suppose $m_{V_1 V_i} m_{V_2 V_j} \neq m_{V_1 V_j} m_{V_2 V_i}$,

$$\mathrm{Cov}(V_2, V_3)V_1 - \mathrm{Cov}(V_1, V_3)V_2$$
$$= (\mathrm{Cov}(V_2, V_3)m_{V_1 V_i} - \mathrm{Cov}(V_1, V_3)m_{V_2 V_i})\epsilon_{V_i} + (\mathrm{Cov}(V_2, V_3)m_{V_1 V_j} - \mathrm{Cov}(V_1, V_3)m_{V_2 V_j})\epsilon_{V_j} + \dots \tag{27}$$

Since $(V_1, V_2, V_3)$ satisfies the independence constraint, $\mathrm{Cov}(V_1, V_3)\mathrm{Cov}(V_2, V_3) \neq 0$, so $\mathrm{Cov}(V_2, V_3)V_1 - \mathrm{Cov}(V_1, V_3)V_2$ contains at least one of $\epsilon_{V_i}$ and $\epsilon_{V_j}$. Based on Darmois-Skitovitch Theorem, $\mathrm{Cov}(V_2, V_3)V_1 - \mathrm{Cov}(V_1, V_3)V_2 \not\perp\!\!\!\perp V_3$, which leads to contradiction. $\square$

## B    PROOF OF THEORETICAL RESULTS IN SEC. 3.2

**Theorem 1.** In Fig. 3(a), $m_{OT} = f(T, O, Z)$.

*Proof.* Note that

$$C_{3,1}(T, Z) = m_{TL}^3 m_{ZL} C_4(\epsilon_L) \neq 0, \tag{28}$$
$$C_{1,3}(T, Z) = m_{TL} m_{ZL}^3 C_4(\epsilon_L) \neq 0, \tag{29}$$
$$\mathrm{Cov}(T, Z) = m_{TL} m_{ZL} \mathrm{Var}(\epsilon_L) \neq 0, \tag{30}$$

so

$$g(T, Z) = \frac{m_{TL}}{m_{ZL}}. \tag{31}$$

Based on Eq. (4) and Thm. 1 in Kivva et al. (2023), this theorem holds. $\square$

**Theorem 2.** In Fig. 3(b), $\lambda = m_{OT}$ iff $(T, O - \lambda T)$ satisfies the cross-cumulant constraint and $(Z, T, O - \lambda T)$ satisfies the independence constraint.

*Proof.* Based on Lem. 2, $(T, O - \lambda T)$ satisfies the cross-cumulant constraint iff $\lambda = m_{OT}$ or $\frac{m_{OL}}{m_{TL}}$.

"Only if": if $\lambda = m_{OT}$,

$$Z = m_{ZL}\epsilon_L + \epsilon_Z, \tag{32}$$
$$T = m_{TL}\epsilon_L + m_{TZ}\epsilon_Z + \epsilon_T, \tag{33}$$
$$O - \lambda T = a_{OL}\epsilon_L + \epsilon_O. \tag{34}$$

Let $\epsilon_L$ be $e$ in Lem. 3, $(Z, T, O - \lambda T)$ satisfies the independence constraint.

"If": If $\lambda = m_{OL}/m_{TL}$,

$$Z = m_{ZL}\epsilon_L + \epsilon_Z, \tag{35}$$
$$T = m_{TL}\epsilon_L + m_{TZ}\epsilon_Z + \epsilon_T, \tag{36}$$
$$O - \lambda T = (m_{OZ} - \frac{m_{OL}}{m_{TL}}m_{TZ})\epsilon_Z + (m_{OT} - \frac{m_{OL}}{m_{TL}})\epsilon_T + \epsilon_O. \tag{37}$$

Note that $m_{OT} - \frac{m_{OL}}{m_{TL}} \neq 0$, $O - \lambda T$ contain $\epsilon_T$. Besides, $m_{TT} \neq 0$ and $m_{ZT} = 0$, so $(Z, T, O - \lambda T)$ does not satisfy the independence constraint based on Lem. 4. $\square$

**Theorem 3.** In Fig. 3(c), $m_{OT} = \lambda$ iff $(T, O - \lambda T)$ satisfies the cross-cumulant constraint and $(O - \lambda T, Z, T)$ satisfies the independence constraint.

*Proof.* Based on Lem. 2, $(T, O - \lambda T)$ satisfies the cross-cumulant constraint iff $\lambda = m_{OT}$ or $\frac{m_{OL}}{m_{TL}}$.

"Only if": If $\lambda = m_{OT}$, $(T, O - \lambda T)$ satisfies the cross-cumulant constraint based on Lem. 2. Besides,

$$O - \lambda T = (m_{OL} - m_{OT}m_{TL})\epsilon_L + m_{OZ}\epsilon_Z + \epsilon_O, \tag{38}$$

$$Z = m_{ZL}\epsilon_L + \epsilon_Z, \tag{39}$$

$$T = m_{TL}\epsilon_L + \epsilon_T. \tag{40}$$

Note that $m_{OL} - m_{OT}m_{TL} \neq 0$, let $\epsilon_L$ be $e$ in Lem. 3, $(O - \lambda T, Z, T)$ satisfies the independence constraint.

"If": If $\lambda = m_{OL}/m_{TL}$,

$$O - \lambda T = m_{OZ}\epsilon_Z + (m_{OT} - \frac{m_{OL}}{m_{TL}})\epsilon_T + \epsilon_O, \tag{41}$$

$$Z = m_{ZL}\epsilon_L + \epsilon_Z, \tag{42}$$

$$T = m_{TL}\epsilon_L + \epsilon_T. \tag{43}$$

Note that $m_{OT} - \frac{m_{OL}}{m_{TL}} \neq 0$, both $O - \lambda T$ and $T$ contain $\epsilon_T$ while $Z$ does not contain $\epsilon_T$, so $(O - \lambda T, Z, T)$ does not satisfy the independence constraint based on Lem. 4. $\qquad\square$

**Theorem 4.** In Fig. 3(d), $m_{OT} = f(T, O, Z - \lambda O)$ where $(O, Z - \lambda O)$ satisfies the cross-cumulant constraint and $(T, O, Z - \lambda O)$ satisfies the independence constraint.

*Proof.* Based on Thm. 2, $\lambda = m_{ZO}$ iff $(O, Z - \lambda O)$ satisfies the cross-cumulant constraint and $(T, O, Z - \lambda O)$ satisfies the independence constraint. Based on Thm. 1, $m_{OT} = f(T, O, Z - \lambda O)$. $\quad\square$

**Theorem 5.** In Fig. 3(e), $m_{OT} = f(T - \lambda_1 Z, O - \lambda_2 Z, Z)$ where both $(Z, T - \lambda_1 Z)$ and $(Z, O - \lambda_2 Z)$ satisfy the cross-cumulants constraint and $(T - \lambda_1 Z, O - \lambda_2 Z, Z)$ satisfies the independence constraint.

*Proof.* We first prove $(\lambda_1, \lambda_2) = (m_{TZ}, m_{OZ})$ or $(m_{TL}/m_{ZL}, m_{OL}/m_{ZL})$ iff both $(Z, T - \lambda_1 Z)$ and $(Z, O - \lambda_2 Z)$ satisfy the cross-cumulants constraint and $(T - \lambda_1 Z, O - \lambda_2 Z, Z)$ satisfies the independence constraint. Based on Lem. 2, $(Z, T - \lambda_1 Z)$ satisfies the cross-cumulant constraint iff $\lambda = m_{TZ}$ or $\frac{m_{TL}}{m_{ZL}}$, $(Z, O - \lambda_2 Z)$ satisfies the cross-cumulants constraint iff $\lambda = m_{OZ}$ or $\frac{m_{OL}}{m_{ZL}}$.

"Only if": If $(\lambda_1, \lambda_2) = (m_{TZ}, m_{OZ})$, both $(Z, T - \lambda_1 Z)$ and $(Z, O - \lambda_2 Z)$ satisfy the cross-cumulant constraint based on Lem. 2. Besides,

$$T - \lambda_1 Z = a_{TL}\epsilon_L + \epsilon_T, \tag{44}$$

$$O - \lambda_2 Z = (m_{OL} - m_{OZ}m_{ZL})\epsilon_L + m_{OT}\epsilon_T + \epsilon_O, \tag{45}$$

$$Z = m_{ZL}\epsilon_L + \epsilon_Z. \tag{46}$$

Note that $m_{OL} - m_{OZ}m_{ZL} \neq 0$, let $\epsilon_L$ be $e$ in Lem. 3, $(T - \lambda_1 Z, O - \lambda_2 Z, Z)$ satisfies the independence constraint.

If $(\lambda_1, \lambda_2) = (m_{TL}/m_{ZL}, m_{OL}/m_{ZL})$,

$$T - \lambda_1 Z = (m_{TZ} - \frac{m_{TL}}{m_{ZL}})\epsilon_Z + \epsilon_T, \tag{47}$$

$$O - \lambda_2 Z = (m_{OZ} - \frac{m_{OL}}{m_{ZL}})\epsilon_Z + m_{OT}\epsilon_T + \epsilon_O, \tag{48}$$

$$Z = m_{ZL}\epsilon_L + \epsilon_Z. \tag{49}$$

Note that $(m_{TZ} - \frac{m_{TL}}{m_{ZL}})(m_{OZ} - \frac{m_{OL}}{m_{ZL}}) \neq 0$, let $\epsilon_Z$ be $e$ in Lem. 3, $(T - \lambda_1 Z, O - \lambda_2 Z, Z)$ satisfies the independence constraint.

"If": If $(\lambda_1, \lambda_2) = (m_{TZ}, m_{OL}/m_{ZL})$, based on Eqs. (44) and (48), $T - \lambda_1 Z$ contains $\epsilon_L$ while $O - \lambda_2 Z$ does not contain $\epsilon_L$, note that $m_{ZL} \neq 0$, so $(T - \lambda_1 Z, O - \lambda_2 Z, Z)$ does not satisfy the independence constraint based on Lem. 4.

If $(\lambda_1, \lambda_2) = (m_{TL}/m_{ZL}, m_{OZ})$, based on Eqs. (47) and (45), $T - \lambda_1 Z$ does not contains $\epsilon_L$ while $O - \lambda_2 Z$ contains $\epsilon_L$, note that $m_{ZL} \neq 0$, so $(T - \lambda_1 Z, O - \lambda_2 Z, Z)$ does not satisfy the independence constraint based on Lem. 4. Therefore, $(\lambda_1, \lambda_2) = (m_{TZ}, m_{OZ})$ or $(m_{TL}/m_{ZL}, m_{OL}/m_{ZL})$.

In the following, we prove $m_{OT} = f(T - \lambda_1 Z, O - \lambda_2 Z, Z)$. Denote $T - \lambda_1 Z, O - \lambda_2 Z$ by $T', O'$. If $(\lambda_1, \lambda_2) = (m_{TZ}, m_{OZ})$, based on Eqs. (44) and (45), $L, Z, T', O'$ can be expressed as

$$
\begin{cases}
L = \epsilon_L, \\
Z = a_{ZL}L + \epsilon_Z, \\
T' = a_{TL}L + \epsilon_T, \\
O' = a_{OL}L + m_{OT}T' + \epsilon_O,
\end{cases} \quad ; \tag{50}
$$

if $(m_{TL}/m_{ZL}, m_{OL}/m_{ZL})$, based on Eqs. (47) and (48), $L, Z, T', O'$ can be expressed as

$$
\begin{cases}
L = \epsilon_Z, \\
Z = L + a_{ZL}\epsilon_L, \\
T' = -\frac{a_{TL}}{m_{ZL}}L + \epsilon_T, \\
O' = -\frac{a_{OL}}{m_{ZL}}L + m_{OT}T' + \epsilon_O,
\end{cases} \quad ; \tag{51}
$$

In both cases, $Z \perp\!\!\!\perp (T', O')|L$, that is, $Z$ is an isolated proxy. Also, it is obvious that $m_{OT} = m_{O'T'}$, so $m_{OT} = f(T - \lambda_1 Z, O - \lambda_2 Z, Z)$ based on Thm. 1. □

**Corollary 1.** The ground truth is Fig. 3(a) iff $(T, O, Z)$ satisfies the independence constraint.

*Proof.* "Only if": In Fig. 3(a),

$$
Z = m_{ZL}\epsilon_L + \epsilon_Z, \tag{52}
$$

$$
T = m_{TL}\epsilon_L + \epsilon_T, \tag{53}
$$

$$
O = m_{OL}\epsilon_L + m_{OT}\epsilon_T + \epsilon_O. \tag{54}
$$

Let $\epsilon_L$ be $e$ in Lem. 3, $(T, O, Z)$ satisfies the independence constraint.

"If": We prove this part by contradiction.

- In Fig. 3(b) and (e), $\{L, Z\} \subseteq \overline{\text{An}}(T) \cap \overline{\text{An}}(O)$ and there exist two non-intersecting directed paths from $\{L, Z\}$ to $\{T, O\}$ ($Z \to T$ and $L \to O$), so $m_{TL}/m_{TZ} \neq m_{OL}/m_{OZ}$. Besides, $m_{ZL}m_{ZZ} \neq 0$. Based on Lem. 5, $(T, O, Z)$ does not satisfy the independence constraint, which leads to contradiction.

- In Fig. 3(c) and (g), $m_{TZ} = 0$ and $m_{ZZ}m_{OZ} \neq 0$. Based on Lem. 4, $(T, O, Z)$ does not satisfy the independence constraint, which leads to contradiction.

- In Fig. 3(d) and (h), $m_{TO} = 0$ and $m_{OO}m_{ZO} \neq 0$. Based on Lem. 4, $(T, O, Z)$ does not satisfy the independence constraint, which leads to contradiction.

- In Fig. 3(f), $\{L, T\} \subseteq \overline{\text{An}}(T) \cap \overline{\text{An}}(O)$ and there exist two non-intersecting directed paths from $\{L, T\}$ to $\{T, O\}$ ($T$ and $L \to O$), so $m_{TL}/m_{TT} \neq m_{OL}/m_{OT}$. Besides, $m_{ZL}m_{ZT} \neq 0$. Based on Lem. 5, $(T, O, Z)$ does not satisfy the independence constraint, which leads to contradiction.

□

**Corollary 2.** If the underlying causal structure is not Fig. 3(a), it is Fig. 3(b) iff there exists one and only one $\lambda \neq 0$ s.t. the constraints in Thm. 2 are satisfied.

*Proof.* "Only if": this part can be proven by Thm. 2.

"If": We prove this part by contradiction.

- In Fig. 3(c) and (g), $m_{TZ} = 0$ and $m_{ZZ} \neq 0$. In particular, for any $\lambda \neq 0$, $O - \lambda T$ contains $\epsilon_Z$. Based on Lem. 4, $(Z, T, O - \lambda T)$ does not satisfy the independence constraint, which leads to contradiction.

- In Fig. 3(d) and (h), $m_{TO} = 0$ and $m_{ZO} \neq 0$. In particular, for any $\lambda \neq 0$, $O - \lambda T$ contains $\epsilon_O$. Based on Lem. 4, $(Z, T, O - \lambda T)$ does not satisfy the independence constraint, which leads to contradiction.

- In Fig. 3(e), if $\lambda \neq m_{OT}$, $O - \lambda T$ contains $\epsilon_T$, note that $m_{ZT} = 0$ and $m_{TT} \neq 0$, so $(Z, T, O - \lambda T)$ does not satisfy the independence constraint based on Lem. 4; otherwise, $O - \lambda T$ contain both $\epsilon_L$ and $\epsilon_Z$, since $\{L, Z\} \subseteq \overline{\text{An}}(Z) \cap \overline{\text{An}}(T)$ and there exist two non-intersecting directed paths from $\{L, Z\}$ to $\{Z, T\}$ ($Z$ and $L \to T$), so $m_{ZL}/m_{TL} \neq m_{ZZ}/m_{TZ}$. Based on Lem. 5, $(Z, T, O - \lambda T)$ does not satisfy the independence constraint, which leads to contradiction.

- In Fig. 3(f), based on Lem. 2, $(T, O - \lambda T)$ satisfies the cross-cumulant constraint iff $\lambda = m_{OT}$ or $m_{OL}/m_{TL}$. If $\lambda = m_{OT}$,

$$Z = m_{ZL}\epsilon_L + m_{ZT}\epsilon_T + \epsilon_Z, \tag{55}$$

$$T = m_{TL}\epsilon_L + \epsilon_T, \tag{56}$$

$$O - \lambda T = a_{OL}\epsilon_L + \epsilon_O. \tag{57}$$

Let $\epsilon_L$ be $e$ in Lem. 3, $(Z, T, O - \lambda T)$ satisfies the independence constraint. If $\lambda = m_{OL}/m_{TL}$,

$$Z = m_{ZL}\epsilon_L + m_{ZT}\epsilon_T + \epsilon_Z, \tag{58}$$

$$T = m_{TL}\epsilon_L + \epsilon_T, \tag{59}$$

$$O - \lambda T = (m_{OT} - \frac{m_{OL}}{m_{TL}})\epsilon_T + \epsilon_O. \tag{60}$$

Note that $m_{OT} - \frac{m_{OL}}{m_{TL}} \neq 0$, let $\epsilon_T$ be $e$ in Lem. 3, $(Z, T, O - \lambda T)$ satisfies the independence constraint. Therefore, there exist two $\lambda$ s.t. $(T, O - \lambda T)$ satisfies the cross-cumulant constraint and $(Z, T, O - \lambda T)$ satisfies the independence constraint, which leads to contradiction.

$\square$

**Corollary 3.** If the underlying causal structure is not one of Fig. 3(a)-(b), it is Fig. 3(c) iff there exists $\lambda \neq 0$ s.t. the constraints in Thm. 3 are satisfied.

*Proof.* "Only if": this part can be proven by Thm. 3.

"If": we prove this part by contradiction.

- In Fig. 3(d), (f), (g), and (h), based on Lem. 2, $(T, O - \lambda T)$ satisfies the cross-cumulant constraint iff $\lambda = m_{OT}$ or $m_{OL}/m_{TL}$. If $\lambda = m_{OT}$, $O - \lambda T$ does not contain $\epsilon_T$, note that $m_{TT}m_{ZT} \neq 0$, so $(O - \lambda T, Z, T)$ does not satisfy the independence constraint based on Lem. 4; if $\lambda = m_{OL}/m_{TL}$, $O - \lambda T$ does not contain $\epsilon_L$, note that $m_{TL}m_{ZL} \neq 0$, so $(O - \lambda T, Z, T)$ does not satisfy the independence constraint based on Lem. 4, which leads to contradiction.

- In Fig. 3(e), if $\lambda \neq m_{OT}$, $O - \lambda T$ contains $\epsilon_T$, note that $m_{ZT} = 0$ and $m_{TT} \neq 0$, so $(O - \lambda T, Z, T)$ does not satisfy the independence constraint based on Lem. 4; otherwise, $O - \lambda T = a_{OL}L + a_{OZ}Z + \epsilon_O$, denote $O - \lambda T$ by $O'$, since $\{L, Z\} \subseteq \overline{\text{An}}(Z) \cap \overline{\text{An}}(O')$ and there exist two non-intersecting directed paths from $\{L, Z\}$ to $\{Z, O'\}$ ($Z$ and $L \to O'$), so $m_{ZL}/m_{O'L} \neq m_{ZZ}/m_{O'Z}$. Besides, $m_{TL}m_{TZ} \neq 0$. Based on Lem. 5, $(O - \lambda T, Z, T)$ does not satisfy the independence constraint, which leads to contradiction.

$\square$

**Corollary 4.** If the underlying causal structure is not one of Fig. 3(a)-(d), it is Fig. 3(d) iff there exists $\lambda \neq 0$ s.t. the constraints in Thm. 4 are satisfied.

*Proof.* "Only if": this part can be proven by Thm. 4.

"If": we prove this part by contradiction.

- In Fig. 3(e), (f), and (g), $m_{TO} = 0$ and $m_{OO} \neq 0$. In particular, for any $\lambda \neq 0$, $Z - \lambda O$ contains $\epsilon_O$. Based on Lem. 4, $(T, O, Z - \lambda O)$ does not satisfy the independence constraint, which leads to contradiction.

- In Fig. 3(h), if $\lambda \neq m_{ZO}$, $Z - \lambda O$ contains $\epsilon_O$, note that $m_{TO} = 0$ and $m_{OO} \neq 0$, so $(T, O, Z - \lambda O)$ does not satisfy the independence constraint based on Lem. 4; otherwise, $Z - \lambda O$ contains both $\epsilon_L$ and $\epsilon_T$, since $\{L, T\} \subseteq \overline{\text{An}}(T) \cap \overline{\text{An}}(O)$ and there exist two non-intersecting directed paths from $\{L, T\}$ to $\{T, O\}$ ($T$ and $L \to O$), so $m_{TL}/m_{OL} \neq m_{TT}/m_{OT}$. Based on Lem. 5, $(T, O, Z - \lambda O)$ does not satisfy the independence constraint, which leads to contradiction.

$\square$

**Corollary 5.** If the underlying causal structure is not one of Fig. 3(a)-(c), it is Fig. 3(d) iff there exist $\lambda_1 \neq 0$ and $\lambda_2 \neq 0$ s.t. the constraints in Thm. 5 are satisfied.

*Proof.* "Only if": this part can be proven by Thm. 5.

"If": We prove this part by contradiction.

- In Fig. 3(f), $Z$ contains $\epsilon_L, \epsilon_T, \epsilon_Z$, and

$$T - \lambda_1 Z = (m_{TL} - \lambda_1 m_{ZL})\epsilon_L + (1 - \lambda_1 m_{ZT})\epsilon_T - \lambda_1 \epsilon_Z, \tag{61}$$
$$O - \lambda_2 Z = (m_{OL} - \lambda_2 m_{ZL})\epsilon_L + (m_{OT} - \lambda_2 m_{ZT})\epsilon_T - \lambda_2 \epsilon_Z + \epsilon_O. \tag{62}$$

Suppose $(T - \lambda_1 Z, O - \lambda_2 Z, Z)$ satisfies the independence constraint, based on Lem. 5,

$$(1 - \lambda_1 m_{ZT})(-\lambda_2) = (m_{OT} - \lambda_2 m_{ZT})(-\lambda_1), \tag{63}$$
$$(m_{TL} - \lambda_1 m_{ZL})(-\lambda_2) = (m_{OL} - \lambda_2 m_{ZL})(-\lambda_1). \tag{64}$$

From Eqs. (63) and (64), we can derive $m_{OL} = m_{OT} m_{TL}$. Note that $m_{OL} = m_{OT} m_{TL} + a_{OL}$ in Fig. 3(f), this leads to contradiction.

- In Fig. 3(g), $Z$ contains $\epsilon_L, \epsilon_T, \epsilon_Z$, and

$$T - \lambda_1 Z = (m_{TL} - \lambda_1 m_{ZL})\epsilon_L + (1 - \lambda_1 m_{ZT})\epsilon_T - \lambda_1 \epsilon_Z, \tag{65}$$
$$O - \lambda_2 Z = (m_{OL} - \lambda_2 m_{ZL})\epsilon_L + (m_{OT} - \lambda_2 m_{ZT})\epsilon_T + (m_{OZ} - \lambda_2)\epsilon_Z + \epsilon_O. \tag{66}$$

Suppose $(T - \lambda_1 Z, O - \lambda_2 Z, Z)$ satisfies the independence constraint, based on Lem. 5,

$$(1 - \lambda_1 m_{ZT})(m_{OZ} - \lambda_2) = (m_{OT} - \lambda_2 m_{ZT})(-\lambda_1), \tag{67}$$
$$(m_{TL} - \lambda_1 m_{ZL})(m_{OZ} - \lambda_2) = (m_{OL} - \lambda_2 m_{ZL})(-\lambda_1). \tag{68}$$

From Eqs. (67) and (68), we can derive $\frac{m_{OL} - m_{OZ} m_{ZL}}{m_{OT} - m_{OZ} m_{ZT}} = m_{TL}$. Note that $m_{OL} - m_{OZ} m_{ZL} = a_{OL} + a_{OT} m_{TL}$ and $m_{OT} - m_{OZ} m_{ZT} = a_{OT}$ in Fig. 3(g), this leads to contradiction.

- In Fig. 3(h), $Z$ contains $\epsilon_L, \epsilon_T, \epsilon_O, \epsilon_Z$, and

$$T - \lambda_1 Z = (m_{TL} - \lambda_1 m_{ZL})\epsilon_L + (1 - \lambda_1 m_{ZT})\epsilon_T - \lambda_1 m_{ZO}\epsilon_O - \lambda_1 \epsilon_Z, \tag{69}$$
$$O - \lambda_2 Z = (m_{OL} - \lambda_2 m_{ZL})\epsilon_L + (m_{OT} - \lambda_2 m_{ZT})\epsilon_T + (1 - \lambda_2 m_{ZO})\epsilon_O - \lambda_2 \epsilon_Z. \tag{70}$$

Suppose $(T - \lambda_1 Z, O - \lambda_2 Z, Z)$ satisfies the independence constraint, based on Lem. 5,

$$(-\lambda_1 m_{ZO})(-\lambda_2) = (-\lambda_1)(1 - \lambda_2 m_{ZO}), \tag{71}$$

that is, $\lambda_1 = 0$, which leads to contradiction.

$\square$

## C  EXTENSION 1: MULTIPLE LATENT CONFOUNDERS

### C.1  THEORETICAL RESULTS

For ease of exposition, we denote the set of $Z_i$ whose causal relationship with $(T, O)$ follows Fig. 3(a), Fig. 3(b), ...,Fig. 3(h) by $\mathbf{Z}_a, \mathbf{Z}_b, ..., \mathbf{Z}_h$. Using this notation, in Fig. 4, we have $Z_1 \in \mathbf{Z}_a, Z_2 \in \mathbf{Z}_b, ..., Z_8 \in \mathbf{Z}_h$. In addition, we abbreviate $\text{An}(T) \cap \mathbf{Z}, \text{An}(O) \cap \mathbf{Z}, \text{Pa}(T) \cap \mathbf{Z}, \text{Pa}(O) \cap \mathbf{Z}$ by $\text{An}_{\mathbf{Z}}(T), \text{An}_{\mathbf{Z}}(O), \text{Pa}_{\mathbf{Z}}(T), \text{Pa}_{\mathbf{Z}}(O)$ respectively.

Thm. 8 establishes the sufficient and necessary condition for $Z_i \in \mathbf{Z}_a$.

**Theorem 8.** $Z_i \in \mathbf{Z}_a$ iff $(T, O, Z_i)$ satisfies the independence constraint.

*Proof.* "Only if": if $Z_i \in \mathbf{Z}_a$,

$$T = m_{TL_i}\epsilon_{L_i} + \sum_{L \in \mathbf{L} \setminus \{L_i\}} m_{TL}\epsilon_L + \sum_{Z \in \mathrm{An}_{\mathbf{Z}}(T)} m_{TZ}\epsilon_Z + \epsilon_T, \tag{72}$$

$$O = m_{OL_i}\epsilon_{L_i} + \sum_{L \in \mathbf{L} \setminus \{L_i\}} m_{OL}\epsilon_L + \sum_{Z \in \mathrm{An}_{\mathbf{Z}}(O)} m_{OZ}\epsilon_Z + m_{OT}\epsilon_T + \epsilon_O, \tag{73}$$

$$Z_i = m_{Z_i L_i}\epsilon_{L_i} + \epsilon_{Z_i}. \tag{74}$$

Note that $Z_i \notin \mathrm{An}_{\mathbf{Z}}(T) \cup \mathrm{An}_{\mathbf{Z}}(O)$, let $\epsilon_{L_i}$ be $e$ in Lem. 3, $(T, O, Z_i)$ satisfies the independence constraint.

"If": We prove this part by contradiction.

- If $Z_i \in \mathbf{Z}_b \cup \mathbf{Z}_e$, $\{L_i, Z_i\} \subseteq \overline{\mathrm{An}}(T) \cap \overline{\mathrm{An}}(O)$ and there exist two non-intersecting directed paths from $\{L_i, Z_i\}$ to $\{T, O\}$ ($Z_i \to T$ and $L_i \to O$), so $m_{TL_i}/m_{TZ_i} \neq m_{OL_i}/m_{OZ_i}$. Besides, $m_{Z_i L_i} m_{Z_i Z_i} \neq 0$. Based on Lem. 5, $(T, O, Z_i)$ does not satisfy the independence constraint, which leads to contradiction.

- If $Z_i \in \mathbf{Z}_c \cup \mathbf{Z}_g$, $m_{TZ_i} = 0$ and $m_{Z_i Z_i} m_{OZ_i} \neq 0$. Based on Lem. 4, $(T, O, Z_i)$ does not satisfy the independence constraint, which leads to contradiction.

- If $Z_i \in \mathbf{Z}_d \cup \mathbf{Z}_h$, $m_{TO} = 0$ and $m_{OO} m_{Z_i O} \neq 0$. Based on Lem. 4, $(T, O, Z_i)$ does not satisfy the independence constraint, which leads to contradiction.

- If $Z_i \in \mathbf{Z}_f$, $\{L_i, T\} \subseteq \overline{\mathrm{An}}(T) \cap \overline{\mathrm{An}}(O)$ and there exist two non-intersecting directed paths from $\{L_i, T\}$ to $\{T, O\}$ ($T$ and $L_i \to O$), so $m_{TL_i}/m_{TT} \neq m_{OL_i}/m_{OT}$. Besides, $m_{Z_i L} m_{Z_i T} \neq 0$. Based on Lem. 5, $(T, O, Z_i)$ does not satisfy the independence constraint, which leads to contradiction.

$\square$

Thm. 9 establishes the sufficient and necessary condition for $Z_i \in \mathbf{Z}_c$. Furthermore, it gives the value of $m_{OZ_i}$ when $Z_i \in \mathbf{Z}_c$.

**Theorem 9.** *Suppose $Z_i \notin \mathbf{Z}_a$, $Z_i \in \mathbf{Z}_c$ iff there exists $\lambda \neq 0$ s.t. $(Z_i, O - \lambda Z_i)$ satisfies the cross-cumulant constraint and $(O - \lambda Z_i, T, Z_i)$ satisfies the independence constraint. Furthermore, if $Z_i \in \mathbf{Z}_c$, $\lambda = m_{OZ_i}$ iff the above constraints are satisfied.*

*Proof.* "If": We prove this part by contradiction.

- If $Z_i \in \mathbf{Z}_d \cup \mathbf{Z}_f \cup \mathbf{Z}_h$, $m_{TZ_i} = 0$ and $m_{Z_i Z_i} \neq 0$. In particular, for any $\lambda \neq 0$, $O - \lambda Z_i$ contains $\epsilon_{Z_i}$. Based on Lem. 4, $(O - \lambda Z_i, T, Z_i)$ does not satisfy the independence constraint, which leads to contradiction.

- If $Z_i \in \mathbf{Z}_b \cup \mathbf{Z}_e$, based on Lem. 2, $(Z_i, O - \lambda Z_i)$ satisfies the cross-cumulant constraint iff $\lambda = m_{OZ_i}$ or $m_{OL_i}/m_{Z_i L_i}$. If $\lambda = m_{OZ_i}$, $O - \lambda Z_i$ does not contain $\epsilon_{Z_i}$, note that $m_{TZ_i} m_{Z_i Z_i} \neq 0$, so $(O - \lambda Z_i, T, Z_i)$ does not satisfy the independence constraint based on Lem. 4, which leads to contradiction; if $\lambda = m_{OL_i}/m_{Z_i L_i}$, $O - \lambda Z_i$ does not contain $\epsilon_{L_i}$, note that $m_{TL_i} m_{Z_i L_i} \neq 0$, so $(O - \lambda Z_i, T, Z_i)$ does not satisfy the independence constraint based on Lem. 4, which leads to contradiction..

- If $Z_i \in \mathbf{Z}_g$, we discuss two cases. If $\lambda \neq m_{OZ_i}$, $O - \lambda Z_i$ contains $\epsilon_{Z_i}$, note that $m_{TZ_i} = 0$ and $m_{Z_i Z_i} \neq 0$, so $(O - \lambda Z_i, T, Z_i)$ does not satisfy the independence constraint based on Lem. 4, which leads to contradiction. If $\lambda = m_{OZ_i}$, $O - \lambda Z_i = a_{OL_i} L_i + a_{OT} T + \sum_{L \in \mathbf{L} \setminus \{L_i\}} a_{OL} L + \sum_{Z \in \mathrm{Pa}_{\mathbf{Z}}(O) \setminus \{Z_i\}} a_{OZ} Z + \epsilon_O$, denote $O - \lambda Z_i$ by $O'$, since $\{L_i, T\} \subseteq \overline{\mathrm{An}}(O') \cap \overline{\mathrm{An}}(T)$ and there exist two non-intersecting directed paths from $\{L_i, T\}$ to $\{T, O'\}$ ($T$ and $L_i \to O'$), so $m_{TL_i}/m_{O'L_i} \neq m_{TT}/m_{O'T}$. Besides, $m_{Z_i L_i} m_{Z_i T} \neq 0$. Based on Lem. 5, $(O - \lambda Z_i, T, Z_i)$ does not satisfy the independence constraint, which leads to contradiction.

"Only if": if $Z_i \in \mathbf{Z}_c$, let $\lambda = m_{OZ_i}$, based on Lem. 2, $(Z_i, O - \lambda Z_i)$ satisfies the cross-cumulant constraint. Besides,

$$O - \lambda Z_i = (m_{OL_i} - m_{OZ_i} m_{Z_i L_i}) \epsilon_{L_i}$$

$$+ \sum_{L \in \mathbf{L} \setminus \{L_i\}} m_{OL} \epsilon_L + \sum_{Z \in \mathrm{An}_{\mathbf{Z}}(O) \setminus \{Z_i\}} m_{OZ} \epsilon_Z + m_{OT} \epsilon_T + \epsilon_O, \tag{75}$$

$$T = m_{TL_i} \epsilon_{L_i} + \sum_{L \in \mathbf{L} \setminus \{L_i\}} m_{TL} \epsilon_L + \sum_{Z \in \mathrm{An}_{\mathbf{Z}}(T)} m_{TZ} \epsilon_Z + \epsilon_T, \tag{76}$$

$$Z_i = m_{Z_i L_i} \epsilon_{L_i} + \epsilon_{Z_i}. \tag{77}$$

Note that $m_{OL_i} - m_{OZ_i} m_{Z_i L_i} \neq 0$ and $Z_i \notin \mathrm{An}_{\mathbf{Z}}(T)$, let $\epsilon_{L_i}$ be $e$ in Lem. 3, $(O - \lambda Z_i, T, Z_i)$ satisfies the independence constraint.

Furthermore, if $Z_i \in \mathbf{Z}_c$ but $\lambda \neq m_{OZ_i}$, $O - \lambda Z_i$ contains $\epsilon_{Z_i}$, note that $m_{TZ_i} = 0$ and $m_{Z_i Z_i} \neq 0$, $(O - \lambda Z_i, T, Z_i)$ does not satisfy the independence constraint based on Lem. 4. $\qquad \square$

Thm. 10 establishes the sufficient and necessary condition for $Z_i \in \mathbf{Z}_d$. Furthermore, it gives the value of $m_{Z_i O}$ when $Z_i \in \mathbf{Z}_d$.

**Theorem 10.** *Suppose $Z_i \notin \mathbf{Z}_a \cup \mathbf{Z}_c$, $Z_i \in \mathbf{Z}_d$ iff there exists $\lambda \neq 0$ s.t. $(O, Z_i - \lambda O)$ satisfies the cross-cumulant constraint and $(T, O, Z_i - \lambda O)$ satisfies the independence constraint. Furthermore, if $Z_i \in \mathbf{Z}_d$, $\lambda = m_{Z_i O}$ iff the above constraints are satisfied.*

*Proof.* "If": We prove this part by contradiction.

- If $Z_i \in \mathbf{Z}_b \cup \mathbf{Z}_e \cup \mathbf{Z}_f \cup \mathbf{Z}_g$, $m_{TO} = 0$ and $m_{OO} \neq 0$. In particular, for any $\lambda \neq 0$, $Z_i - \lambda O$ contains $\epsilon_O$. Based on Lem. 4, $(T, O, Z_i - \lambda O)$ does not satisfy the independence constraint, which leads to contradiction.

- If $Z_i \in \mathbf{Z}_h$, we discuss two cases. If $\lambda \neq m_{Z_i O}$, $Z_i - \lambda O$ contains $\epsilon_{Z_i}$, note that $m_{TO} = 0$ and $m_{OO} \neq 0$, so $(T, O, Z_i - \lambda O)$ does not satisfy the independence constraint based on Lem. 4, which leads to contradiction. If $\lambda = m_{Z_i O}$, $Z_i - \lambda O = a_{Z_i T} T + a_{Z_i L_i} L_i + \epsilon_{Z_i}$, which contains both $\epsilon_{L_i}$ and $\epsilon_T$. Since $\{L_i, T\} \subseteq \overline{\mathrm{An}}(T) \cap \overline{\mathrm{An}}(O)$ and there exist two non-intersecting directed paths from $\{L_i, T\}$ to $\{T, O\}$ ($T$ and $L_i \to O$), $m_{TL_i}/m_{OL_i} \neq m_{TT}/m_{OT}$. Based on Lem. 5, $(T, O, Z_i - \lambda O)$ does not satisfy the independence constraint, which leads to contradiction.

"Only if": if $Z_i \in \mathbf{Z}_d$, let $\lambda = m_{Z_i O}$, based on Lem. 2, $(O, Z_i - \lambda O)$ satisfies the cross-cumulant constraint. Besides,

$$T = m_{TL_i} \epsilon_{L_i} + \sum_{L \in \mathbf{L} \setminus \{L_i\}} m_{TL} \epsilon_L + \sum_{Z \in \mathrm{An}_{\mathbf{Z}}(T)} m_{TZ} \epsilon_Z + \epsilon_T, \tag{78}$$

$$O = m_{OL_i} \epsilon_{L_i} + \sum_{L \in \mathbf{L} \setminus \{L_i\}} m_{OL} \epsilon_L + \sum_{Z \in \mathrm{An}_{\mathbf{Z}}(O)} m_{OZ} \epsilon_Z + m_{OT} \epsilon_T + \epsilon_O, \tag{79}$$

$$Z_i - \lambda O = a_{Z_i L_i} \epsilon_{L_i} + \epsilon_{Z_i} \tag{80}$$

Note that $Z_i \notin \mathrm{An}_{\mathbf{Z}}(T) \cup \mathrm{An}_{\mathbf{Z}}(O)$, let $\epsilon_{L_i}$ be $e$ in Lem. 3, $(T, O, Z_i - \lambda O)$ satisfies the independence constraint.

Furthermore, if $Z_i \in \mathbf{Z}_d$ but $\lambda \neq m_{Z_i O}$, $Z_i - \lambda O$ contains $\epsilon_O$, note that $m_{TO} = 0$ and $m_{OO} \neq 0$, $(T, O, Z_i - \lambda O)$ does not satisfy the independence constraint based on Lem. 4. $\qquad \square$

Thm. 11 establishes the sufficient and necessary condition for $Z_i \in \mathbf{Z}_f$. Furthermore, it gives the value of $m_{Z_i T}$ when $Z_i \in \mathbf{Z}_f$.

**Theorem 11.** *Suppose $Z_i \notin \mathbf{Z}_a \cup \mathbf{Z}_c \cup \mathbf{Z}_d$, $Z_i \in \mathbf{Z}_f$ iff there exists $\lambda \neq 0$ s.t. $(T, Z_i - \lambda T)$ satisfies the cross-cumulant constraint and $(T, O, Z_i - \lambda T)$ satisfies the independence constraint. Furthermore, if $Z_i \in \mathbf{Z}_f$, $\lambda = m_{Z_i T}$ iff the above constraints are satisfied.*

*Proof.* "If": we prove this part by contradiction.

- If $Z_i \in \mathbf{Z}_b \cup \mathbf{Z}_e$, let $L_j \in \mathbf{L} \backslash \{L_i\}$, since $\{T, L_j\} \subseteq \overline{\mathrm{An}}(T) \cap \overline{\mathrm{An}}(O)$ and there exist two non-intersecting directed paths from $\{T, L_j\}$ to $\{T, O\}$ ($T$ and $L_j \to O$), so $m_{TT}/m_{TL_j} \neq m_{OT}/m_{OL_j}$. Besides, for any $\lambda \neq 0$, $Z_i - \lambda T$ contains $\epsilon_T$ and $\epsilon_{L_j}$, based on Lem. 5, $(T, O, Z_i - \lambda T)$ does not satisfy the independence constraint, which leads to contradiction.

- If $Z_i \in \mathbf{Z}_g$, $m_{TZ_i} = 0$ and $m_{OZ_i} \neq 0$. In particular, for any $\lambda \neq 0$, $Z_i - \lambda T$ contains $\epsilon_{Z_i}$. Based on Lem. 4, $(T, O, Z_i - \lambda T)$ does not satisfy the independence constraint, which leads to contradiction.

- If $Z_i \in \mathbf{Z}_h$, $m_{TO} = 0$ and $m_{OO} \neq 0$. In particular, for any $\lambda \neq 0$, $Z_i - \lambda T$ contains $\epsilon_O$. Based on Lem. 4, $(T, O, Z_i - \lambda T)$ does not satisfy the independence constraint, which leads to contradiction.

"Only if": if $Z_i \in \mathbf{Z}_f$, let $\lambda = m_{Z_i T}$, based on Lem. 2, $(T, Z_i - \lambda T)$ satisfies the cross-cumulant constraint. Besides,

$$T = m_{TL_i} \epsilon_{L_i} + \sum_{L \in \mathbf{L} \backslash \{L_i\}} m_{TL} \epsilon_L + \sum_{Z \in \mathrm{An}_{\mathbf{Z}}(T)} m_{TZ} \epsilon_Z + \epsilon_T, \tag{81}$$

$$O = m_{OL_i} \epsilon_{L_i} + \sum_{L \in \mathbf{L} \backslash \{L_i\}} m_{OL} \epsilon_L + \sum_{Z \in \mathrm{An}_{\mathbf{Z}}(O)} m_{OZ} \epsilon_Z + m_{OT} \epsilon_T + \epsilon_O, \tag{82}$$

$$Z_i - \lambda T = a_{Z_i L_i} \epsilon_{L_i} + \epsilon_{Z_i} \tag{83}$$

Note that $Z_i \notin \mathrm{An}_{\mathbf{Z}}(T) \cup \mathrm{An}_{\mathbf{Z}}(O)$, let $\epsilon_{L_i}$ be $e$ in Lem. 3, $(T, O, Z_i - \lambda T)$ satisfies the independence constraint.

Furthermore, If $\lambda \neq m_{Z_i T}$, $Z_i - \lambda T$ contains $\epsilon_T$. Besides, let $L_j \in \mathbf{L} \backslash \{L_i\}$, $Z_i - \lambda T$ contains $\epsilon_{L_j}$. Since $\{L_j, T\} \subseteq \overline{\mathrm{An}}(T) \cap \overline{\mathrm{An}}(O)$ and there exist two non-intersecting directed paths from $\{L_j, T\}$ to $\{T, O\}$ ($T$ and $L_j \to O$), $m_{TL_j}/m_{OL_j} \neq m_{TT}/m_{OT}$. Based on Lem. 5, $(T, O, Z_i - \lambda T)$ does not satisfy the independence constraint. □

Thm. 12 establishes the sufficient and necessary condition for $Z_i \in \mathbf{Z}_b \cup \mathbf{Z}_e$. Furthermore, it gives the values of $(m_{TZ_i}, m_{OZ_i})$ and $(m_{TL_i}/m_{Z_i L_i}, m_{OL_i}/m_{Z_i L_i})$ when $Z_i \in \mathbf{Z}_f$.

**Theorem 12.** *Suppose $Z_i \notin \mathbf{Z}_a \cup \mathbf{Z}_c \cup \mathbf{Z}_d \cup \mathbf{Z}_f$, $Z_i \in \mathbf{Z}_b \cup \mathbf{Z}_e$ iff there exist $\lambda_1 \neq 0, \lambda_2 \neq 0$ s.t. both $(Z_i, T - \lambda_1 Z_i)$ and $(Z_i, O - \lambda_2 Z_i)$ satisfy the cross-cumulant constraint and $(T - \lambda_1 Z_i, O - \lambda_2 Z_i, Z_i)$ satisfies the independence constraint. Furthermore, if $Z_i \in \mathbf{Z}_b \cup \mathbf{Z}_e$, $(\lambda_1, \lambda_2) = (m_{TZ_i}, m_{OZ_i})$ or $(m_{TL_i}/m_{Z_i L_i}, m_{OL_i}/m_{Z_i L_i})$ iff the above constraints are satisfied.*

*Proof.* "If": We prove this part by contradiction.

- If $Z_i \in \mathbf{Z}_g$, $Z_i$ contains $\epsilon_{L_i}, \epsilon_T, \epsilon_{Z_i}$, and

$$T - \lambda_1 Z_i = (m_{TL_i} - \lambda_1 m_{Z_i L_i}) \epsilon_{L_i} + (1 - \lambda_1 m_{Z_i T}) \epsilon_T - \lambda_1 \epsilon_{Z_i} + ..., \tag{84}$$

$$O - \lambda_2 Z_i = (m_{OL_i} - \lambda_2 m_{Z_i L_i}) \epsilon_{L_i} + (m_{OT} - \lambda_2 m_{Z_i T}) \epsilon_T + (m_{OZ_i} - \lambda_2) \epsilon_{Z_i} + ... \tag{85}$$

Suppose $(T - \lambda_1 Z, O - \lambda_2 Z, Z)$ satisfies the independence constraint, based on Lem. 5,

$$(1 - \lambda_1 m_{Z_i T})(m_{OZ_i} - \lambda_2) = (m_{OT} - \lambda_2 m_{Z_i T})(-\lambda_1), \tag{86}$$

$$(m_{TL_i} - \lambda_1 m_{Z_i L_i})(m_{OZ_i} - \lambda_2) = (m_{OL_i} - \lambda_2 m_{Z_i L_i})(-\lambda_1). \tag{87}$$

From Eqs. (86) and (87), we can derive $\frac{m_{OL_i} - m_{OZ_i} m_{Z_i L_i}}{m_{OT} - m_{OZ_i} m_{Z_i T}} = m_{TL_i}$, note that if $Z_i \in \mathbf{Z}_g$, $m_{OZ_i} m_{Z_i L_i} = m_{OZ_i} m_{Z_i T} m_{TL_i}$ but $m_{OL_i} \neq m_{OT} m_{TL_i}$, this leads to contradiction.

- If $Z_i \in \mathbf{Z}_h$, $Z_i$ contains $\epsilon_O, \epsilon_{Z_i}$, and

$$T - \lambda_1 Z_i = -\lambda_1 m_{Z_i O} \epsilon_O - \lambda_1 \epsilon_{Z_i} + ..., \tag{88}$$

$$O - \lambda_2 Z_i = (1 - \lambda_2 m_{Z_i O}) \epsilon_O - \lambda_2 \epsilon_{Z_i} + ... \tag{89}$$

Suppose $(T - \lambda_1 Z, O - \lambda_2 Z, Z)$ satisfies the independence constraint, based on Lem. 5,

$$(-\lambda_1 m_{Z_i O})(-\lambda_2) = (-\lambda_1)(1 - \lambda_2 m_{Z_i O}), \tag{90}$$

that is, $\lambda_1 = 0$, this leads to contradiction.

"Only if": If $Z_i \in \mathbf{Z}_b \cup \mathbf{Z}_e$, let $(\lambda_1, \lambda_2) = (m_{TZ_i}, m_{OZ_i})$, based on Lem. 2, both $(Z_i, T - \lambda_1 Z_i)$ and $(Z_i, O - \lambda_2 Z_i)$ satisfy the cross-cumulant constraint. Besides,

$$
\begin{aligned}
T - \lambda_1 Z_i =& a_{TL_i} \epsilon_{L_i} \\
& + \sum_{L \in \mathbf{L} \setminus \{L_i\}} m_{TL} \epsilon_L + \sum_{Z \in \mathrm{An}_\mathbf{Z}(T) \setminus \{Z_i\}} m_{TZ} \epsilon_Z + \epsilon_T, 
\end{aligned} \tag{91}
$$

$$
\begin{aligned}
O - \lambda_2 Z_i =& (m_{OL_i} - m_{OZ_i} m_{Z_i L_i}) \epsilon_{L_i} \\
& + \sum_{L \in \mathbf{L} \setminus \{L_i\}} m_{OL} \epsilon_L + \sum_{Z \in \mathrm{An}_\mathbf{Z}(O) \setminus \{Z_i\}} m_{OZ} \epsilon_Z + m_{OT} \epsilon_T + \epsilon_O,
\end{aligned} \tag{92}
$$

$$
Z_i = m_{Z_i L_i} \epsilon_{L_i} + \epsilon_{Z_i}. \tag{93}
$$

Note that $m_{OL_i} - m_{OZ_i} m_{Z_i L_i} \neq 0$, let $\epsilon_{L_i}$ be $e$ in Lem. 3, $(T - \lambda_1 Z_i, O - \lambda_2 Z_i, Z_i)$ satisfies the independence constraint.

Furthermore, based on Lem. 2, $(Z_i, T - \lambda_1 Z_i)$ satisfy the cross-cumulant constraint iff $\lambda_1 = m_{TZ_i}$ or $m_{TL_i}/m_{Z_i L_i}$, $(Z_i, O - \lambda_2 Z_i)$ satisfy the cross-cumulant constraint iff $\lambda_2 = m_{OZ_i}$ or $m_{OL_i}/m_{Z_i L_i}$.

- If $(\lambda_1, \lambda_2) = (m_{TL_i}/m_{Z_i L_i}, m_{OL_i}/m_{Z_i L_i})$,

$$
\begin{aligned}
T - \lambda_1 Z_i =& (m_{TZ_i} - \frac{m_{TL_i}}{m_{Z_i L_i}}) \epsilon_{Z_i} \\
& + \sum_{L \in \mathbf{L} \setminus \{L_i\}} m_{TL} \epsilon_L + \sum_{Z \in \mathrm{An}_\mathbf{Z}(T) \setminus \{Z_i\}} m_{TZ} \epsilon_Z + \epsilon_T,
\end{aligned} \tag{94}
$$

$$
\begin{aligned}
O - \lambda_2 Z_i =& (m_{OZ_i} - \frac{m_{OL_i}}{m_{Z_i L_i}}) \epsilon_{Z_i} \\
& + \sum_{L \in \mathbf{L} \setminus \{L_i\}} m_{OL} \epsilon_L + \sum_{Z \in \mathrm{An}_\mathbf{Z}(O) \setminus \{Z_i\}} m_{OZ} \epsilon_Z + m_{OT} \epsilon_T + \epsilon_O,
\end{aligned} \tag{95}
$$

$$
Z_i = m_{Z_i L_i} \epsilon_{L_i} + \epsilon_{Z_i}. \tag{96}
$$

  Note that $(m_{TZ_i} - \frac{m_{TL_i}}{m_{Z_i L_i}})(m_{OZ_i} - \frac{m_{OL_i}}{m_{Z_i L_i}}) \neq 0$, let $\epsilon_{Z_i}$ be $e$ in Lem. 3, $(T - \lambda_1 Z_i, O - \lambda_2 Z_i, Z_i)$ satisfies the independence constraint.

- If $(\lambda_1, \lambda_2) = (m_{TZ_i}, m_{OL_i}/m_{Z_i L_i})$, based on Eqs. (91) and (95), $T - \lambda_1 Z_i$ contains $\epsilon_{L_i}$ while $O - \lambda_2 Z_i$ does not contain $\epsilon_{L_i}$, note that $m_{Z_i L_i} \neq 0$, $(T - \lambda_1 Z_i, O - \lambda_2 Z_i, Z_i)$ does not satisfy the independence constraint based on Lem. 4.

- If $(\lambda_1, \lambda_2) = (m_{TL_i}/m_{Z_i L_i}, m_{OZ_i})$, based on Eqs. (94) and (92), $T - \lambda_1 Z_i$ does not contain $\epsilon_{L_i}$ while $O - \lambda_2 Z_i$ contains $\epsilon_{L_i}$, note that $m_{Z_i L_i} \neq 0$, so $(T - \lambda_1 Z_i, O - \lambda_2 Z_i, Z_i)$ does not satisfy the independence constraint based on Lem. 4.

$\square$

Thm. 13 establishes the sufficient and necessary condition for $Z_i \in \mathbf{Z}_g$. Furthermore, it gives the value of $(m_{Z_i T}, m_{OZ_i})$ when $Z_i \in \mathbf{Z}_g$.

**Theorem 13.** *Suppose $Z_i \notin \mathbf{Z}_a \cup \mathbf{Z}_b \cup \mathbf{Z}_c \cup \mathbf{Z}_d \cup \mathbf{Z}_e \cup \mathbf{Z}_f$, $Z_i \in \mathbf{Z}_g$ iff there exist $\lambda_1 \neq 0, \lambda_2 \neq 0$ s.t. both $(T, Z_i - \lambda_1 T)$ and $(Z_i - \lambda_1 T, O - \lambda_2(Z_i - \lambda_1 T))$ satisfy the cross-cumulant constraint and $(T, O - \lambda_2(Z_i - \lambda_1 T), Z_i - \lambda_1 T)$ satisfies the independence constraint. Furthermore, $(\lambda_1, \lambda_2) = (m_{Z_i T}, m_{OZ_i})$ iff the above constraints are satisfied.*

*Proof.* "If": We prove this part by contradiction. If $Z_i \in \mathbf{Z}_h$, $m_{TZ_i} = 0$. In particular, both $O - \lambda_2(Z_i - \lambda_1 T)$ and $Z_i - \lambda_1 T$ contain $\epsilon_{Z_i}$ for any $\lambda_1 \neq 0, \lambda_2 \neq 0$. Based on Lem. 4, $(T, O - \lambda_2(Z_i - \lambda_1 T), Z_i - \lambda_1 T)$ does not satisfy the independence constraint.

"Only if": If $Z_i \in \mathbf{Z}_g$, let $(\lambda_1, \lambda_2) = (m_{Z_i T}, m_{OZ_i})$,

$$T = m_{TL_i}\epsilon_{L_i} + \sum_{L \in \mathbf{L} \setminus \{L_i\}} m_{TL}\epsilon_L + \sum_{Z \in \mathrm{An}_{\mathbf{Z}}(T)} m_{TZ}\epsilon_Z + \epsilon_T, \tag{97}$$

$$Z_i - \lambda_1 T = a_{Z_i L_i}\epsilon_{L_i} + \epsilon_{Z_i}, \tag{98}$$

$$O - \lambda_2(Z_i - \lambda_1 T) = (m_{OL_i} - m_{OZ_i}a_{Z_i L_i})\epsilon_{L_i} + \sum_{L \in \mathbf{L} \setminus \{L_i\}} m_{OL}\epsilon_L + \sum_{Z \in \mathrm{An}_{\mathbf{Z}}(O) \setminus \{Z_i\}} m_{OZ}\epsilon_Z$$
$$+ m_{OT}\epsilon_T + \epsilon_O. \tag{99}$$

Note that $m_{OL_i} - m_{OZ_i}m_{Z_i L_i} \neq 0$ and $Z_i \notin \mathrm{An}_{\mathbf{Z}}(T)$, let $\epsilon_{L_i}$ be $\epsilon_{V_1}$ in Lem. 1, both $(T, Z_i - \lambda_1 T)$ and $(T, O - \lambda_2(Z_i - \lambda_1 T), Z_i - \lambda_1 T)$ satisfy the cross-cumulant constraint. Let $\epsilon_{L_i}$ be $e$ in Lem. 3, $(T, O - \lambda_2(Z_i - \lambda_1 T), Z_i - \lambda_1 T)$ satisfies the independence constraint.

Furthermore,

- If $\lambda_2 \neq m_{OZ_i}$, $O - \lambda_2(Z_i - \lambda_1 T)$ contains $\epsilon_{Z_i}$ for any $\lambda_1 \neq 0$. Note that $m_{TZ_i} = 0$ and $Z_i - \lambda_1 T$ contains $\epsilon_{Z_i}$ for any $\lambda_1 \neq 0$, $(T, O - \lambda_2(Z_i - \lambda_1 T), Z_i - \lambda_1 T)$ does not satisfy the independence constraint based on Lem. 4.

- If $\lambda_2 = m_{OZ_i}$ and $\lambda_1 \neq m_{Z_i T}$, because $(T, Z_i - \lambda_1 T)$ satisfies the cross-cumulant constraint, we have $\lambda_1 = \frac{m_{Z_i L_i}}{m_{TL_i}}$ based on Lem. 2. Let $L_j \in \mathbf{L} \setminus \{L_i\}$, $Z_i - \lambda_1 T$ contains $\epsilon_T, \epsilon_{L_j}$, and

$$T = \epsilon_T + m_{TL_j}\epsilon_{L_j} + ..., \tag{100}$$

$$O - \lambda_2(Z_i - \lambda_1 T) = (m_{OT} - m_{OZ_i}m_{Z_i T} + \frac{m_{OZ_i}m_{Z_i L_i}}{m_{TL_i}})\epsilon_T$$
$$+ (m_{OL_j} - m_{OZ_i}m_{Z_i L_j} + \frac{m_{OZ_i}m_{Z_i L_i}m_{TL_j}}{m_{TL_i}})\epsilon_{L_j} + ... \tag{101}$$

Note that $m_{OL_j} \neq m_{OT}m_{TL_j}$ and $m_{Z_i L_j} = m_{Z_i T}m_{TL_j}$, we have

$$m_{TL_j}(m_{OT} - m_{OZ_i}m_{Z_i T} + \frac{m_{OZ_i}m_{Z_i L_i}}{m_{TL_i}}) \neq m_{OL_j} - m_{OZ_i}m_{Z_i L_j} + \frac{m_{OZ_i}m_{Z_i L_i}m_{TL_j}}{m_{TL_i}}, \tag{102}$$

so $(T, O - \lambda_2(Z_i - \lambda_1 T), Z_i - \lambda_1 T)$ does not satisfy the independence constraint based on Lem. 5.

□

Thm. 14 gives the value of $(a_{Z_i T}, a_{Z_i O})$ when $Z_i \in \mathbf{Z}_h$.

**Theorem 14.** *If $Z_i \in \mathbf{Z}_h$, $(\lambda_1, \lambda_2) = (a_{Z_i T}, a_{Z_i O})$ iff both $(T, Z_i - \lambda_1 T - \lambda_2 O)$ and $(O, Z_i - \lambda_1 T - \lambda_2 O)$ satisfy the cross-cumulant constraint and $(T, O, Z_i - \lambda_1 T - \lambda_2 O)$ satisfies the independence constraint.*

*Proof.* "Only if": If $(\lambda_1, \lambda_2) = (a_{Z_i T}, a_{Z_i O})$,

$$T = m_{TL_i}\epsilon_{L_i} + \sum_{L \in \mathbf{L} \setminus \{L_i\}} m_{TL}\epsilon_L + \sum_{Z \in \mathrm{An}_{\mathbf{Z}}(T)} m_{TZ}\epsilon_Z + \epsilon_T, \tag{103}$$

$$O = m_{OL_i}\epsilon_{L_i} + \sum_{L \in \mathbf{L} \setminus \{L_i\}} m_{OL}\epsilon_L + \sum_{Z \in \mathrm{An}_{\mathbf{Z}}(O)} m_{OZ}\epsilon_Z + m_{OT}\epsilon_T + \epsilon_O, \tag{104}$$

$$Z_i - \lambda_1 T - \lambda_2 O = a_{Z_i L_i}\epsilon_{L_i} + \epsilon_{Z_i}. \tag{105}$$

Note that $Z_i \notin \mathrm{An}_{\mathbf{Z}}(T) \cup \mathrm{An}_{\mathbf{Z}}(O)$, let $\epsilon_{L_i}$ be $\epsilon_{V_1}$ in Lem. 1, both $(T, Z_i - \lambda_1 T - \lambda_2 O)$ and $(O, Z_i - \lambda_1 T - \lambda_2 O)$ satisfy the cross-cumulant constraint. Let $\epsilon_{L_i}$ be $e$ in Lem. 3, $(T, O, Z_i - \lambda_1 T - \lambda_2 O)$ satisfies the independence constraint.

"If": We prove this part by contradiction. If $\lambda_2 \neq a_{Z_i O}$, $Z_i - \lambda_1 T - \lambda_2 O$ contains $\epsilon_O$. Note that $m_{OO} \neq 0$ and $m_{TO} = 0$, $(T, O, Z_i - \lambda_1 T - \lambda_2 O)$ does not satisfy the independence constraint based on Lem. 4. Moreover, if $\lambda_2 = a_{Z_i O}$ and $\lambda_1 \neq a_{Z_i T}$, similar to the proof of Thm. 11, we can prove $(T, O, Z_i - \lambda_1 T - \lambda_2 O)$ does not satisfy the independence constraint. Therefore, $(\lambda_1, \lambda_2)$ has a unique value $(a_{Z_i T}, a_{Z_i O})$. □

---

**Algorithm 2:** Identification procedure (multiple latent confounders).

**Input:** The joint distribution of $(T, O, \mathbf{Z})$

**Output:** $m_{OT}$

1   $\mathbf{Z}_a, \mathbf{Z}_{be}, ..., \mathbf{Z}_h := \emptyset, \emptyset, ..., \emptyset$.

2   **for** $Z_i \in \mathbf{Z}$ **do**

3      **if** *the constraint in Thm. 8 is satisfied* **then**

4        $\mathbf{Z}_a := \mathbf{Z}_a \cup \{Z_i\}$.

5      **else if** *there exists $\lambda \neq 0$ s.t. the constraints in Thm. 9 are satisfied* **then**

6        $\mathbf{Z}_c := \mathbf{Z}_c \cup \{(Z_i, \lambda)\}$.

7      **else if** *there exists $\lambda \neq 0$ s.t. the constraints in Thm. 10 are satisfied* **then**

8        $\mathbf{Z}_d := \mathbf{Z}_d \cup \{(Z_i, \lambda)\}$.

9      **else if** *there exists $\lambda \neq 0$ s.t. the constraints in Thm. 11 are satisfied* **then**

10        $\mathbf{Z}_f := \mathbf{Z}_f \cup \{(Z_i, \lambda)\}$.

11      **else if** *there exist $\lambda_1 \neq 0, \lambda_2 \neq 0$ s.t. the constraints in Thm. 12 are satisfied* **then**

12        $\mathbf{Z}_{be} := \mathbf{Z}_{be} \cup \{(Z_i, \lambda_1, \lambda_2)\}$.

13      **else if** *there exist $\lambda_1 \neq 0, \lambda_2 \neq 0$ s.t. the constraints in Thm. 13 are satisfied* **then**

14        $\mathbf{Z}_g := \mathbf{Z}_g \cup \{(Z_i, \lambda_1, \lambda_2)\}$.

15      **else**

16        $\mathbf{Z}_h := \mathbf{Z}_h \cup \{(Z_i, \lambda_1, \lambda_2)\}$ where $\lambda_1, \lambda_2$ satisfy Thm. 14.

17   **for** $(Z_i, \lambda) \in \mathbf{Z}_d$ **do**

18      $Z_i := Z_i - \lambda O$.

19   **for** $(Z_i, \lambda) \in \mathbf{Z}_f$ **do**

20      $Z_i := Z_i - \lambda T$.

21   **for** $(Z_i, \lambda_1, \lambda_2) \in \mathbf{Z}_h$ **do**

22      $Z_i := Z_i - \lambda_1 T - \lambda_2 O$.

23   **for** $(Z_i, \lambda_1, \lambda_2) \in \mathbf{Z}_g$ **do**

24      $O, Z_i := O - \lambda_2(Z_i - \lambda_1 T), Z_i - \lambda_1 T$.

25   **for** $(Z_i, \lambda_1, \lambda_2) \in \mathbf{Z}_{be}$ **do**

26      $T, O := T - \lambda_1 Z_i, O - \lambda_2 Z_i$.

27   **for** $(Z_i, \lambda) \in \mathbf{Z}_c$ **do**

28      $O := O - \lambda Z_i$.

29   **return** $f(T, O, \mathbf{Z})$

---

## C.2   IDENTIFICATION PROCEDURE

Let

$$f(T, O, \mathbf{Z}) := \frac{\text{Cov}(T, O) - \sum_{Z_i \in \mathbf{Z}} g(T, Z_i)\text{Cov}(O, Z_i)}{\text{Var}(T) - \sum_{Z_i \in \mathbf{Z}} g(T, Z_i)\text{Cov}(T, Z_i)}, \tag{106}$$

our identification procedure is summarized in Alg. 2.

**Theorem 15.** *If Asmps. 1 and 2 hold, Alg. 2 correctly identifies the true causal effect.*

*Proof.* Denote the updated $T, O, Z_i$ by $T', O', Z_i'$, according to Eqs. (72)-(74), (75)-(77), (78)-(80), (81)-(83), (91)-(93), (94)-(96), (97)-(99), and (103)-(105), $Z_i' \perp\!\!\!\perp (T', O')|L_i$ for each $Z_i'$ and $m_{OT} = m_{O'T'}$, so $m_{OT} = f(T', O', \mathbf{Z}')$ based on Thm. 2 in Kivva et al. (2023). $\qquad\square$

**Remark 4.** While $m_{OT}$ may be unidentifiable with a single latent confounder, it is always identifiable with multiple latent confounders.

## C.3   EXPERIMENTS

We construct a plug-in estimation method by substituting sample-level cross-cumulants and independence tests into Alg. 2, and then conduct numerical experiments.

First, with the type of each proxy not known a priori, we check whether our estimation method can correctly determine it. Specifically, we synthesize datasets according to the causal structures shown in Fig. 9. Clearly, in Fig. 9(a), $Z \in \mathbf{Z}_a$; in Fig. 9(b), $Z \in \mathbf{Z}_b$; ...; in Fig. 9(h), $Z \in \mathbf{Z}_h$. Experimental results are summarized in Fig. 10. Since $\mathbf{Z}_b$ and $\mathbf{Z}_e$ are grouped into the same class (i.e., $\mathbf{Z}_{be}$), we

use only $\mathbf{Z}_b$ as their representative. Obviously, the errors gradually approach zero as sample size increases.

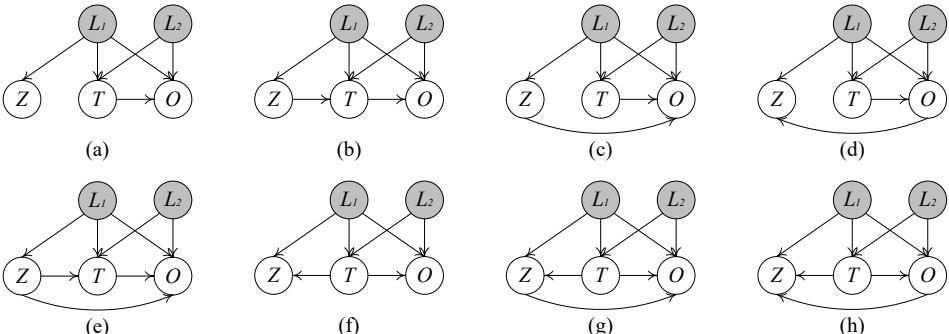

(a)  (b)  (c)  (d)

(e)  (f)  (g)  (h)

Figure 9: Causal structures used to evaluate the ability of the estimation method based on Alg. 2 to determine the type of the proxy.

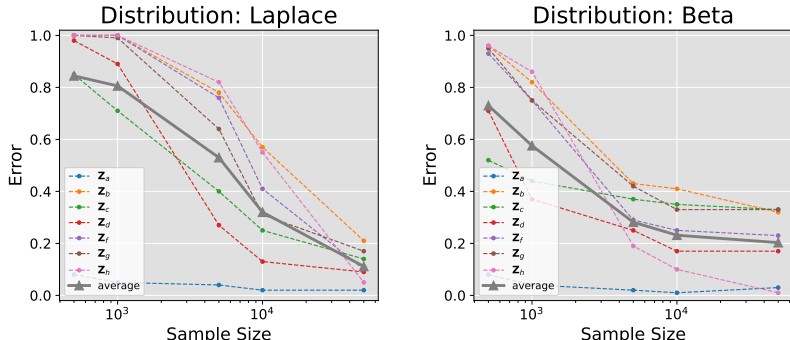

Figure 10: Error in determining the type of the proxy w.r.t. sample size (multiple latent confounders).

Second, with the type of each proxy known a priori, we check whether our estimation method can precisely estimate the causal effect. We synthesize datasets according to the causal structures shown in Fig. 11. The experimental results are summarized in Fig. 12. As sample size increases, the errors of our estimation method gradually approach zero.

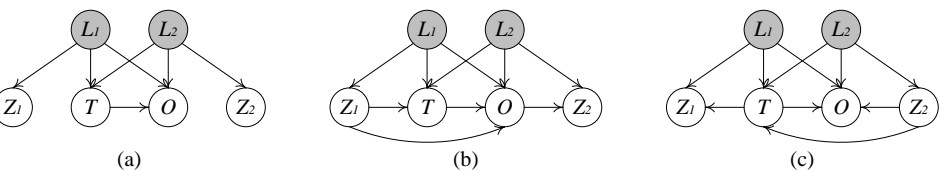

(a)  (b)  (c)

Figure 11: Causal structures used to evaluate the ability of the estimation method based on Alg. 2 to estimate the causal effect.

# D  EXTENSION 2: GENERALIZED AGNOSTIC PROXY

## D.1  THEORETICAL RESULTS

Thms. 16 and 17 demonstrate that if the underlying causal structure is one of Fig. 5(a)-(l) and it is unknown which specific one the underlying causal structure is, $m_{OT}$ is unidentifiable.

**Theorem 16.** *Given two SCMs $\mathcal{M}$ and $\tilde{\mathcal{M}}$ where the causal structure of $\mathcal{M}$ is shown as Fig. 5(i) and that of $\tilde{\mathcal{M}}$ is shown as Fig. 5(b), these two SCMs can entail the same observational distribution but $m_{OT} \neq m_{\tilde{O}\tilde{T}}$.*

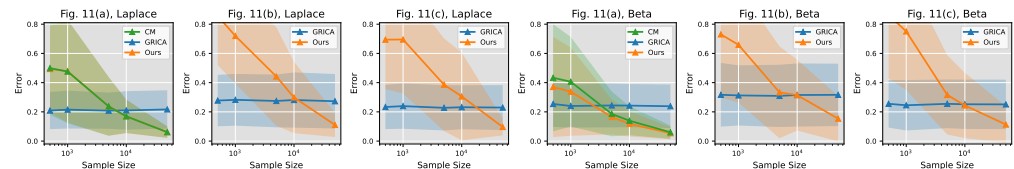

Figure 12: Error in estimating the causal effect w.r.t. sample size (multiple latent confounders).

*Proof.* Let

$$\mathcal{M} := \begin{cases} Z = \epsilon_Z, \\ L = a_{LZ}Z + \epsilon_L, \\ T = a_{TL}L + \epsilon_T, \\ O = a_{OT}T + a_{OL}L + \epsilon_O, \end{cases} \qquad , \qquad \tilde{\mathcal{M}} := \begin{cases} \tilde{Z} = \epsilon_Z, \\ \tilde{L} = \epsilon_T, \\ \tilde{T} = a_{TL}a_{LZ}\tilde{Z} + \tilde{L} + a_{TL}\epsilon_L, \\ \tilde{O} = \frac{a_{OT}a_{TL}+a_{OL}}{a_{TL}}\tilde{T} - \frac{a_{OL}}{a_{TL}}\tilde{L} + \epsilon_O, \end{cases} \qquad . \qquad (107)$$

$T = \tilde{T} = a_{TL}a_{LZ}\epsilon_Z + a_{TL}\epsilon_L + \epsilon_T$, $O = \tilde{O} = (a_{OT}a_{TL}a_{LZ} + a_{OL}a_{LZ})\epsilon_Z + (a_{OT}a_{TL} + a_{OL})\epsilon_L + a_{OT}\epsilon_T + \epsilon_O$, $Z = \tilde{Z} = \epsilon_Z$, that is, $\mathcal{M}$ and $\tilde{\mathcal{M}}$ entail the same observational distribution. However, $m_{OT} = a_{OT}$ while $m_{\tilde{O}\tilde{T}} = \frac{a_{OT}a_{TL}+a_{OL}}{a_{TL}}$. $\qquad \square$

**Theorem 17.** *In Fig. 5(a), (c)-(h), and (j-l), $m_{OT}$ is unidentifiable.*

*Proof.* We construct two SCMs $\mathcal{M}$ and $\tilde{\mathcal{M}}$ that share the same causal structure shown as Fig. 5(a):

$$\mathcal{M} := \begin{cases} L = \epsilon_L, \\ Z = \epsilon_Z, \\ T = a_{TL}L + \epsilon_T, \\ O = a_{OT}T + a_{OL}L + \epsilon_O, \end{cases} \qquad , \qquad \tilde{\mathcal{M}} := \begin{cases} \tilde{L} = \epsilon_T, \\ \tilde{Z} = \epsilon_Z, \\ \tilde{T} = \tilde{L} + a_{TL}\epsilon_L, \\ \tilde{O} = \frac{a_{OT}a_{TL}+a_{OL}}{a_{TL}}\tilde{T} - \frac{a_{OL}}{a_{TL}}\tilde{L} + \epsilon_O, \end{cases} \qquad . \qquad (108)$$

$T = \tilde{T} = a_{TL}\epsilon_L + \epsilon_T$, $O = \tilde{O} = (a_{OT}a_{TL} + a_{OL})\epsilon_L + a_{OT}\epsilon_T + \epsilon_O$, $Z = \tilde{Z} = \epsilon_Z$, that is, $\mathcal{M}$ and $\tilde{\mathcal{M}}$ entail the same observational distribution. However, $m_{OT} = a_{OT}$ while $m_{\tilde{O}\tilde{T}} = \frac{a_{OT}a_{TL}+a_{OL}}{a_{TL}}$. Analogously, we can also construct such two SCMs for Fig. 5(c)-(d) and (f)-(h).

- For Fig. 5(c), we add $a_{OZ}Z$ to $O$, add $a_{OZ}\tilde{Z}$ to $\tilde{O}$, and keep all other assignments fixed in Eq. (108).

- For Fig. 5(d), we add $a_{ZO}O$ to $Z$, add $a_{ZO}\tilde{O}$ to $\tilde{Z}$, and keep all other assignments fixed in Eq. (108).

- For Fig. 5(f), we add $a_{ZT}T$ to $Z$, add $a_{ZT}\tilde{T}$ to $\tilde{Z}$, and keep all other assignments fixed in Eq. (108).

- For Fig. 5(g), we add $a_{ZT}T, a_{OZ}Z$ to $Z, O$ respectively, add $a_{ZT}\tilde{T}, a_{OZ}\tilde{Z}$ to $\tilde{Z}, \tilde{O}$ respectively, and keep all other assignments fixed in Eq. (108).

- For Fig. 5(h), we add $a_{ZT}T + a_{ZO}$ to $Z$, add $a_{ZT}\tilde{T} + a_{ZO}\tilde{O}$ to $\tilde{Z}$, and keep all other assignments fixed in Eq. (108).

We construct two SCMs $\mathcal{M}$ and $\tilde{\mathcal{M}}$ that share the same causal structure shown as Fig. 5(e):

$$\mathcal{M} := \begin{cases} L = \epsilon_L, \\ Z = \epsilon_Z, \\ T = a_{TZ}Z + a_{TL}L + \epsilon_T, \\ O = a_{OZ}Z + a_{OT}T + a_{OL}L + \epsilon_O, \end{cases} \qquad , \qquad (109)$$

$$\tilde{\mathcal{M}} := \begin{cases} \tilde{L} = \epsilon_T, \\ \tilde{Z} = \epsilon_Z, \\ \tilde{T} = a_{TZ}\tilde{Z} + \tilde{L} + a_{TL}\epsilon_L, \\ \tilde{O} = \frac{a_{OZ}a_{TL}-a_{TZ}a_{OL}}{a_{TL}}\tilde{Z} + \frac{a_{OT}a_{TL}+a_{OL}}{a_{TL}}\tilde{T} - \frac{a_{OL}}{a_{TL}}\tilde{L} + \epsilon_O, \end{cases} \qquad . \qquad (110)$$

$T = \tilde{T} = a_{TL}\epsilon_L + a_{TZ}\epsilon_Z + \epsilon_T$, $O = \tilde{O} = (a_{OT}a_{TL} + a_{OL})\epsilon_L + (a_{OT}a_{TZ} + a_{OZ})\epsilon_Z + a_{OT}\epsilon_T + \epsilon_O$, $Z = \tilde{Z} = \epsilon_Z$, that is, $\mathcal{M}$ and $\tilde{\mathcal{M}}$ entail the same observational distribution. However, $m_{OT} = a_{OT}$ while $m_{\tilde{O}\tilde{T}} = \frac{a_{OT}a_{TL}+a_{OL}}{a_{TL}}$.

We construct two SCMs $\mathcal{M}$ and $\tilde{\mathcal{M}}$ that share the same causal structure shown as Fig. 5(j):

$$\mathcal{M} := \begin{cases} Z = \epsilon_Z, \\ L = a_{LZ}Z + \epsilon_L, \\ T = a_{TZ}Z + a_{TL}L + \epsilon_T, \\ O = a_{OT}T + a_{OL}L + \epsilon_O, \end{cases} \quad, \quad \tilde{\mathcal{M}} := \begin{cases} \tilde{Z} = \epsilon_Z, \\ \tilde{L} = a_{TZ}\tilde{Z} + \epsilon_T, \\ \tilde{T} = a_{TL}a_{LZ}\tilde{Z} + \tilde{L} + a_{TL}\epsilon_L, \\ \tilde{O} = \frac{a_{OT}a_{TL}+a_{OL}}{a_{TL}}\tilde{T} - \frac{a_{OL}}{a_{TL}}\tilde{L} + \epsilon_O, \end{cases} \quad . \tag{111}$$

$T = \tilde{T} = (a_{TL}a_{LZ} + a_{TZ})\epsilon_Z + a_{TL}\epsilon_L + \epsilon_T$, $O = \tilde{O} = (a_{OT}a_{TL}a_{LZ} + a_{OT}a_{TZ} + a_{OL}a_{LZ})\epsilon_Z + (a_{OT}a_{TL} + a_{OL})\epsilon_L + a_{OT}\epsilon_T + \epsilon_O$, $Z = \tilde{Z} = \epsilon_Z$, that is, $\mathcal{M}$ and $\tilde{\mathcal{M}}$ entail the same observational distribution. However, $m_{OT} = a_{OT}$ while $m_{\tilde{O}\tilde{T}} = \frac{a_{OT}a_{TL}+a_{OL}}{a_{TL}}$. Analogously, we can also construct such two SCMs for Fig. 5(l). Specifically, we add $a_{OZ}Z$ to $O$, add $a_{OZ}\tilde{Z}$ to $\tilde{O}$, and keep all other assignments fixed in Eq. (111).

We construct two SCMs $\mathcal{M}$ and $\tilde{\mathcal{M}}$ that share the same causal structure shown as Fig. 5(k):

$$\mathcal{M} := \begin{cases} Z = \epsilon_Z, \\ L = a_{LZ}Z + \epsilon_L, \\ T = a_{TL}L + \epsilon_T, \\ O = a_{OZ}Z + a_{OT}T + a_{OL}L + \epsilon_O, \end{cases} \quad, \tag{112}$$

$$\tilde{\mathcal{M}} := \begin{cases} \tilde{Z} = \epsilon_Z, \\ \tilde{L} = a_{TL}a_{LZ}\tilde{Z} + \epsilon_T, \\ \tilde{T} = \tilde{L} + a_{TL}\epsilon_L, \\ \tilde{O} = \frac{a_{OT}a_{TL}+a_{OL}}{a_{TL}}\tilde{T} - \frac{a_{OL}}{a_{TL}}\tilde{L} + (a_{OL}a_{LZ} + a_{OZ})\tilde{Z} + \epsilon_O, \end{cases} \quad . \tag{113}$$

$T = \tilde{T} = a_{TL}a_{LZ}\epsilon_Z + a_{TL}\epsilon_L + \epsilon_T$, $O = \tilde{O} = (a_{OT}a_{TL}a_{LZ} + a_{OL}a_{LZ} + a_{OZ})\epsilon_Z + (a_{OT}a_{TL} + a_{OL})\epsilon_L + a_{OT}\epsilon_T + \epsilon_O$, $Z = \tilde{Z} = \epsilon_Z$, that is, $\mathcal{M}$ and $\tilde{\mathcal{M}}$ entail the same observational distribution. However, $m_{OT} = a_{OT}$ while $m_{\tilde{O}\tilde{T}} = \frac{a_{OT}a_{TL}+a_{OL}}{a_{TL}}$. $\quad\square$

Thms. 18-21 collectively establish the sufficient and necessary condition for the underlying causal structure to be one of Fig. 5(a)-(l).

**Theorem 18.** *The ground truth is one of Fig. 5(a)-(c), (e), and (i)-(l) iff* $T - \frac{\text{Cov}(T,Z)}{\text{Var}(Z)}Z \perp\!\!\!\perp Z$ *and* $O - \frac{\text{Cov}(O,Z)}{\text{Var}(Z)}Z \perp\!\!\!\perp Z$.

*Proof.* "Only if": In Fig. 5(a), $T - \frac{\text{Cov}(T,Z)}{\text{Var}(Z)}Z = T \perp\!\!\!\perp Z$ and $O - \frac{\text{Cov}(O,Z)}{\text{Var}(Z)}Z = O \perp\!\!\!\perp Z$.

In Fig. 5(b)-(c), (e), and (i)-(l), $Z, T, O$ can be expressed as

$$Z = \epsilon_Z, \quad T = m_{TZ}\epsilon_Z + e_T, \quad O = m_{OZ}\epsilon_Z + e_O, \tag{114}$$

where $e_T \perp\!\!\!\perp \epsilon_Z$ and $e_O \perp\!\!\!\perp \epsilon_Z$. So $T - \frac{\text{Cov}(T,Z)}{\text{Var}(Z)}Z = e_T \perp\!\!\!\perp Z$ and $O - \frac{\text{Cov}(O,Z)}{\text{Var}(Z)}Z = e_O \perp\!\!\!\perp Z$.

"If": We prove this part by contradiction.

- In Fig. 5(d), (f)-(h), (m), (o)-(p), and (r)-(t), $m_{TZ} = 0$, $m_{ZZ} \neq 0$, and $\text{Cov}(T, Z) \neq 0$, so both $T - \frac{\text{Cov}(T,Z)}{\text{Var}(Z)}Z$ and $Z$ contains $\epsilon_Z$, thus $T - \frac{\text{Cov}(T,Z)}{\text{Var}(Z)}Z \not\perp\!\!\!\perp Z$.

- In Fig. 5(n) and (q), $\{L, Z\} \subseteq \overline{\text{An}}(Z) \cap \overline{\text{An}}(T)$ and there exist two non-intersecting paths from $\{L, Z\}$ to $\{Z, T\}$ (e.g., $Z$ and $L \to T$), so $m_{ZL}/m_{ZZ} \neq m_{TL}/m_{TZ}$, that is, $T - \frac{\text{Cov}(T,Z)}{\text{Var}(Z)}Z$ contains at least one of $\epsilon_L$ and $\epsilon_Z$. Note that $Z$ contains both $\epsilon_L$ and $\epsilon_Z$, so $T - \frac{\text{Cov}(T,Z)}{\text{Var}(Z)}Z \not\perp\!\!\!\perp Z$.

$\square$

**Theorem 19.** *Suppose the ground truth is not one of Fig. 5(a)-(c), (e), and (i)-(l), it is Fig. 5(f) or (g) iff* $Z - \frac{\text{Cov}(Z,T)}{\text{Var}(T)}T \perp\!\!\!\perp T$.

*Proof.* "Only if": In Fig. 5(f) and (g), $Z = a_{ZT}T + \epsilon_Z$ where $T \perp\!\!\!\perp \epsilon_Z$, $Z - \frac{\text{Cov}(Z,T)}{\text{Var}(T)}T = \epsilon_Z \perp\!\!\!\perp T$.

"If": We prove this part by contradiction.

- In Fig. 5(d), (h), (p), and (r)-(t), $\{L, T\} \subseteq \overline{\text{An}}(Z) \cap \overline{\text{An}}(T)$ and there exist two non-intersecting paths from $\{L, T\}$ to $\{Z, T\}$ (e.g., $T$ and $L \to O \to Z$ in Fig. 5(d) and (h), $T$ and $L \to Z$ in Fig. 5(p) and (r)-(t)), so $m_{ZL}/m_{ZT} \neq m_{TL}/m_{TT}$, that is, $Z - \frac{\text{Cov}(Z,T)}{\text{Var}(T)}T$ contains at least one of $\epsilon_L$ and $\epsilon_T$. Note that $T$ contains both $\epsilon_L$ and $\epsilon_T$, so $Z - \frac{\text{Cov}(Z,T)}{\text{Var}(T)}T \not\perp\!\!\!\perp T$.

- In Fig. 5(m)-(o) and (q), $m_{ZT} = 0$, $m_{TT} \neq 0$, and $\text{Cov}(Z, T) \neq 0$, so both $Z - \frac{\text{Cov}(Z,T)}{\text{Var}(T)}T$ and $T$ contains $\epsilon_T$, thus $Z - \frac{\text{Cov}(Z,T)}{\text{Var}(T)}T \not\perp\!\!\!\perp T$.

$\square$

**Theorem 20.** *Suppose the ground truth is not one of Fig. 5(a)-(c), (e)-(g), and (i)-(l), it is Fig. 5(d) iff $Z - \frac{\text{Cov}(Z,O)}{\text{Var}(O)}O \perp\!\!\!\perp O$.*

*Proof.* "Only if": In Fig. 5(d), $Z = a_{ZO}O + \epsilon_Z$ where $O \perp\!\!\!\perp \epsilon_Z$, $Z - \frac{\text{Cov}(Z,O)}{\text{Var}(O)}O = \epsilon_Z \perp\!\!\!\perp O$.

"If": We prove this part by contradiction.

- In Fig. 5(h), (p), and (t), $\{L, O\} \subseteq \overline{\text{An}}(Z) \cap \overline{\text{An}}(O)$ and there exist two non-intersecting paths from $\{L, O\}$ to $\{Z, O\}$ (e.g., $O$ and $L \to T \to Z$ in Fig. 5(h), $O$ and $L \to Z$ in Fig. 5(p) and (t)), so $m_{ZL}/m_{ZO} \neq m_{OL}/m_{OO}$, that is, $Z - \frac{\text{Cov}(Z,O)}{\text{Var}(O)}O$ contains at least one of $\epsilon_L$ and $\epsilon_O$. Note that $O$ contains both $\epsilon_L$ and $\epsilon_O$, so $Z - \frac{\text{Cov}(Z,O)}{\text{Var}(O)}O \not\perp\!\!\!\perp O$.

- In Fig. 5(m)-(o) and (q)-(s), $m_{ZO} = 0$, $m_{OO} \neq 0$, and $\text{Cov}(Z, O) \neq 0$, so both $Z - \frac{\text{Cov}(Z,O)}{\text{Var}(O)}O$ and $O$ contains $\epsilon_O$, thus $Z - \frac{\text{Cov}(Z,O)}{\text{Var}(O)}O \not\perp\!\!\!\perp O$.

$\square$

**Theorem 21.** *Suppose the ground truth is not one of Fig. 5(a)-(g) and (i)-(l), it is Fig. 5(h) iff*

$$Z - \begin{bmatrix} T & O \end{bmatrix} \begin{bmatrix} \text{Var}(T) & \text{Cov}(T,O) \\ \text{Cov}(T,O) & \text{Var}(O) \end{bmatrix}^{-1} \begin{bmatrix} \text{Cov}(T,Z) \\ \text{Cov}(O,Z) \end{bmatrix} \perp\!\!\!\perp \{T, O\}$$

*Proof.* "Only if": In Fig. 5(h), $Z = a_{ZT}T + z_{ZO}O + \epsilon_Z$, there is

$$\begin{bmatrix} \text{Cov}(T,Z) \\ \text{Cov}(O,Z) \end{bmatrix} = \begin{bmatrix} \text{Var}(T) & \text{Cov}(T,O) \\ \text{Cov}(T,O) & \text{Var}(O) \end{bmatrix} \begin{bmatrix} a_{ZT} \\ a_{ZO} \end{bmatrix}, \tag{115}$$

so

$$Z - \begin{bmatrix} T & O \end{bmatrix} \begin{bmatrix} \text{Var}(T) & \text{Cov}(T,O) \\ \text{Cov}(T,O) & \text{Var}(O) \end{bmatrix}^{-1} \begin{bmatrix} \text{Cov}(T,Z) \\ \text{Cov}(O,Z) \end{bmatrix} = \epsilon_Z \perp\!\!\!\perp \{T, O\}. \tag{116}$$

"If": We prove this part by contradiction. Specifically, we prove for any $\lambda_1, \lambda_2$, $Z - \lambda_1 T - \lambda_2 O \not\perp\!\!\!\perp \{T, O\}$.

- In Fig. 5(m)-(o) and (q), if $\lambda_2 \neq 0$ or $\lambda_1 \neq \frac{m_{ZL}}{m_{TL}}$, $Z - \lambda_1 T - \lambda_2 O$ contains $\epsilon_O$ or $\epsilon_L$, so $Z - \lambda_1 T - \lambda_2 O \not\perp\!\!\!\perp \{T, O\}$. If $\lambda_2 = 0$ and $\lambda_1 = \frac{m_{ZL}}{m_{TL}} \neq 0$, note that $m_{ZT} = 0$ and $m_{TT} \neq 0$, $Z - \lambda_1 T - \lambda_2 O$ contains $\epsilon_T$, so $Z - \lambda_1 T - \lambda_2 O \not\perp\!\!\!\perp T$.

- In Fig. 5(r) and (s), if $\lambda_2 \neq 0$ or $\lambda_1 \neq \frac{m_{ZL}}{m_{TL}}$, $Z - \lambda_1 T - \lambda_2 O$ contains $\epsilon_O$ or $\epsilon_L$, so $Z - \lambda_1 T - \lambda_2 O \not\perp\!\!\!\perp O$. If $\lambda_2 = 0$ and $\lambda_1 = \frac{m_{ZL}}{m_{TL}}$, note that

$$Z = (a_{ZT}a_{TL} + a_{ZL})\epsilon_L + a_{ZT}\epsilon_T + \epsilon_Z, \tag{117}$$

$$T = a_{TL}\epsilon_L + \epsilon_T, \tag{118}$$

then $Z - \lambda_1 T - \lambda_2 O = -\frac{a_{ZL}}{a_{TL}}\epsilon_T + \epsilon_Z$ contains $\epsilon_T$, so $Z - \lambda_1 T - \lambda_2 O \not\perp\!\!\!\perp T$.

- In Fig. 5(p), if $\lambda_2 \neq m_{ZO}$ or $\lambda_1 \neq \frac{m_{ZL} - m_{ZO}m_{OL}}{m_{TL}}$, $Z - \lambda_1 T - \lambda_2 O$ contains $\epsilon_O$ or $\epsilon_L$, so $Z - \lambda_1 T - \lambda_2 O \not\perp\!\!\!\perp O$. If $\lambda_2 = m_{ZO}$ and $\lambda_1 = \frac{m_{ZL} - m_{ZO}m_{OL}}{m_{TL}}$, note that

$$T = a_{TL}\epsilon_L + \epsilon_T, \tag{119}$$

$$O = (a_{OL} + a_{OT}a_{TL})\epsilon_L + a_{OT}\epsilon_T + \epsilon_O, \tag{120}$$

$$Z = (a_{ZL} + a_{ZO}a_{OL} + a_{ZO}a_{OT}a_{TL})\epsilon_L + a_{ZO}a_{OT}\epsilon_T + a_{ZO}\epsilon_O + \epsilon_Z, \tag{121}$$

---

**Algorithm 3:** Identification procedure (generalized agnostic proxy).

**Input:** The joint distribution of $(T, O, Z)$

**Output:** $m_{OT}$

1 **if** *the condition in Thm. 18 is satisfied* **then**
2     **raise** Error("$m_{OT}$ is unidentifiable.")
3 **else if** *the condition in Thm. 19 is satisfied* **then**
4     **raise** Error("$m_{OT}$ is unidentifiable.")
5 **else if** *the condition in Thm. 20 is satisfied* **then**
6     **raise** Error("$m_{OT}$ is unidentifiable.")
7 **else if** *the condition in Thm. 21 is satisfied* **then**
8     **raise** Error("$m_{OT}$ is unidentifiable.")
9 **else**
10     Run Alg. 1

---

then $Z - \lambda_1 T - \lambda_2 O = \epsilon_Z - \frac{a_{ZL}}{a_{TL}} \epsilon_T$ contains $\epsilon_T$, so $Z - \lambda_1 T - \lambda_2 O \not\perp\!\!\!\perp T$.

- In Fig. 5(t), if $\lambda_2 \neq m_{ZO}$ or $\lambda_1 \neq \frac{m_{ZL} - m_{ZO} m_{OL}}{m_{TL}}$, $Z - \lambda_1 T - \lambda_2 O$ contains $\epsilon_O$ or $\epsilon_L$, so $Z - \lambda_1 T - \lambda_2 O \not\perp\!\!\!\perp O$. If $\lambda_2 = m_{ZO}$ and $\lambda_1 = \frac{m_{ZL} - m_{ZO} m_{OL}}{m_{TL}}$, note that

$$T = a_{TL}\epsilon_L + \epsilon_T, \tag{122}$$

$$O = (a_{OL} + a_{OT} a_{TL})\epsilon_L + a_{OT}\epsilon_T + \epsilon_O, \tag{123}$$

$$Z = (a_{ZL} + a_{ZO} a_{OL} + a_{ZT} a_{TL} + a_{ZO} a_{OT} a_{TL})\epsilon_L + (a_{ZO} a_{OT} + a_{ZT})\epsilon_T + a_{ZO}\epsilon_O + \epsilon_Z, \tag{124}$$

then $Z - \lambda_1 T - \lambda_2 O = \epsilon_Z - \frac{a_{ZL}}{a_{TL}} \epsilon_T$ contains $\epsilon_T$, so $Z - \lambda_1 T - \lambda_2 O \not\perp\!\!\!\perp T$.

$\square$

### D.2 IDENTIFICATION PROCEDURE

Our identification procedure is summarized in Alg. 3.

**Theorem 22.** *If Asmps. 1 and 2 hold, Alg. 3 correctly identifies the true causal effect when identifiable, and correctly reports unidentifiability otherwise.*

### D.3 EXPERIMENTS

We construct a plug-in estimation method by substituting sample-level cross-cumulants and independence tests into Alg. 3, and then conduct numerical experiments.

With the underlying causal structure not known a priori, we check whether our estimation method can correctly determine whether it is one of Fig. 5(a)-(l). We use Fig. 5(d), (f), (h), (i) as the representatives of Fig. 5(a)-(l) and Fig. 5(m), (n), (q), (r) as the representatives of others. Experimental results are summarized in Fig. 13. Obviously, the errors gradually approach zero as sample size increases.

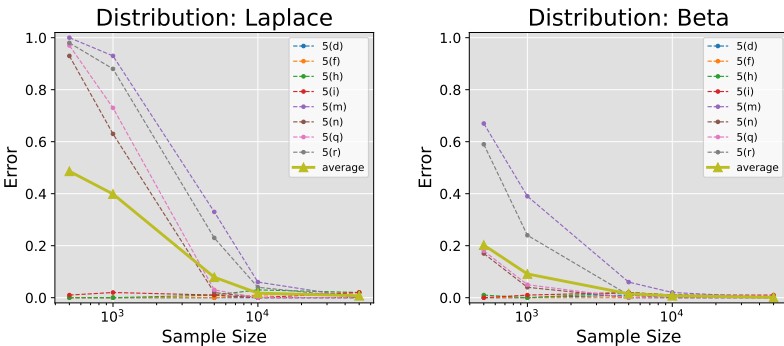

Figure 13: Error in determining the causal structure w.r.t. sample size (generalized agnostic proxy).

# E SUPPLEMENTARY EXPERIMENTAL RESULTS

This section provides more experimental results about our plug-in estimation method directly derived from Alg. 1. We would like to highlight that the experimental setup in this section differs from that in Sec. 6 in a key aspect. Specifically, in Sec. 6, we separately evaluated our estimation method's ability to determine the causal structure (with the underlying causal structure unknown) and its ability to estimate the causal effect (with the underlying causal structure known). This separation is primarily for a fair comparison with the baselines, as all three baselines (CM, cumulant, and GRICA) require the underlying causal structure to be known a priori for causal effect estimation. In this section, we always assume the underlying causal structure is unknown to our estimation method. That is, the method must estimate the causal effect based on its predicted causal structure rather than the ground truth. We only report the errors on estimating the causal effect, omitting the errors related to determining the causal structure.

§ **Comparison with RICA**

We compare our estimation method with RICA (Salehkaleybar et al., 2020), which is a baseline capable of estimating the causal effect without knowing the underlying causal structure. The experimental results are summarized in Fig. 14. Clearly, our estimation method consistently outperforms RICA.

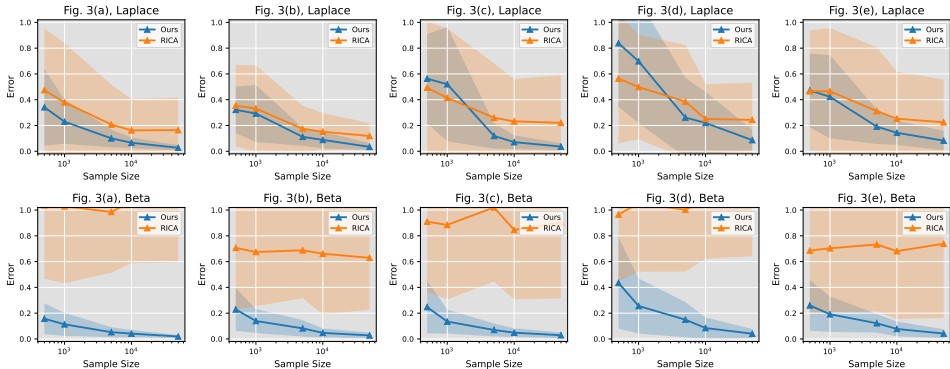

Figure 14: Comparison with RICA.

§ **Robustness to Significance Level**

The independence test employed by our estimation method is HSIC (Hilbert-Schmidt Independence Criterion) (Zhang et al., 2018), a widely-used non-parametric method capable of detecting arbitrarily complex, nonlinear dependencies. Conventionally, we set the significance level of HSIC to 0.05. Also, we evaluate the performance at a significance level of 0.1 and 0.01, which are the other two common values, as shown in Fig. 15. The experimental results demonstrate that our estimation method is robust to the choice of significance level.

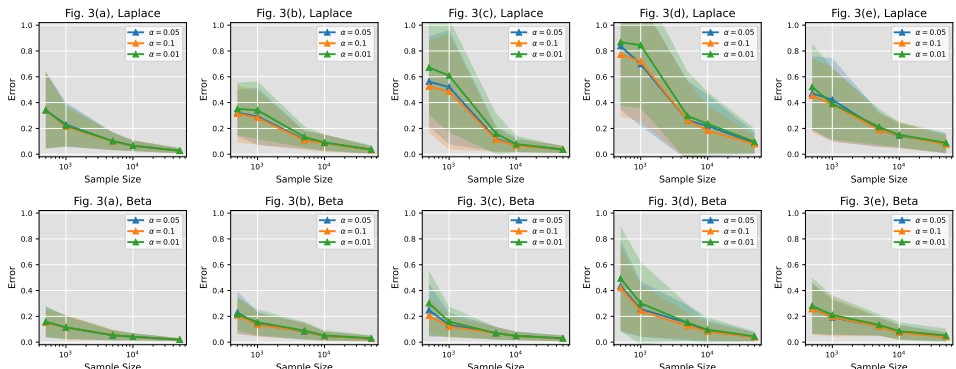

Figure 15: Robustness to significance level.

§ **Robustness to Weak Confounding**

We evaluate the robustness of our estimation method against weak confounding. Specifically, we reduce the standard error of the latent confounder $L$ to $1/10$ of its original value.

Let $R(Y|X) = Y - \frac{\text{Cov}(X,Y)}{\text{Var}(X)} X$ (aka the residual of $Y$ on $X$), it is easy to verify that there is no latent confounding if and only if $R(O|T) \perp\!\!\!\perp T$ or $R(R(O|Z), R(T|Z)) \perp\!\!\!\perp R(T|Z)$. Futhermore, if $R(O|T) \perp\!\!\!\perp T$, then $m_{OT} = \frac{\text{Cov}(T,O)}{\text{Var}(T)}$; if $R(R(O|Z), R(T|Z)) \perp\!\!\!\perp R(T|Z)$, then $m_{OT} = \frac{\text{Cov}(R(T|Z), R(O|Z))}{\text{Var}(R(T|Z))}$. Following this logic, we incorporate a pre-processing step into our estimation method: it first checks if the confounding is weak enough to be ignored, and applies different estimation strategies accordingly.

The experimental results, summarized in Fig. 16, show that our estimation method performs even better under weak confounding because of the pre-processing step.

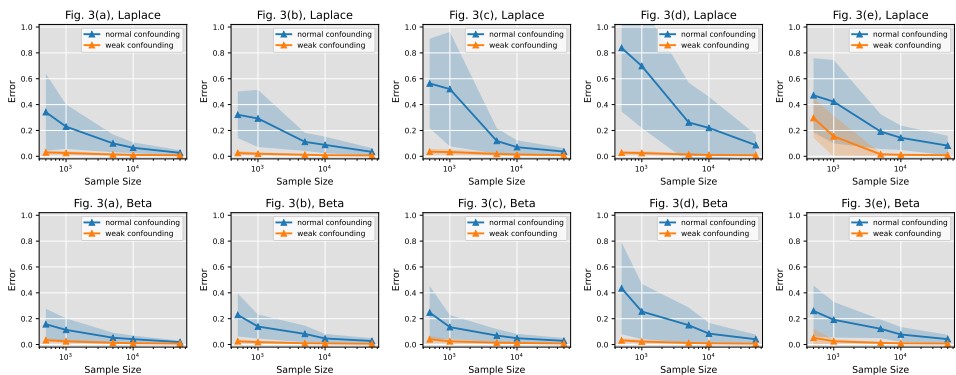

Figure 16: Robustness to weak confounding.

## § Robustness to Noisy Proxy

We evaluate the robustness of our estimation method against a noisy proxy by subjecting the proxy to measurement error. Specifically, we cannot access the true proxy $Z$ but only a noisy version $\check{Z}$, where $\check{Z} = Z + \tilde{\epsilon}$. Here, $\tilde{\epsilon}$ is a uniformly distributed noise independent of $Z$, and the standard error of $\tilde{\epsilon}$ is set to 0.25 of the standard error of $Z$. The experimental results shown in Fig. 17 indicate that our estimation method's performance does not significantly degrade even in the presence of a noisy proxy.

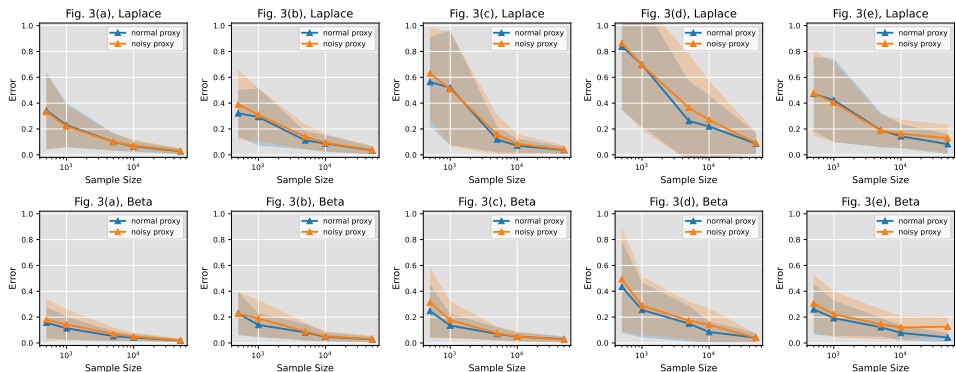

Figure 17: Robustness to noisy proxy.

## § Robustness to Nonlinearity

We evaluate the robustness of our estimation method against mild nonlinearity by letting the latent confounder causally influence the treatment, outcome, and proxy in a nonlinear manner. Taking the treatment $T$ in Fig. 1(a) as an example, we set $T = a_{TL}(L + 0.25 \sin(L))$. The results (Fig. 18) show that our estimation method's performance does not significantly deteriorate when the nonlinearity is not excessively strong.

## § Robustness to Gaussianity

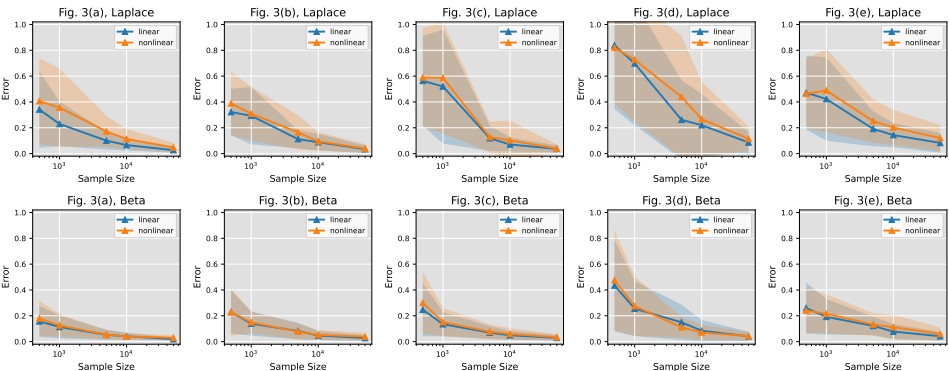

Figure 18: Robustness to nonlinearity.

We evaluate the robustness of our estimation method against Gaussianity by generating the exogenous noises from a generalized Gaussian distribution. Specifically, the probability density function of a generalized Gaussian $X$ is given by $f_X(x;\beta) = \frac{\beta}{2\Gamma(1/\beta)} \exp(-|x|^\beta)$. Note that $X$ follows a Gaussian distribution when $\beta = 2$. We test the performance for $\beta = 0.75, 1.0, 1.25, 1.5$. The experimental results (Fig. 19) demonstrate that our estimation method performs better when the distribution exhibits stronger non-Gaussianity. Even in cases with weak non-Gaussianity, it remains effective in the large-sample regime.

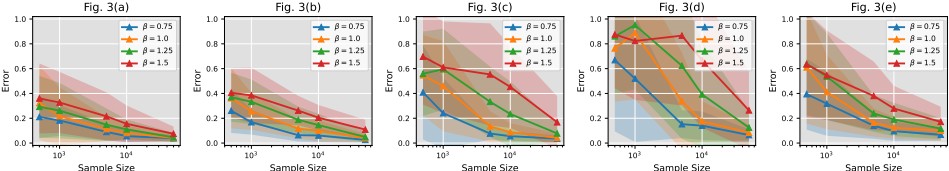

Figure 19: Robustness to Gaussianity.

## F    USAGE OF LLMs

LLMs are utilized as a supportive tool for enhancing language fluency, correcting grammar, and rephrasing sentences. The core method development in this research does not involve LLMs as any important, original, or non-standard components.

