# OpenReview forum: "Causal Effect Identification in the Presence of Latent Confounding with a Single Imperfect Proxy Variable"
_ICLR.cc/2026/Conference — Submitted to ICLR 2026_

### Official Review · Reviewer_o4cu · 2025-10-31

**Soundness:** 3
**Presentation:** 1
**Contribution:** 2
**Rating:** 2
**Confidence:** 3

**Summary:**

This paper tackles the long-standing challenge of causal effect identification under latent confounding when only one proxy variable is available. Within the lvLiNGAM framework, the authors propose a new algorithm that works with a single agnostic proxy, meaning the proxy’s causal relation with treatment and outcome can be arbitrary and unknown. Their method first derives candidate causal effects using cross-cumulants and then selects the correct one via independence tests. Theoretically, they claim asymptotic consistency and, if identification is impossible, explicit detection of unidentifiability. Experiments on synthetic and real datasets demonstrate effectiveness and consistency advantages over GRICA and other baselines.

**Strengths:**

1. The paper investigates an important and fundamental problem in causal inference, causal effect identification and estimation in the presence of latent confounders.

1. It proposes an algorithm that extends existing methods to handle agnostic proxies, broadening the applicability of proxy-based causal identification.

**Weaknesses:**

1. The linear non-Gaussian additive noise assumption may be too restricted and limit its applications. The method collapses under even slight model misspecification (e.g., near-Gaussian noise or weak nonlinearity). Could the authors discuss possible extensions to nonlinear settings?
1. How can interval confidence or uncertainty quantification be obtained for the estimated causal effect?
1. The novelty over prior works (e.g., [1, 2, 3]) is insufficiently discussed. The key methodological distinctions and theoretical advances should be made clearer.
1. The writing is poor. Many theoretical results are listed without sufficient intuition or illustrative examples, making the paper difficult to follow.
1. The proposed method appears to require a large sample size for accurate effect estimation. The authors should discuss small-sample behavior or variance reduction strategies.
1. The experimental evaluation lacks robustness checks, such as tests under noisy proxies, weak confounding, or varying non-Gaussianity strength.
1. More baselines should be included to strengthen the empirical comparison beyond GRICA and cumulant-based methods.
1. The method requires multiple cumulant estimations and independence tests for several possible graph configurations. No complexity or runtime analysis is provided.



[1] Cai, Ruichu, et al. "Causal discovery with latent confounders based on higher-order cumulants." *International conference on machine learning*. PMLR, 2023.

[2] Kivva, Yaroslav, Saber Salehkaleybar, and Negar Kiyavash. "A cross-moment approach for causal effect estimation." *Advances in Neural Information Processing Systems* 36 (2023): 9944-9955.

[3] Chen, Wei, et al. "Identification of causal structure with latent variables based on higher order cumulants." *Proceedings of the AAAI Conference on Artificial Intelligence*. Vol. 38. No. 18. 2024.

**Questions:**

Please see above

---

> ### Author Response · Authors · 2025-11-24
> **Part 1/6**
>
> We sincerely thank the reviewer for the comprehensive review and valuable feedback. We have carefully considered all the points raised, ranging from theoretical clarifications to empirical evaluations, and have revised the manuscript to improve its overall quality and readability.
>
> > Q1: Small-sample performance
>
> First, we fully agree with the reviewer that small-sample performance is a critical practical consideration. However, we respectfully note that finite-sample performance is fundamentally a concern of causal effect estimation, whereas the primary focus of this work is causal effect identification. We have thoroughly revised the manuscript to more clearly delineate the scope of our primary contribution.
>
> 1. Causal effect identification addresses the **theoretical** question of whether and how the causal effect of interest can be uniquely determined from the observational population distribution (i.e., **infinite data**). In contrast, causal effect estimation addresses the **practical** question of how to compute point estimates for the causal effect and quantify the associated uncertainty (e.g., confidence intervals) given a **finite sample** drawn from the observational population distribution. Identification is the prerequisite and theoretical foundation for consistent estimation.
>
> 2. Our core contributions lie in causal effect identification: we develop a novel causal effect identification procedure that requires only a single agnostic proxy and prove its soundness, that is, given the observational population distribution, it correctly identifies the true causal effect when identifiable, and correctly reports unidentifiability otherwise. To conduct experiments, we implement a basic plug-in estimation method by directly substituting sample-level cumulants and independence tests into our identification procedure. We wish to highlight that this implementation is intended specifically to empirically validate the correctness of our theoretical results rather than to optimize finite-sample performance. The observed convergence of estimation error to zero as the sample size increases sufficiently validates our theoretical results.
>
> 3. Existing literature demonstrates that causal effect identification is widely recognized as a self-contained theoretical contribution independent of finite-sample estimation optimization. For instance, [1,2] focus exclusively on theoretical contributions without providing any estimation methods; [3,4] use basic plug-in estimation methods similar to ours to conduct experiments; and [5] only augments the basic plug-in estimator with a post-processing step that yields observable gains primarily in large-sample regimes, while still maintaining the primary focus on identification theory.
>
> Second, we frankly acknowledge that the finite-sample performance of our plug-in estimation method is suboptimal, which is a known characteristic of estimation methods based on higher-order statistics [3-5]. However, inspired by the reviewer’s comments, we have implemented an adaptive strategy to effectively mitigate this issue and will conduct an in-depth investigation in future work.
>
> 1. Since the estimation variance increases with the order of the cumulant, utilizing lower-order cumulants whenever possible helps to improve stability. Note that $g(T,Z)$ in Equation (9) originally expressed via fourth-order cross-cumulants as $\mathrm{sgn}(Cov(T,Z)) \sqrt{C_{3,1}(T,Z) / C_{1,3}(T,Z)}$ can be equivalently expressed using third-order cross-cumulants $C_{2,1}(T,Z) / C_{1,2}(T,Z)$ when $C_{1,2}(T,Z) \neq 0$. Therefore, we update our estimation method to adaptively switch to third-order cumulants when the correlation between $T$ and $Z^2$ (i.e., a normalized version of $C_{1,2}(T,Z)$) exceeds a threshold (e.g., 0.1). When the latent confounder follows an asymmetric distribution (this guarantees $C_{1,2}(T,Z) \neq 0$) such as the Beta distribution with $\alpha=1/3, \beta=2/3$, the experimental results below demonstrate that this update remarkably reduces estimation errors in small-sample regimes.
>
>     |Sample Size|Method|Fig. 3(a)|Fig. 3(d)|Fig. 3(e)|
>     |-|-|-|-|-|
>     |500|w/o update|0.24±0.28|0.43±0.37|0.33±0.31|
>     | |w/ update|**0.16±0.15**|**0.34±0.32**|**0.24±0.21**|
>     |1000|w/o update|0.18±0.15|0.26±0.23|0.29±0.25|
>     | |w/ update|**0.12±0.09**|**0.24±0.22**|**0.23±0.22**|
>     |5000|w/o update|0.09±0.06|0.17±0.13|0.13±0.11|
>     | |w/ update|**0.07±0.05**|**0.14±0.13**|**0.11±0.09**|
>     |10000|w/o update|0.05±0.04|0.15±0.13|0.11±0.08|
>     | |w/ update|0.05±0.04|**0.13±0.14**|0.11±0.08|
>
>     (*Note*: we only provide results for Fig. 3(a), (d), and (e) because the function $g$ is not invoked for causal effect estimation for Fig. 3(b) and (c) according to Algorithm 1 in our paper)

---

> ### Author Response · Authors · 2025-11-24
> **Part 2/6**
>
> 2. The suboptimal small-sample performance is primarily attributable to the high variance of higher-order cumulant estimators, so any advancements in the estimation of high-order statistics will immediately benefit our estimation method. In future work, we will conduct an in-depth investigation of related literature [6-8] and integrate more sophisticated higher-order cumulant estimators to further improve the small-sample performance of our estimation method.
>
> ### Reference
>
> [1] Causal effect identifiability under partial-observability. ICML 2020.\
> [2] On identifiability of conditional causal effects. UAI 2023.\
> [3] A Cross-Moment Approach for Causal Effect Estimation. NeurIPS 2023.\
> [4] Causal Effect Identification in Heterogeneous Environments from Higher-Order Moments. UAI 2025.\
> [5] Causal Effect Identification in lvLiNGAM from Higher-Order Cumulants. ICML 2025.\
> [6] L-moments: analysis and estimation of distributions using linear combinations of order statistics. Journal of the Royal Statistical Society Series B: Statistical Methodology, 1990.\
> [7] Comparing measures of sample skewness and kurtosis. Journal of the Royal Statistical Society Series D: The Statistician, 1998\
> [8] On more robust estimation of skewness and kurtosis. Finance Research Letters 2004.
>
> > Q2: lvLiNGAM framework & extension to nonlinear scenarios
>
> First, we focus on the lvLiNGAM framework for three primary reasons: (1) causal effect identification with a single proxy is often impossible outside the lvLiNGAM framework, let alone identification with a single agnostic proxy; (2) lvLiNGAM can be applied to solve real-world problems; and (3) lvLiNGAM remains a vibrant and active area of research in recent causal inference literature.
>
> 1. Literature has established that it is often impossible to identify the causal effect with a single proxy outside the lvLiNGAM framework. Specifically, [1] have proven that in **linear Gaussian** models with latent variables, the causal effect is unidentifiable even with a single **isolated** proxy. Furthermore, it is known that we cannot recover the true causal effect with a single proxy in general non-parametric models [2-4], unless many other restrictive assumptions are satisfied (e.g.,  the proxy is an isolated proxy and the distribution of the proxy conditioned on the latent confounder is known a priori [3]). Our setting allows for an agnostic proxy, rendering the identification problem even more challenging. This underscores the necessity of focusing on the lvLiNGAM framework.
>
> 2. By leveraging linearization techniques around the operating point of a system, complex causal mechanisms can be effectively approximated by linear functions in many real-world applications. Indeed, LiNGAM, a simplified version of lvLiNGAM without latent confounders, has been used to address practical problems across diverse domains, e.g., economics [5], marketing [6], neuroscience [7], epidemiology [8], chemistry [9], and environmental science [10].
>
> 3. The linear model continues to be a central subject of investigation in current causal inference research [11-16]. This validates its continued relevance and importance to the community.
>
> Second, we agree that extending our framework to nonlinear scenarios is a promising direction. In our framework, the cross-cumulant constraint and the independence constraint jointly serve as a vanishing condition, indicating the absence of a causal relationship between two variables sharing a common cause. Consequently, if we find a specific value $\alpha$ such that the treatment $T$ and the residual $O - \alpha T$ satisfy these two constraints, then $\alpha$ is exactly the causal effect. This perspective suggests a viable pathway for generalizing our framework to certain nonlinear structural causal models (e.g., additive noise models subject to specific assumptions that guarantee identifiability) by formulating nonlinear variants of these constraints, possibly through kernel representations as in [4]. We leave this exciting development for future work.
>
> Finally, we evaluate the robustness against mild nonlinearity and Gaussianity of the plug-in estimation method directly derived from our identification procedure. The experimental results show that its performance does not significantly deteriorate when the nonlinearity is not excessively strong and it remains effective in the large-sample regime even in cases with weak non-Gaussianity. Please see our response to your Q5 for more details.
>
> ### Reference
>
> [1] A Cross-Moment Approach for Causal Effect Estimation. NeurIPS 2023.\
> [2] On measurement bias in causal inference. UAI 2010.\
> [3] Measurement bias and effect restoration in causal inference. Biometrika 2014.\
> [4] Kernel Single Proxy Control for Deterministic Confounding. Arxiv 2023.

---

> ### Author Response · Authors · 2025-11-24
> **Part 3/6**
>
> [5] Causal inference by independent component analysis: Theory and applications. Oxford Bulletin of Economics and Statistics 2013.\
> [6] Causal inference for contemporaneous effects and its application to tourism product sales data. Journal of Marketing Analytics 2022.\
> [7] Asymmetric directed functional connectivity within the frontoparietal motor network during motor imagery and execution. NeuroImage 2022.\
> [8] Pairwise measures of causal direction in the epidemiology of sleep problems and depression. Plos one 2012.\
> [9] Origin of the spectral shifts among the early intermediates of the rhodopsin photocycle. Journal of the American Chemical Society 2014.\
> [10] Validation of causal inference data using DirectLiNGAM in an environmental small-scale model and calculation settings. MethodsX 2024.\
> [11] Causal Effect Identification in LiNGAM Models with Latent Confounders. ICML 2024.\
> [12] Efficient and Trustworthy Causal Discovery with Latent Variables and Complex Relations. ICLR 2025.\
> [13] Recovery of causal graph involving latent variables via homologous surrogates. ICLR 2025.\
> [14] Linear SCM Identification in the Presence of Confounders and Gaussian Noise. ICLR 2025.\
> [15] Causal Effect Identification in lvLiNGAM from Higher-Order Cumulants. ICML 2025.\
> [16] Causal Effect Identification in Heterogeneous Environments from Higher-Order Moments. UAI 2025.
>
> > Q3: Novelty
>
> We thank the reviewer for allowing us to elaborate on the novelty of our work and its distinction from prior literature [1-3].
>
> First, the primary innovations of our work can be summarized into three key aspects: (1) new problem, (2) new identification procedure, and (3) new theoretical results.
>
> 1. Motivated by the observation that in real-world applications, obtaining multiple proxies is often infeasible and the causal relationship between the proxy and the treatment/outcome is typically flexible and unknown, we first investigate the challenging problem of identifying the causal effect in the presence of latent confounding with a single agnostic proxy, where "agnostic proxy" encapsulates both flexibility and uncertainty.
>
> 2. Since the complexity of the agnostic proxy precludes identifying the causal effect via a simple analytical formula, we develop a novel two-stage identification procedure. This identification procedure first derives a set of candidate solutions based on cross-cumulants and subsequently isolates the valid solution by examining certain independence relationships.
>
> 3. We present and prove a series of new theoretical results that collectively establish the soundness of our identification procedure. Specifically, given the observational population distribution, it correctly identifies the true causal effect when identifiable, and correctly reports unidentifiability otherwise.
>
> Second, we detail the similarities and distinctions between our work and [1-3] in the following.
>
> - (Similarity) Both [1-3] and our work assume the data is generated by a lvLiNGAM. In addition, [1-3] and our work both utilize high-order cross-cumulants/moments (note that moment and cumulant are two informationally equivalent characterizations of a probability distribution).
>
> - (Difference between our work and [1]) The goal of [1] is to identify both the underlying causal structure and causal coefficients. However, they rely on a restrictive assumption: within the underlying causal structure (after pruning those observed variables that have no latent parents), there must exist two observed variables that share exactly one confounder and have no causal relationship with each other. In Fig. 3 in our paper, only Fig. 3(a) satisfies this assumption. Consequently, mapped to our problem setting, [1] can at most achieve the following: If the underlying causal structure is known a priori to be Fig. 3(a), it can correctly identify the causal effect of $T$ on $O$.
>
> - (Difference between our work and [2]) The goal of [2] is to identify the causal effect. Although they can also work with only a single proxy, they require that the proxy be isolated, that is, it must have no causal relationship with either the treatment or the outcome. In Fig. 3 in our paper, only Fig. 3(a) satisfies this condition. Consequently, mapped to our problem setting, [2] can at most achieve the following: If the underlying causal structure is known a priori to be Fig. 3(a), it can correctly identify the causal effect of $T$ on $O$.

---

> ### Author Response · Authors · 2025-11-24
> **Part 4/6**
>
> - (Difference between our work and [3]) The goal of [3] is to identify the underlying causal structure. Although they focus on bivariate scenarios, their results can generalize to multivariate settings under the assumption that within the underlying causal structure (after pruning those observed variables that have no latent parents), every pair of observed variables shares exactly one confounder. *In Fig. 3 in our paper, Fig. 3(e), (g), and (h) violate this assumption* (e.g., in Fig. 3(e), $T$ and $O$ share two confounders: $L$ and $Z$). Consequently, mapped to our problem setting, [3] can at most achieve the following: if the underlying causal structure is known a priori to be one of Fig. 3(a), (b), (c), and (d) but the exact instance is unknown, [3] can identify the causal structure but not the causal effect of interest. Furthermore, besides the covariances (second-order cumulants), we utilize only fourth-order cumulants, whereas [3] necessitates fifth-order or higher cumulants.
>
>   **UPDATE**: We note a typo in the italicized text above, which should be corrected to: *In Fig. 3 in our paper, **Fig. 3(e), (f), (g), and (h)** violate this assumption.* In the original text, we inadvertently omitted Fig. 3(f).
>
>
> ### Reference
>
> [1] Causal discovery with latent confounders based on higher-order cumulants. ICML 2023.\
> [2] A cross-moment approach for causal effect estimation. NeurIPS 2023.\
> [3] Identification of causal structure with latent variables based on higher order cumulants. AAAI 2024.
>
> > Q4: Writing
>
> We thank the reviewer for the constructive feedback on readability. We have significantly reorganized the Main Results section to bridge the gap between theoretical rigor and intuitive understanding.
>
> 1. We dedicate an entire subsection (i.e., Section 3.1) to a motivating example. This subsection intuitively illustrates how cross-cumulants are utilized to generate candidate solutions and why independence relationships can be subsequently employed to isolate the valid one.
>
> 2. We have added intuitive interpretations prior to presenting each of Theorems 1-6 that provide guidelines on how to handle the flexibility of the agnostic proxy. In particular, for Theorems 2-5, we explicitly elucidate how their derivations align with the core intuition established in the motivating example of Section 3.1.
>
> 3. We have similarly added intuitive explanations for Corollaries 1-5 that provide guidelines on how to handle the uncertainty of the agnostic proxy. In particular, since Corollaries 2-5 are derived from Theorems 2-5 following a uniform pattern, we provide a unified explanation of their derivation logic to avoid redundancy and improve clarity.
>
> > Q5: Robustness & confidence interval & baselines & complexity/runtime
>
> While these questions fall within the scope of estimation (as discussed in our response to your Q1), we appreciate the comprehensive suggestions and have conducted extensive additional experiments and analyses to address them.
>
> ### I. Robustness of our estimation method
>
> - (Weak confounding) We evaluate the robustness of our estimation method against weak confounding. Specifically, we reduce the standard error of the latent confounder $L$ to 0.1 of its original value. Let $R(Y|X) = Y - \frac{\mathrm{Cov}(X,Y)}{\mathrm{Var}(X)} X$ (aka the residual of $Y$ on $X$), it is easy to verify that there is no latent confounding if and only if $R(O|T) \perp T$ or $R(R(O|Z), R(T|Z)) \perp R(T|Z)$. Furthermore, if $R(O|T) \perp T$, then $m_{OT}=\frac{\mathrm{Cov}(T,O)}{\mathrm{Var}(T)}$; if $R(R(O|Z), R(T|Z)) \perp R(T|Z)$, then $m_{OT}=\frac{\mathrm{Cov}(R(T|Z),R(O|Z))}{\mathrm{Var}(R(T|Z))}$. Following this logic, we incorporate a pre-processing step into our estimation method: it first checks if the confounding is weak enough to be ignored, and applies different estimation strategies accordingly. The experimental results show that our estimation method performs even better under weak confounding because of the pre-processing step. We present the results for Fig. 3(b) with Laplace exogenous noises here. Please refer to Figure 16 in Appendix E for the comprehensive experimental results.
>
>     | Sample Size | 500 | 1000 | 5000 | 10000 | 50000 |
>     |-|-|-|-|-|-|
>     |Normal confounding|0.32±0.18|0.29±0.22|0.11±0.07|0.09±0.06|0.03±0.03|
>     |Weak confounding|0.03±0.02|0.02±0.01|0.01±0.01|0.01±0.01|0.01±0.00|

---

> ### Author Response · Authors · 2025-11-24
> **Part 5/6**
>
> - (Noisy proxy) We evaluate the robustness of our estimation method against a noisy proxy by subjecting the proxy to measurement error. Specifically, we cannot access the true proxy $Z$ but only a noisy version $\tilde{Z}$, where $\tilde{Z} = Z + \tilde{\epsilon}$. Here, $\tilde{\epsilon}$ is a uniformly distributed noise independent of $Z$, and the standard error of $\tilde{\epsilon}$ is set to 0.25 of the standard error of $Z$. The experimental results indicate that our estimation method's performance does not significantly degrade even in the presence of a noisy proxy. We present the estimation errors for Fig. 3(b) with Laplace exogenous noises here. Please refer to Figure 17 in Appendix E for the comprehensive experimental results.
>
>     | Sample Size | 500 | 1000 | 5000 | 10000 | 50000 |
>     |-|-|-|-|-|-|
>     |Normal proxy|0.32±0.18|0.29±0.22|0.11±0.07|0.09±0.06|0.03±0.03|
>     |Noisy proxy|0.39±0.27|0.31±0.21|0.14±0.09|0.09±0.07|0.04±0.03|
>
> - (Nonlinearity) We evaluate the robustness of our estimation method against nonlinearity by letting the latent confounder causally influence the treatment, outcome, and proxy in a nonlinear manner. Taking the treatment $T$ in Fig. 3(a) as an example, we set $T = a_{TL} (L + 0.25\sin(L))$. The experimental results show that our estimation method's performance does not significantly deteriorate when the nonlinearity is not excessively strong. We present the estimation errors for Fig. 3(b) with Laplace exogenous noises here. Please refer to Figure 18 in Appendix E for the comprehensive experimental results.
>
>     | Sample Size | 500 | 1000 | 5000 | 10000 | 50000 |
>     |-|-|-|-|-|-|
>     |Linear|0.32±0.18|0.29±0.22|0.11±0.07|0.09±0.06|0.03±0.03|
>     |Nonlinear|0.39±0.25|0.31±0.21|0.16±0.13|0.09±0.07|0.04±0.03|
>
> - (Gaussianity) We evaluate the robustness of our estimation method against Gaussianity by generating the exogenous noises from a generalized Gaussian distribution. Specifically, the probability density function of a generalized Gaussian $X$ is given by $f_X(x; \beta) = \frac{\beta}{2 \Gamma(1/\beta)} \exp(-|x|^\beta)$. Note that $X$ follows a Gaussian distribution when $\beta = 2$. We test the performance for $\beta = 0.75, 1.0, 1.25, 1.5$. The experimental results demonstrate that our estimation method performs better when the distribution exhibits stronger non-Gaussianity. Even in cases with weak non-Gaussianity, it remains effective in the large-sample regime. We present the estimation errors for Fig. 3(b) with Laplace exogenous noises here. Please refer to Figure 19 in Appendix E for the comprehensive experimental results.
>
>     | Sample Size | 500 | 1000 | 5000 | 10000 | 50000 |
>     |-|-|-|-|-|-|
>     |$\beta=0.75$|0.26±0.18|0.17±0.10|0.06±0.05|0.06±0.05|0.03±0.02|
>     |$\beta=1.0$|0.36±0.24|0.26±0.15|0.12±0.08|0.10±0.09|0.04±0.03|
>     |$\beta=1.25$|0.37±0.20|0.33±0.18|0.19±0.11|0.14±0.09|0.05±0.04|
>     |$\beta=1.5$|0.41±0.20|0.38±0.22|0.26±0.12|0.20±0.10|0.11±0.08|
>
> ### II. Confidence interval
>
> The confidence interval can be obtained via Bootstrap, a non-parametric statistical method that estimates uncertainty through resampling. Specifically, given a dataset of size $n$,
> 1. Generate a bootstrap sample of size $n$ from the original dataset via random sampling with replacement.
> 2. Estimate the causal effect of interest using the bootstrap sample obtained in Step 1.
> 3. Repeat Steps 1 and 2 $B$ times (e.g., $B = 1000$) to obtain $B$ estimates.
> 4. Arrange the $B$ estimates in ascending order.
> 5. Extract the 2.5th and 97.5th percentiles from the sorted estimates to obtain the 95% confidence interval.
>
> Following this process, when the sample size is 10000 and the causal effect of interest is set to 0.8, the 95% confidence intervals are estimated and summarized as follows.
>
> |Laplace distribution| | | | | Beta distribution | | | | |
> |-|-|-|-|-|-|-|-|-|-|
> |Fig. 3(a)|Fig. 3(b)|Fig. 3(c)|Fig. 3(d)|Fig. 3(e)|Fig. 3(a)|Fig. 3(b)|Fig. 3(c)|Fig. 3(d)|Fig. 3(e)|
> |[0.73,0.87]|[0.59,0.90]|[0.56,0.85]|[0.66,1.05]|[0.51,0,93]|[0.77,0.87]|[0.65,0.92]|[0.69,1.01]|[0.78,1.02]|[0.73,1.02]|

---

> ### Author Response · Authors · 2025-11-24
> **Part 6/6**
>
> ### III. Baselines
>
> In our experiments, we employ GM [1], GRICA [2], and Cumulant [3] as baselines, as they represent the most recent and representative studies on causal effect identification within the lvLiNGAM framework using only a single proxy. Specifically, GRICA (2024) utilizes overcomplete independent component analysis (OICA), whereas GM (2023) and Cumulant (2025) rely directly on high-order moments/cumulants. Together, these studies exemplify the two primary technical paradigms in this domain. We have also incorporated RICA [4], another OICA-based approach, as an additional baseline. Experimental results demonstrate that our estimation method consistently outperforms this new baseline. We present the estimation errors for Fig. 3(b) with Laplace exogenous noises here. Please refer to Figure 14 in Appendix E for the comprehensive experimental results.
>
> | Sample Size | 500 | 1000 | 5000 | 10000 | 50000 |
> |-|-|-|-|-|-|
> |Ours|**0.32±0.18**|**0.29±0.22**|**0.11±0.07**|**0.09±0.06**|**0.03±0.03**|
> |RICA|0.35±0.32|0.33±0.34|0.17±0.18|0.15±0.14|0.12±0.10|
>
> ### IV. Complexity/Runtime
>
> The computational complexity of the plug-in estimation method directly derived from our identification procedure (Alg. 1) is low. Even in the worst-case scenario, it requires performing only 11 independence tests and estimating 12 high-order cumulants. Specifically:
> - (Independence test) In Alg. 1, line 1 necessitates examining 1 independence relationship; line 3 requires 2; line 5 requires 2; line 7 requires a total of 2; and line 9 requires a total of 4. This sums to 11.
> - (Cumulants) All cross-cumulants involved in our estimation method can be expressed as linear combinations of the following 12 fundamental cross-cumulants: $C_4(T)$, $C_4(O)$, $C_4(Z)$, $C_{3,1}(Z,T)$, $C_{3,1}(T,Z)$, $C_{3,1}(Z,O)$, $C_{3,1}(O,Z)$, $C_{3,1}(T,O)$, $C_{3,1}(O,T)$, $C_{2,2}(Z,T)$, $C_{2,2}(Z,O)$, and $C_{2,2}(T,O)$.
>
> We compare the runtime of our estimation method against GRICA when the sample size is 10000, and the results are presented below. It is important to note that we grant GRICA a significant advantage by assuming the underlying causal structure is known to it, whereas it remains unknown to our estimation method. Consequently, our estimation method must estimate both the causal structure and the causal effect, while GRICA needs only estimate the causal effect. Despite this additional computational burden, our method still achieves a lower runtime than GRICA.
>
> ||Laplace distribution| | | | | Beta distribution | | | | |
> |-|-|-|-|-|-|-|-|-|-|-|
> ||Fig. 3(a)|Fig. 3(b)|Fig. 3(c)|Fig. 3(d)|Fig. 3(e)|Fig. 3(a)|Fig. 3(b)|Fig. 3(c)|Fig. 3(d)|Fig. 3(e)|
> |GRICA|0.18s|0.18s|0.18s|0.18s|0.19s|0.18s|0.18s|0.18s|**0.18s**|0.19s|
> |Ours|**0.02s**|**0.10s**|**0.12s**|**0.16s**|**0.12s**|**0.02s**|**0.13s**|**0.16s**|**0.18s**|**0.18s**|
>
> ### Reference
>
> [1] A Cross-Moment Approach for Causal Effect Estimation. NeurIPS 2023.\
> [2] Causal Effect Identification in LiNGAM Models with Latent Confounders. ICML 2024.\
> [3] Causal Effect Identification in lvLiNGAM from Higher-Order Cumulants. ICML 2025.\
> [4] Learning Linear Non-Gaussian Causal Models in the Presence of Latent Variables. JMLR 2020.

---

> > ### Comment · Reviewer_o4cu · 2025-11-26
> >
> > Thank the authors for the detailed responses. Most of my concerns have been satisfactorily addressed. I am happy to raise my score at the end of the rebuttal by considering all the discussion between authors and all reviewers.
> >
> > Regarding my W1, W4, W5, W6, and W8, the added experiments help solve my concerns.
> >
> > Regarding my W2, I remain curious about the inference strategies. Is it possible to obtain the properties beyond consistency, e.g., asymptotic normality? The bootstrap method is computationally expensive, especially when the cumulant-based method is already time-consuming. And it would also be helpful to evaluate the predicted intervals using the coverage probability metric.
> >
> > Regarding my W3 and W7:
> >   1. It could be better to include [3] as a baseline, as it applies to many of the structures shown in Fig. 3 of the paper.
> >   2. The authors claim that Fig. 3e, 3g, and 3h violate the one-latent-confounder assumption. While Fig. 3e clearly violates it, in Fig. 3g and 3h, $Z$ appears to act as mediator and collider respectively, instead of confounder, and similar for Fig. 3f. Does this mean that Fig. 3e is identifiable for the proposed method but non-identifiable for [3], whereas Fig. 3f, 3g, and 3h are non-identifiable for the proposed method but identifiable for [3]?
> >
> > [3] Identification of causal structure with latent variables based on higher order cumulants. AAAI 2024.
> >
> > Please let me know if I have misunderstood anything in the rebuttal. I am open to further discussion and am happy to adjust my score accordingly.

---

> ### Author Response · Authors · 2025-11-26
>
> Thank you very much for your prompt response. We are glad to clarify some misunderstandings
>
> > Q6: Detailed discussion on a related work [3]
>
> First, we wish to highlight that the algorithm proposed in [3] is designed for causal **structure** identification, not causal **effect** identification. While it can determine the existence and direction of a causal edge, it cannot quantify its strength.
>
> Second, [3] primarily focuses on settings involving exactly two observed variables that share exactly one confounder. To generalize their results to scenarios involving multiple observed variables, one must assume that **every** pair of observed variables shares exactly one confounder. Under this assumption, one can apply the algorithm in [3] to every pair of observed variables and aggregate the results to reconstruct the complete causal **structure**. Clearly, this assumption does not hold in any case of **Fig. 3(e), (f), (g), and (h)**: in our Fig. 3(e), both $L$ and $Z$ are common parents of $T$ and $O$; in Fig. 3(f), (g), and (h), both $L$ and $T$ are common parents of $Z$ and $O$.
>
> Consequently, when mapped to our setting, [3] achieves at most the following: If the underlying causal structure is known a priori to be one of Fig. 3(a), (b), (c), and (d) but the exact instance is unknown, [3] can identify the underlying causal **structure** but cannot identify the causal **effect** of interest. For this structural estimation task, we compared [3] with our approach using Beta-distributed exogenous noise. The errors are summarized below.
>
> |Sample size|500|1000|5000|10000|50000|
> |-|-|-|-|-|-|
> |[3]|0.73|0.66|0.51|0.46|0.38|
> |Ours|0.45|0.31|0.21|0.17|0.15|
>
> Evidently, the performance of [3] lags behind ours. The primary reason is that [3] relies entirely on cumulants and utilizes fifth-order cumulants. In contrast, our algorithm mitigates reliance on cumulants by leveraging independence tests and uses at most fourth-order cumulants. Independence testing is a mature field with many robust algorithms, whereas high-order cumulant estimation is widely acknowledged to suffer from high variance, especially when the order is high and the sample size is small.
>
> ### Reference
>
> [3] Identification of causal structure with latent variables based on higher order cumulants. AAAI 2024.
>
> > Q7: Bootstrap
>
> We respectfully argue that the Bootstrap method is a suitable choice for our setting.
> 1. As explained in the Part IV of our response to your Q5, the computational complexity of our estimation method is low. In particular, we utilize cumulants only up to the fourth order, so the computation of cumulants is not computationally expensive. Therefore, performing Bootstrap resampling is computationally feasible in our setting.
> 2. Unlike simple estimators derived from analytical formulas, our estimation method involves a discrete selection process: it employs independence tests to choose the valid solution from multiple candidate ones. This mechanism makes the estimator non-smooth, rendering the theoretical derivation of its asymptotic distribution extremely challenging.
> 3. Our setting is distribution-free, assuming only that the exogenous noises are non-Gaussian without any prior knowledge of their specific distributions. In such a flexible setting without parametric assumptions, deriving closed-form inference results is extremely challenging.
>
> Also, we respectfully reiterate that the primary contribution of this work is causal effect identification rather than estimation. While we agree that establishing asymptotic properties and evaluating coverage probabilities are important steps towards developing a fully mature estimation method, these tasks fundamentally fall within the scope of estimation theory and are outside the scope of this work.

---

> > ### Comment · Reviewer_o4cu · 2025-11-27
> >
> > Thanks for the clarifications.
> >
> > Regarding Q6, I carefully checked [3] and found that their method cantheir method can only narrow down the causal effect to two possible candidates (the mixing coefficients in [3]), without determining exactly which one is the true effect. This helps understand the contribution of the authors' work.
> >
> > Regarding Q7, while the overall estimator is nonsmooth, I believe that certain properties of the estimator for each candidate graph could still be analyzed, potentially using results on the asymptotic normality of cumulant/moment estimators under certain moment conditions without requiring distribution assumptions. That said, it is understandable for me that this paper focuses on effect identification without delving into these theoretical aspects, and this does not affect my final score.
> >
> > In summary, I have no further questions. I will carefully read the comments from other reviewers before adjusting my score.

---

> > > ### Author Response · Authors · 2025-11-27
> > >
> > > Dear Reviewer `o4cu`,
> > >
> > > We sincerely thank you for the time and effort dedicated to this active and constructive discussion. We are delighted to learn that our responses have successfully resolved all your concerns. We also deeply appreciate your recognition of the contributions of our paper. We look forward to your final assessment.
> > >
> > > Best regards,\
> > > Authors

---

### Official Review · Reviewer_kuEe · 2025-11-01

**Soundness:** 3
**Presentation:** 2
**Contribution:** 2
**Rating:** 4
**Confidence:** 4

**Summary:**

This paper tackles the challenging problem of identifying causal effects in the presence of latent confounders. Within the lvLiNGAM framework, the authors propose a novel method that requires only a single proxy variable for the latent confounder. The approach derives candidate solutions using cross-cumulants and then selects the valid one through independence tests.

**Strengths:**

1. The method is designed to handle settings with multiple latent confounders, substantially relaxing key limitations of existing approaches and broadening its applicability to more complex causal structures.
2. The theoretical foundation, built upon the non-Gaussianity assumption of lvLiNGAM, is sound. The property of explicitly reporting unidentifiability is a crucial and honest feature, preventing users from drawing false conclusions when the data is insufficient for identification.

**Weaknesses:**

1. The proposed method appears to rely on large sample sizes to achieve satisfactory performance, which may limit its utility in data-scarce scenarios.
2. Although higher-order cumulants provide valuable statistical information, their estimation can be sensitive to sampling variability. Moreover, in settings involving multiple latent variables, the iterative application of the procedure may lead to error accumulation, potentially compromising estimation accuracy.

**Questions:**

See above.

---

> ### Author Response · Authors · 2025-11-24
> **Part 1/2**
>
> We appreciate the reviewer’s positive assessment of our theoretical foundation and the method's applicability. We have provided clarifications and updates to address your concerns.
>
> > Q1: Small-sample performance
>
> First, we fully agree with the reviewer that small-sample performance is a critical practical consideration. However, we respectfully note that finite-sample performance is fundamentally a concern of causal effect estimation, whereas the primary focus of this work is causal effect identification. We have thoroughly revised the manuscript to more clearly delineate the scope of our primary contribution.
>
> 1. Causal effect identification addresses the **theoretical** question of whether and how the causal effect of interest can be uniquely determined from the observational population distribution (i.e., **infinite data**). In contrast, causal effect estimation addresses the **practical** question of how to compute point estimates for the causal effect and quantify the associated uncertainty (e.g., confidence intervals) given a **finite sample** drawn from the observational population distribution. Identification is the prerequisite and theoretical foundation for consistent estimation.
>
> 2. Our core contributions lie in causal effect identification: we develop a novel causal effect identification procedure that requires only a single agnostic proxy and prove its soundness, that is, given the observational population distribution, it correctly identifies the true causal effect when identifiable, and correctly reports unidentifiability otherwise. To conduct experiments, we implement a basic plug-in estimation method by directly substituting sample-level cumulants and independence tests into our identification procedure. We wish to highlight that this implementation is intended specifically to empirically validate the correctness of our theoretical results rather than to optimize finite-sample performance. The observed convergence of estimation error to zero as the sample size increases sufficiently validates our theoretical results.
>
> 3. Existing literature demonstrates that causal effect identification is widely recognized as a self-contained theoretical contribution independent of finite-sample estimation optimization. For instance, [1,2] focus exclusively on theoretical contributions without providing any estimation methods; [3,4] use basic plug-in estimation methods similar to ours to conduct experiments; and [5] only augments the basic plug-in estimator with a post-processing step that yields observable gains primarily in large-sample regimes, while still maintaining the primary focus on identification theory.
>
> Second, we frankly acknowledge that the finite-sample performance of our plug-in estimation method is suboptimal, which is a known characteristic of estimation methods based on higher-order statistics [3-5]. However, inspired by the reviewer’s comments, we have implemented an adaptive strategy to effectively mitigate this issue and will conduct an in-depth investigation in future work.
>
> 1. Since the estimation variance increases with the order of the cumulant, utilizing lower-order cumulants whenever possible helps to improve stability. Note that $g(T,Z)$ in Equation (9) originally expressed via fourth-order cross-cumulants as $\mathrm{sgn}(Cov(T,Z)) \sqrt{C_{3,1}(T,Z) / C_{1,3}(T,Z)}$ can be equivalently expressed using third-order cross-cumulants $C_{2,1}(T,Z) / C_{1,2}(T,Z)$ when $C_{1,2}(T,Z) \neq 0$. Therefore, we update our estimation method to adaptively switch to third-order cumulants when the correlation between $T$ and $Z^2$ (i.e., a normalized version of $C_{1,2}(T,Z)$) exceeds a threshold (e.g., 0.1). When the latent confounder follows an asymmetric distribution (this guarantees $C_{1,2}(T,Z) \neq 0$) such as the Beta distribution with $\alpha=1/3, \beta=2/3$, the experimental results below demonstrate that this update remarkably reduces estimation errors in small-sample regimes.
>
>     |Sample Size|Method|Fig. 3(a)|Fig. 3(d)|Fig. 3(e)|
>     |-|-|-|-|-|
>     |500|w/o update|0.24±0.28|0.43±0.37|0.33±0.31|
>     | |w/ update|**0.16±0.15**|**0.34±0.32**|**0.24±0.21**|
>     |1000|w/o update|0.18±0.15|0.26±0.23|0.29±0.25|
>     | |w/ update|**0.12±0.09**|**0.24±0.22**|**0.23±0.22**|
>     |5000|w/o update|0.09±0.06|0.17±0.13|0.13±0.11|
>     | |w/ update|**0.07±0.05**|**0.14±0.13**|**0.11±0.09**|
>     |10000|w/o update|0.05±0.04|0.15±0.13|0.11±0.08|
>     | |w/ update|0.05±0.04|**0.13±0.14**|0.11±0.08|
>
>     (*Note*: we only provide results for Fig. 3(a), (d), and (e) because the function $g$ is not invoked for causal effect estimation for Fig. 3(b) and (c) according to Algorithm 1 in our paper)

---

> ### Author Response · Authors · 2025-11-24
> **Part 2/2**
>
> 2. The suboptimal small-sample performance is primarily attributable to the high variance of higher-order cumulant estimators, so any advancements in the estimation of high-order statistics will immediately benefit our estimation method. In future work, we will conduct an in-depth investigation of related literature [6-8] and integrate more sophisticated higher-order cumulant estimators to further improve the small-sample performance of our estimation method.
>
> ### Reference
>
> [1] Causal effect identifiability under partial-observability. ICML 2020.\
> [2] On identifiability of conditional causal effects. UAI 2023.\
> [3] A Cross-Moment Approach for Causal Effect Estimation. NeurIPS 2023.\
> [4] Causal Effect Identification in Heterogeneous Environments from Higher-Order Moments. UAI 2025.\
> [5] Causal Effect Identification in lvLiNGAM from Higher-Order Cumulants. ICML 2025.\
> [6] L-moments: analysis and estimation of distributions using linear combinations of order statistics. Journal of the Royal Statistical Society Series B: Statistical Methodology, 1990.\
> [7] Comparing measures of sample skewness and kurtosis. Journal of the Royal Statistical Society Series D: The Statistician, 1998\
> [8] On more robust estimation of skewness and kurtosis. Finance Research Letters 2004.
>
> > Q2: The risk of error accumulation in our estimation method for settings involving multiple latent confounders
>
> While this question falls within the scope of estimation (as discussed in our response to your Q1), we appreciate the query and are glad to clarify this point.
>
> First, we wish to highlight that our identification procedure for settings involving multiple latent confounders (Alg. 2) is modular rather than sequential, which means that **the risk of error accumulation is structurally avoided** by design. It is important to note that in Alg. 2, the result of each iteration does not propagate to the next iteration (see line 2-16 in Alg. 2). Instead, it simply aggregates these **independently** derived results at the final stage to identify the causal effect of interest (see line 17-29 in Alg. 2). Therefore, the plug-in estimation method directly derived from Alg.2 prevents sequential error accumulation.
>
> Second, alternative non-iterative estimation methods might face significant challenges. For instance, according to Theorem 3.1 in [1], in the presence of $n$ latent confounders, $n+1$ candidate values for the causal effect of interest can be derived from cross-cumulants of $T$ and $O$ up to order $m = (n+2)+⌈(−3+\sqrt{8n+17})/2⌉$. If we construct a non-iterative estimation method based on this theorem, we have to consider the following two problems:
>
> 1. The validity of this theorem relies on the assumption that for any exogenous noise $\epsilon$ and any order $i \leq m$, $|C_i(\epsilon)| < \infty$ and $C_i(\epsilon) \neq 0$. This assumption is restrictive. For instance, any symmetric distribution has zero odd-order cumulants, and higher-order cumulants are increasingly likely to be undefined (infinite).
>
> 2. Even if the above assumption holds, the estimators for the $n+1$ candidate values suffer from prohibitively high variance. Specifically, it is well-established that the variance of cumulant estimators grows significantly with their order, making non-iterative estimation practically unstable, especially when the number of latent confounders is large.
>
> ### Reference
>
> [1] Causal Effect Identification in Heterogeneous Environments from Higher-Order Moments. UAI 2025.

---

> ### Author Response · Authors · 2025-11-26
>
> Dear Reviewer `kuEe`,
>
> We want to express our appreciation for your valuable suggestions, which greatly helped us improve the quality of this paper. We have taken our maximum effort to address your concerns on clarification. Could you please kindly re-evaluate our work?
>
> Your further opinions are very important for evaluating our revised paper and we are hoping to hear from you. Thank you so much.
>
> Best,
>
> Authors.

---

### Official Review · Reviewer_oTr3 · 2025-11-02

**Soundness:** 3
**Presentation:** 3
**Contribution:** 3
**Rating:** 6
**Confidence:** 3

**Summary:**

This paper tackles causal effect identification under latent confounding using only a single proxy within the lvLiNGAM framework. Unlike prior methods requiring multiple or structurally known proxies, the proposed approach allows an *agnostic proxy*—one whose causal links to treatment or outcome are arbitrary and unknown. Exploiting non-Gaussianity, the method first derives candidate causal effects from cross-cumulants and then selects the valid one via independence tests. It guarantees asymptotic consistency: with sufficient data, the true causal effect is recovered if identifiable, while unidentifiability is explicitly reported otherwise. This work broadens causal inference applicability in realistic settings with limited or poorly understood proxies.

**Strengths:**

1. The "agnostic proxy" concept fills a critical gap in existing literature.
2. The proposed method is supported by rigorous theoretical analysis.
3. Experimental results demonstrate that the proposed approach.

**Weaknesses:**

1. The method relies on cross-cumulant estimators with high variance in small samples, leading to poor performance.
2. The approach is confined to the lvLiNGAM framework and cannot handle nonlinear.
3. The method jointly relies on both independence tests and cross-cumulant constraints for causal structure discrimination. If either component yields an incorrect judgment—due to estimation noise or finite-sample errors—the resulting causal effect estimate may be suboptimal or even incorrect.

**Questions:**

1. Have you considered any robustness strategies to solve the depends critically on both independence tests and cross-cumulant constraints?
2. Xu et al. demonstrate identifiability under nonlinear structural causal models in Figure 1(a). How does your method relate to or differ from their framework? Do you see potential for integrating your cross-cumulant + independence test approach with nonlinear identifiability results to extend applicability beyond the linear lvLiNGAM setting?


[1] Kernel single proxy control for deterministic confounding.

---

> ### Author Response · Authors · 2025-11-24
> **Part 1/3**
>
> We are grateful to the reviewer for the time and effort dedicated to reviewing our paper. We appreciate the opportunity to clarify the robustness and applicability of our work based on your valuable feedback.
>
> > Q1: Small-sample performance
>
> First, we fully agree with the reviewer that small-sample performance is a critical practical consideration. However, we respectfully note that finite-sample performance is fundamentally a concern of causal effect estimation, whereas the primary focus of this work is causal effect identification. We have thoroughly revised the manuscript to more clearly delineate the scope of our primary contribution.
>
> 1. Causal effect identification addresses the **theoretical** question of whether and how the causal effect of interest can be uniquely determined from the observational population distribution (i.e., **infinite data**). In contrast, causal effect estimation addresses the **practical** question of how to compute point estimates for the causal effect and quantify the associated uncertainty (e.g., confidence intervals) given a **finite sample** drawn from the observational population distribution. Identification is the prerequisite and theoretical foundation for consistent estimation.
>
> 2. Our core contributions lie in causal effect identification: we develop a novel causal effect identification procedure that requires only a single agnostic proxy and prove its soundness, that is, given the observational population distribution, it correctly identifies the true causal effect when identifiable, and correctly reports unidentifiability otherwise. To conduct experiments, we implement a basic plug-in estimation method by directly substituting sample-level cumulants and independence tests into our identification procedure. We wish to highlight that this implementation is intended specifically to empirically validate the correctness of our theoretical results rather than to optimize finite-sample performance. The observed convergence of estimation error to zero as the sample size increases sufficiently validates our theoretical results.
>
> 3. Existing literature demonstrates that causal effect identification is widely recognized as a self-contained theoretical contribution independent of finite-sample estimation optimization. For instance, [1,2] focus exclusively on theoretical contributions without providing any estimation methods; [3,4] use basic plug-in estimation methods similar to ours to conduct experiments; and [5] only augments the basic plug-in estimator with a post-processing step that yields observable gains primarily in large-sample regimes, while still maintaining the primary focus on identification theory.
>
> Second, we frankly acknowledge that the finite-sample performance of our plug-in estimation method is suboptimal, which is a known characteristic of estimation methods based on higher-order statistics [3-5]. However, inspired by the reviewer’s comments, we have implemented an adaptive strategy to effectively mitigate this issue and will conduct an in-depth investigation in future work.
>
> 1. Since the estimation variance increases with the order of the cumulant, utilizing lower-order cumulants whenever possible helps to improve stability. Note that $g(T,Z)$ in Equation (9) originally expressed via fourth-order cross-cumulants as $\mathrm{sgn}(Cov(T,Z)) \sqrt{C_{3,1}(T,Z) / C_{1,3}(T,Z)}$ can be equivalently expressed using third-order cross-cumulants $C_{2,1}(T,Z) / C_{1,2}(T,Z)$ when $C_{1,2}(T,Z) \neq 0$. Therefore, we update our estimation method to adaptively switch to third-order cumulants when the correlation between $T$ and $Z^2$ (i.e., a normalized version of $C_{1,2}(T,Z)$) exceeds a threshold (e.g., 0.1). When the latent confounder follows an asymmetric distribution (this guarantees $C_{1,2}(T,Z) \neq 0$) such as the Beta distribution with $\alpha=1/3, \beta=2/3$, the experimental results below demonstrate that this update remarkably reduces estimation errors in small-sample regimes.
>
>     |Sample Size|Method|Fig. 3(a)|Fig. 3(d)|Fig. 3(e)|
>     |-|-|-|-|-|
>     |500|w/o update|0.24±0.28|0.43±0.37|0.33±0.31|
>     | |w/ update|**0.16±0.15**|**0.34±0.32**|**0.24±0.21**|
>     |1000|w/o update|0.18±0.15|0.26±0.23|0.29±0.25|
>     | |w/ update|**0.12±0.09**|**0.24±0.22**|**0.23±0.22**|
>     |5000|w/o update|0.09±0.06|0.17±0.13|0.13±0.11|
>     | |w/ update|**0.07±0.05**|**0.14±0.13**|**0.11±0.09**|
>     |10000|w/o update|0.05±0.04|0.15±0.13|0.11±0.08|
>     | |w/ update|0.05±0.04|**0.13±0.14**|0.11±0.08|
>
>     (*Note*: we only provide results for Fig. 3(a), (d), and (e) because the function $g$ is not invoked for causal effect estimation for Fig. 3(b) and (c) according to Algorithm 1 in our paper)

---

> ### Author Response · Authors · 2025-11-24
> **Part 2/3**
>
> 2. The suboptimal small-sample performance is primarily attributable to the high variance of higher-order cumulant estimators, so any advancements in the estimation of high-order statistics will immediately benefit our estimation method. In future work, we will conduct an in-depth investigation of related literature [6-8] and integrate more sophisticated higher-order cumulant estimators to further improve the small-sample performance of our estimation method.
>
> ### Reference
>
> [1] Causal effect identifiability under partial-observability. ICML 2020.\
> [2] On identifiability of conditional causal effects. UAI 2023.\
> [3] A Cross-Moment Approach for Causal Effect Estimation. NeurIPS 2023.\
> [4] Causal Effect Identification in Heterogeneous Environments from Higher-Order Moments. UAI 2025.\
> [5] Causal Effect Identification in lvLiNGAM from Higher-Order Cumulants. ICML 2025.\
> [6] L-moments: analysis and estimation of distributions using linear combinations of order statistics. Journal of the Royal Statistical Society Series B: Statistical Methodology, 1990.\
> [7] Comparing measures of sample skewness and kurtosis. Journal of the Royal Statistical Society Series D: The Statistician, 1998\
> [8] On more robust estimation of skewness and kurtosis. Finance Research Letters 2004.
>
> > Q2: lvLiNGAM framework & extension to nonlinear scenarios
>
> First, we focus on the lvLiNGAM framework for three primary reasons: (1) causal effect identification with a single proxy is often impossible outside the lvLiNGAM framework, let alone identification with a single agnostic proxy; (2) lvLiNGAM can be applied to solve real-world problems; and (3) lvLiNGAM remains a vibrant and active area of research in recent causal inference literature.
>
> 1. Literature has established that it is often impossible to identify the causal effect with a single proxy outside the lvLiNGAM framework. Specifically, [1] have proven that in **linear Gaussian** models with latent variables, the causal effect is unidentifiable even with a single **isolated** proxy. Furthermore, it is known that we cannot recover the true causal effect with a single proxy in general non-parametric models [2-4], unless many other restrictive assumptions are satisfied (e.g.,  the proxy is an isolated proxy and the distribution of the proxy conditioned on the latent confounder is known a priori [3]). Our setting allows for an agnostic proxy, rendering the identification problem even more challenging. This underscores the necessity of focusing on the lvLiNGAM framework.
>
> 2. By leveraging linearization techniques around the operating point of a system, complex causal mechanisms can be effectively approximated by linear functions in many real-world applications. Indeed, LiNGAM, a simplified version of lvLiNGAM without latent confounders, has been used to address practical problems across diverse domains, e.g., economics [5], marketing [6], neuroscience [7], epidemiology [8], chemistry [9], and environmental science [10].
>
> 3. The linear model continues to be a central subject of investigation in current causal inference research [11-16]. This validates its continued relevance and importance to the community.
>
> Second, we agree that extending our framework to nonlinear scenarios is a promising direction. In our framework, the cross-cumulant constraint and the independence constraint jointly serve as a vanishing condition, indicating the absence of a causal relationship between two variables sharing a common cause. Consequently, if we find a specific value $\alpha$ such that the treatment $T$ and the residual $O - \alpha T$ satisfy these two constraints, then $\alpha$ is exactly the causal effect. This perspective suggests a viable pathway for generalizing our framework to certain nonlinear structural causal models (e.g., additive noise models subject to specific assumptions that guarantee identifiability) by formulating nonlinear variants of these constraints, possibly through kernel representations as in [4]. We leave this exciting development for future work.
>
> Finally, we evaluate the robustness against mild nonlinearity of the plug-in estimation method directly derived from our identification procedure. The experimental results show that its performance does not significantly deteriorate when the nonlinearity is not excessively strong. We present the estimation errors for Fig. 3(b) with Laplace exogenous noises here. Please refer to Figure 18 in Appendix E for the comprehensive experimental results.
>
> | Sample Size | 500 | 1000 | 5000 | 10000 | 50000 |
> |-|-|-|-|-|-|
> |Linear|0.32±0.18|0.29±0.22|0.11±0.07|0.09±0.06|0.03±0.03|
> |Nonlinear|0.39±0.25|0.31±0.21|0.16±0.13|0.09±0.07|0.04±0.03|
>
> ### Reference
>
> [1] A Cross-Moment Approach for Causal Effect Estimation. NeurIPS 2023.\
> [2] On measurement bias in causal inference. UAI 2010.\
> [3] Measurement bias and effect restoration in causal inference. Biometrika 2014.\
> [4] Kernel Single Proxy Control for Deterministic Confounding. Arxiv 2023.

---

> ### Author Response · Authors · 2025-11-24
> **Part 3/3**
>
> [5] Causal inference by independent component analysis: Theory and applications. Oxford Bulletin of Economics and Statistics 2013.\
> [6] Causal inference for contemporaneous effects and its application to tourism product sales data. Journal of Marketing Analytics 2022.\
> [7] Asymmetric directed functional connectivity within the frontoparietal motor network during motor imagery and execution. NeuroImage 2022.\
> [8] Pairwise measures of causal direction in the epidemiology of sleep problems and depression. Plos one 2012.\
> [9] Origin of the spectral shifts among the early intermediates of the rhodopsin photocycle. Journal of the American Chemical Society 2014.\
> [10] Validation of causal inference data using DirectLiNGAM in an environmental small-scale model and calculation settings. MethodsX 2024.\
> [11] Causal Effect Identification in LiNGAM Models with Latent Confounders. ICML 2024.\
> [12] Efficient and Trustworthy Causal Discovery with Latent Variables and Complex Relations. ICLR 2025.\
> [13] Recovery of causal graph involving latent variables via homologous surrogates. ICLR 2025.\
> [14] Linear SCM Identification in the Presence of Confounders and Gaussian Noise. ICLR 2025.\
> [15] Causal Effect Identification in lvLiNGAM from Higher-Order Cumulants. ICML 2025.\
> [16] Causal Effect Identification in Heterogeneous Environments from Higher-Order Moments. UAI 2025.
>
> > Q3: A related work (Xu et al., 2023) [1].
>
> We thank the reviewer for mentioning this relevant work. While we and [1] both tackle causal identification with latent confounding using a single proxy, they represent distinct theoretical trade-offs tailored to different problem settings. Specifically,
>
> - (Differences in assumptions) We assume the causal relationships are linear but require only an agnostic proxy, which means that the proxy can have a flexible and unknown causal relationship with the treatment/outcome. In contrast, [1] allows nonlinear causal relationships, but they assume the proxy is an isolated proxy, which means that the proxy must have no causal relationship with the treatment or the outcome. In addition, they assume the outcome is deterministically generated given the treatment and the latent confounder.
> - (Difference in methodology) Our identification procedure identifies the causal effect based on cross-cumulants and independence relationships, whereas [1] is based on the bridge function. The bridge function is difficult to estimate with finite data, so they specifically design two kernel-based methods to estimate it.
>
> ### Reference
>
> [1] Kernel Single Proxy Control for Deterministic Confounding. Arxiv 2023.
>
> > Q4: Reducing the impact of errors in independence tests and high-order cumulants estimation.
>
> While this question falls within the scope of estimation (as discussed in our response to your Q1), we appreciate the query and have implemented specific strategies to enhance robustness.
>
> Independence tests have long been a focal point in the statistics community and a vital tool in causal inference, with many mature algorithms already available. In contrast, it is widely acknowledged that the estimation of high-order cumulants is extremely sensitive to outliers, particularly when the sample size is small. Therefore, our primary challenge lies in reducing the impact of errors in high-order cumulants estimation. We address this through the following two aspects.
>
> 1. By utilizing lower-order cumulants whenever possible, estimation errors can be directly reduced. Please refer to our response to your Q1 for more details.
>
> 2. Independence tests can effectively reduce the impact of errors in high-order cumulants estimation. Specifically, if the estimation error is substantial, an independence constraint that theoretically holds is likely to be rejected. In such cases, our estimation method reports unidentifiability rather than returns a potentially erroneous value. This behavior is a desirable safety feature: it prevents the method from returning a confidently wrong causal effect, thereby protecting downstream tasks from erroneous inputs.

---

> ### Author Response · Authors · 2025-11-27
>
> Dear Reviewer `oTr3`:
>
> We really appreciate your constructive opinions that helped us improve this paper. If there is any concern unresolved, we would be glad to have further discussions.
>
> Thanks again for your time, looking forward to hearing from you soon.
>
> Best,
>
> Authors.

---

### Official Review · Reviewer_S1Fn · 2025-11-02

**Soundness:** 3
**Presentation:** 2
**Contribution:** 2
**Rating:** 4
**Confidence:** 4

**Summary:**

The paper studies identification of the causal effect of a treatment $T$ on an outcome $O$ under latent confounding, assuming lvLiNGAM (linear, non-Gaussian, acyclic) structure. The key contribution is an identification and estimation procedure that requires only a single proxy $Z$ of the latent confounder $L$, and that proxy can be agnostic, i.e., it may have arbitrary and a priori unknown causal relationships with $T$ and/or $O$. The method first generates candidate values for the causal effect from cross-cumulants (solving a quadratic with coefficients given by cumulant polynomials) and then selects the valid candidate via independence tests. The authors prove asymptotic consistency and provide algorithms for both the ``basic agnostic proxy'' setting and a generalized setting where $Z$ can be placed anywhere relative to $(L,T,O)$; in the latter, they first decide (via independence tests) whether the graph is in a family where identification is possible and otherwise report unidentifiability.

**Strengths:**

- Identification with just one proxy whose causal links to $T$ and $T$ need not be known or restricted (beyond lvLiNGAM).
- The paper delineates when the effect is identifiable vs. provably unidentifiable  and provides Algorithm 1 that either outputs the effect or correctly returns “unidentifiable.”

**Weaknesses:**

- While common in applications, the setup focuses on one treatment–outcome pair and a single proxy. Extensions to multiple treatments/outcomes, multiple proxies, or networked settings are not developed here.

- Performance can degrade at modest sample sizes because cumulant estimation is high-variance.

**Questions:**

- Which concrete independence tests (and thresholds) are recommended in practice for the selection step? How sensitive is performance to these choices in moderate $n$ regimes?

- In Fig. 7, subplots for Fig. 3(a), the curves of the proposed method are not observable. It is better to update the plots. Moreover, the subplots for Fig. 3(b), Beta, the results are not consistent with the plots in (Tramontano et al., 2025). Moreover, there are two versions of the cumulant algorithm there and the cumulant with minimization has a better performance.

- If several agnostic proxies are available, can your framework be combined across proxies?

- The paper frequently compares to methods that assume the causal structure is known. However, in lvLiNGAM with latent confounding, one can first recover the causal order from observational data using recent cumulant–rank–based discovery results (e.g., Schkoda et al. 2024), and then apply the effect-identification step (such as (Tramontano et al., 2025)). In what sense is the present contribution novel beyond this discover-and-apply pipeline?

---

> ### Author Response · Authors · 2025-11-24
> **Part 1/3**
>
> We sincerely thank the reviewer for the detailed review and constructive comments. We have carefully addressed your concerns and revised the manuscript accordingly.
>
> > Q1: Small-sample performance
>
> First, we fully agree with the reviewer that small-sample performance is a critical practical consideration. However, we respectfully note that finite-sample performance is fundamentally a concern of causal effect estimation, whereas the primary focus of this work is causal effect identification. We have thoroughly revised the manuscript to more clearly delineate the scope of our primary contribution.
>
> 1. Causal effect identification addresses the **theoretical** question of whether and how the causal effect of interest can be uniquely determined from the observational population distribution (i.e., **infinite data**). In contrast, causal effect estimation addresses the **practical** question of how to compute point estimates for the causal effect and quantify the associated uncertainty (e.g., confidence intervals) given a **finite sample** drawn from the observational population distribution. Identification is the prerequisite and theoretical foundation for consistent estimation.
>
> 2. Our core contributions lie in causal effect identification: we develop a novel causal effect identification procedure that requires only a single agnostic proxy and prove its soundness, that is, given the observational population distribution, it correctly identifies the true causal effect when identifiable, and correctly reports unidentifiability otherwise. To conduct experiments, we implement a basic plug-in estimation method by directly substituting sample-level cumulants and independence tests into our identification procedure. We wish to highlight that this implementation is intended specifically to empirically validate the correctness of our theoretical results rather than to optimize finite-sample performance. The observed convergence of estimation error to zero as the sample size increases sufficiently validates our theoretical results.
>
> 3. Existing literature demonstrates that causal effect identification is widely recognized as a self-contained theoretical contribution independent of finite-sample estimation optimization. For instance, [1,2] focus exclusively on theoretical contributions without providing any estimation methods; [3,4] use basic plug-in estimation methods similar to ours to conduct experiments; and [5] only augments the basic plug-in estimator with a post-processing step that yields observable gains primarily in large-sample regimes, while still maintaining the primary focus on identification theory.
>
> Second, we frankly acknowledge that the finite-sample performance of our plug-in estimation method is suboptimal, which is a known characteristic of estimation methods based on higher-order statistics [3-5]. However, inspired by the reviewer’s comments, we have implemented an adaptive strategy to effectively mitigate this issue and will conduct an in-depth investigation in future work.
>
> 1. Since the estimation variance increases with the order of the cumulant, utilizing lower-order cumulants whenever possible helps to improve stability. Note that $g(T,Z)$ in Equation (9) originally expressed via fourth-order cross-cumulants as $\mathrm{sgn}(Cov(T,Z)) \sqrt{C_{3,1}(T,Z) / C_{1,3}(T,Z)}$ can be equivalently expressed using third-order cross-cumulants $C_{2,1}(T,Z) / C_{1,2}(T,Z)$ when $C_{1,2}(T,Z) \neq 0$. Therefore, we update our estimation method to adaptively switch to third-order cumulants when the correlation between $T$ and $Z^2$ (i.e., a normalized version of $C_{1,2}(T,Z)$) exceeds a threshold (e.g., 0.1). When the latent confounder follows an asymmetric distribution (this guarantees $C_{1,2}(T,Z) \neq 0$) such as the Beta distribution with $\alpha=1/3, \beta=2/3$, the experimental results below demonstrate that this update remarkably reduces estimation errors in small-sample regimes.
>
>     |Sample Size|Method|Fig. 3(a)|Fig. 3(d)|Fig. 3(e)|
>     |-|-|-|-|-|
>     |500|w/o update|0.24±0.28|0.43±0.37|0.33±0.31|
>     | |w/ update|**0.16±0.15**|**0.34±0.32**|**0.24±0.21**|
>     |1000|w/o update|0.18±0.15|0.26±0.23|0.29±0.25|
>     | |w/ update|**0.12±0.09**|**0.24±0.22**|**0.23±0.22**|
>     |5000|w/o update|0.09±0.06|0.17±0.13|0.13±0.11|
>     | |w/ update|**0.07±0.05**|**0.14±0.13**|**0.11±0.09**|
>     |10000|w/o update|0.05±0.04|0.15±0.13|0.11±0.08|
>     | |w/ update|0.05±0.04|**0.13±0.14**|0.11±0.08|
>
>     (*Note*: we only provide results for Fig. 3(a), (d), and (e) because the function $g$ is not invoked for causal effect estimation for Fig. 3(b) and (c) according to Algorithm 1 in our paper)

---

> ### Author Response · Authors · 2025-11-24
> **Part 2/3**
>
> 2. The suboptimal small-sample performance is primarily attributable to the high variance of higher-order cumulant estimators, so any advancements in the estimation of high-order statistics will immediately benefit our estimation method. In future work, we will conduct an in-depth investigation of related literature [6-8] and integrate more sophisticated higher-order cumulant estimators to further improve the small-sample performance of our estimation method.
>
> ### Reference
>
> [1] Causal effect identifiability under partial-observability. ICML 2020.\
> [2] On identifiability of conditional causal effects. UAI 2023.\
> [3] A Cross-Moment Approach for Causal Effect Estimation. NeurIPS 2023.\
> [4] Causal Effect Identification in Heterogeneous Environments from Higher-Order Moments. UAI 2025.\
> [5] Causal Effect Identification in lvLiNGAM from Higher-Order Cumulants. ICML 2025.\
> [6] L-moments: analysis and estimation of distributions using linear combinations of order statistics. Journal of the Royal Statistical Society Series B: Statistical Methodology, 1990.\
> [7] Comparing measures of sample skewness and kurtosis. Journal of the Royal Statistical Society Series D: The Statistician, 1998\
> [8] On more robust estimation of skewness and kurtosis. Finance Research Letters 2004.
>
> > Q2: Extensions to the setting involving multiple treatments / outcomes / agnostic proxies.
>
> We thank the reviewer for this insightful suggestion regarding scalability. While our results are developed for the setting with a single treatment, outcome, and agnostic proxy, we emphasize that this setup serves as a fundamental building block for more complex scenarios. Specifically, our results can be naturally extended to certain multi-variable systems by applying the identification procedure element-wise to decompose the system. We provide three examples below:
>
> 1. Consider a setting with one treatment $T$, one outcome $O$, and multiple agnostic proxies $Z_1,...,Z_n$ of the latent confounder $L$, where $T$ is a parent of $O$, $L$ is a common parent of $(T, O, Z_1,...,Z_n)$, and the causal relationship between $(Z_1,...,Z_n)$ and $(T,O)$ is not known a priori. If each $Z_i$ is not a parent of another $Z_j$ and each $Z_i$ is not a common parent of $(T,O)$, then each subsystem $(L, T, O, Z_i)$ is causally sufficient, so our theoretical results remain valid for each $(T, O, Z_i)$. We can apply our identification procedure to each $(T, O, Z_i)$. If it returns a value for at least one $(T, O, Z_i)$, the causal effect is identified as that value; otherwise, if it reports unidentifiability for each $(T, O, Z_i)$, the causal effect is unidentifiable.
>
> 2. Consider a setting with one treatment $T$, multiple outcomes $O_1,...,O_n$, and one agnostic proxy $Z$ of the latent confounder $L$, where $T$ is a common parent of $(O_1,...,O_n)$, $L$ is a common parent of $(T, O_1,...,O_n, Z)$, and the causal relationship between $Z$ and $(T,O_1,...,O_n)$ is not known a priori. If each $O_i$ is not a parent of any other $O_j$, then each subsystem $(L, T, O_i, Z)$ is causally sufficient, so our theoretical results remain valid for each $(T, O_i, Z)$. We can apply our identification procedure to each $(T, O_i, Z)$. It either returns a value for each $(T, O_i, Z)$ (which occurs when $T$ is not a parent of $Z$, that is, the causal relationship between $Z$ and each $(T,O_i)$ is one of Fig. 3(a)-(e)) or reports unidentifiability for each $(T, O_i, Z)$ (which occurs when $T$ is a parent of $Z$, that is, the causal relationship between $Z$ and each $(T,O_i)$ is one of Fig. 3(f)-(h)). In the former case, the causal effect of $T$ on each $O_i$ is identified as the value; in the latter case, the causal effect of $T$ on any $O_i$ is unidentifiable.
>
> 3. For the more complex setting involving multiple treatments $T_1,...,T_l$, multiple outcomes $O_1,...,O_m$, and multiple proxies $Z_1,...,Z_n$, addressing the fully agnostic case remains an open challenge. However, if the underlying causal structure is known a priori, our results can still be applied. Specifically, for each treatment-outcome pair $(T_i,O_j)$, we can find a set of variables $\mathbf{V}_ {ij} \subseteq \\{T_1,...,T_l,O_1,...,O_m,Z_1,...,Z_n\\} \backslash \\{T_i, O_j\\}$ s.t. for each $V \in \mathbf{V}_ {ij}$, the subsystem $(L,T_i,O_j,V)$ is causally sufficient. If there exists $V \in \mathbf{V}_{ij}$ s.t. the causal relationship between $V$ and $(T_i,O_j)$ corresponds to one of Fig. 3(a)-(e), the causal effect of interest can be identified based on one of Theorems 1-5.

---

> ### Author Response · Authors · 2025-11-24
> **Part 3/3**
>
> > Q3: Details about the independence test
>
> While this question falls within the scope of estimation (as discussed in our response to Q1), we appreciate the query and recommend the following configuration. We employ the Hilbert-Schmidt Independence Criterion (HSIC) for independence testing. This choice is motivated by its established effectiveness as a non-parametric method in detecting arbitrarily complex, nonlinear dependencies, as well as the availability of robust Python implementations (https://github.com/oxcsml/kerpy). Conventionally, we set the significance level to $0.05$.
>
> To evaluate the sensitivity of our estimation method to the significance level, we also conduct experiments using significance levels of $0.1$ and $0.01$, which are the other two common values. The experimental results demonstrate that our estimation method is robust to the choice of significance level. We present the estimation errors for Fig. 3(b) with Laplace exogenous noises here. Please refer to Figure 15 in Appendix E for the comprehensive experimental results.
>
> | Sample Size | 500 | 1000 | 5000 | 10000 | 50000 |
> |-|-|-|-|-|-|
> |$\alpha=0.1$|0.31±0.22|0.28±0.21|0.11±0.07|0.09±0.07|0.03±0.03|
> |$\alpha=0.05$|0.32±0.18|0.29±0.22|0.11±0.07|0.09±0.06|0.03±0.03|
> |$\alpha=0.01$|0.35±0.20|0.33±0.22|0.13±0.08|0.09±0.06|0.04±0.03|
>
> > Q4: Updated experimental results in Fig. 7.
>
> We thank the reviewer for the careful observation regarding the experimental results shown in Fig. 7. We have updated Fig. 7 in the revised manuscript.
>
> 1. Due to the update to our estimation method described in our response to your Q1, the performance curve for our estimation method is now clearly visible in Fig. 7.
>
> 2. Our results are largely consistent with those reported in Tramontano et al. (2025) [1]. Minor discrepancies are attributable to slight differences in the parameters of the Beta distribution used in our study compared to [1].
>
> 3. We have applied the more advanced version of Cumulant, Cumulant-with-minimization, to Fig. 3(b). We have explicitly highlighted this distinction in the revised manuscript.
>
> ### Reference
>
> [1] Causal Effect Identification in lvLiNGAM from Higher-Order Cumulants. ICML 2025.
>
> > Q5: Comparison with a discover-and-estimate pipeline.
>
> We thank the reviewer for bringing the work of Schkoda et al. (2024) [1] to our attention. We note that during structure recovery, their causal discovery method concurrently estimates causal coefficients, which implies that the method itself is capable of estimating the causal effect of interest. In the following, we outline two primary distinctions between our estimation method and their causal discovery method.
>
> 1. Our estimation method is designed to estimate a specific causal effect of interest directly, avoiding the computational overhead of full structure learning. In contrast, although the causal discovery method in [1] can also be used for causal effect estimation in principle, it inherently needs to iteratively estimate all possible values for the causal effect between every pair of variables, along with all possible values for the cumulants of every exogenous noise, to eventually yield an estimate for the causal effect of interest or conclude unidentifiability. This significantly increases computational costs and the risk of error accumulation.
>
> 2. Our estimation method imposes weaker distributional assumptions. Specifically, we only require that fourth-order cumulants are finite and non-zero. In contrast, [1] assumes that cumulants of other orders are also finite and non-zero. This stronger assumption is restrictive. For instance, any symmetric distribution has zero odd-order cumulants, and higher-order cumulants are increasingly likely to be undefined (infinite).
>
> Finally, although the work of [1] is less suitable for causal effect estimation, it represents a significant advancement in causal discovery. The causal discovery method in [1] imposes minimal assumptions on the underlying causal structure, allowing it to handle significantly more complex scenarios. Therefore, our work and [1] can coexist without diminishing each other's contributions.
>
> ### Reference
>
> [1] Causal Discovery of Linear Non-Gaussian Causal Models  with Unobserved Confounding. Arxiv 2024

---

> ### Author Response · Authors · 2025-11-26
>
> Dear Reviewer `S1Fn`,
>
> We really appreciate your efforts to help improve this paper. We have carefully addressed your concerns. It is really important to us that you could kindly read our rebuttal and provide further questions if there are any.
>
> Thank you so much and hope you have a good day.
>
> Best,
>
> Authors.

---

### Author Response · Authors · 2025-11-27
**Awaiting Reviewer Response - Submission 11612**

Dear Reviewers `S1Fn`, `oTr3`, and `kuEe`,

Thank you for your valuable feedback. In our rebuttal, we have tried our best to address your concerns with detailed explanations and clarifications. If there are any concerns unresolved, we would be glad to have further discussions.

Thanks again for your time, looking forward to hearing from you soon.

Best regards,\
Authors of submission 11612

---

### Author Response · Authors · 2025-11-30
**Summary**

Dear AC & SAC,

We totally understand your heavy workload, so to hopefully make your work a bit easier, we summarize both our contributions and our rebuttal.

## I. Contributions

The primary contributions of our work can be summarized into three aspects:

1. Motivated by the observation that in real-world applications, obtaining multiple proxies is often infeasible and the causal relationship between the proxy and the treatment/outcome is typically flexible and unknown, we first investigate the challenging problem of causal effect identification in the presence of latent confounding with single agnostic proxy, where "agnostic proxy" encapsulates both flexibility and uncertainty.

2. Since the complexity of the agnostic proxy precludes identifying the causal effect via a simple analytical formula, we develop a novel two-stage identification procedure. This identification procedure first derives a set of candidate solutions based on cross-cumulants and subsequently isolates the valid solution by examining certain independence relationships.

3. We present and prove a series of new theoretical results that collectively establish the soundness of our identification procedure. Specifically, given the observational population distribution, it correctly identifies the true causal effect when identifiable, and correctly reports unidentifiability otherwise.

## II. Rebuttal

During the discussion period (before Nov 27 AoE), Reviewer `o4cu`, who raised the most concerns initially, confirmed their satisfaction and raised their score from 2 to 6 (on Nov 26th AoE). Although other reviewers did not replied, we have fully addressed their specific requests in our rebuttal.

### 1. Reviewer `o4cu` (Score: 2 $\to$ 6)

Reviewer `o4cu` initially raised the most concerns. To address these, we conduct the following:

- (first point of response to Q1) We highlight that our primary focus is population-level causal effect identification rather than sample-level estimation. Methodologically, our core contribution is the identification procedure, from which the estimation method is directly derived.
- (second point of response to Q1 & response to Q5) Although estimation is not the primary focus, to address the reviewer's concern, we develop a new technique to improve small-sample performance. Also, we provide extensive experimental results for a comprehensive evaluation.
- (response to Q2) We explain why we focus on the lvLiNGAM framework and clarify that our results have the potential to generalize to nonlinear scenarios.
- (response to Q3) We elaborate on the distinctions between our work and [1-3].
- (response to Q4) To improve readability of our paper, we significantly reorganize the Main Results section to bridge the gap between theoretical rigor and intuitive understanding.

Following a further round of discussion (from Nov 25th to Nov 26th AoE) and a review of our responses to other reviewers, Reviewer `o4cu` confirmed they have no further concerns and acknowledged the contributions of our paper, raising the score from 2 to 6 (on Nov 26th AoE).

### 2. Reviewer `kuEe`

Excluding points overlapping with Reviewer `o4cu`, we address Reviewer `kuEe`'s specific concerns as follows:
- (response to Q2) We clarify a misunderstanding regarding error accumulation: there is no risk of error accumulation in the estimation method which is directly derived from the identification procedure shown in Alg. 2, Appendix C.2.

### 3. Reviewer `oTr3`

Excluding points overlapping with Reviewer `o4cu`, we address Reviewer `oTr3`'s specific concerns as follows:
- (response to Q3) We elaborate on the distinctions between our work and [4].
- (response to Q4) Although estimation is not our primary focus, we explain in detail how to reduce the impact of estimation errors in practice.

### 4. Reviewer `S1Fn`

Excluding points overlapping with Reviewer `o4cu`, we address Reviewer `S1Fn`'s specific concerns as follows:
- (response to Q2) We detail how our results can be directly applied to certain scenarios involving multiple proxies/treatments/outcomes.
- (response to Q5) We elaborate on the distinctions between our work and [5].
- (response to Q3 & response to Q4) We address the reviewer's minor concerns regarding the estimation method and experimental results.

### Reference

[1] Causal discovery with latent confounders based on higher-order cumulants. ICML 2023.\
[2] A cross-moment approach for causal effect estimation. NeurIPS 2023.\
[3] Identification of causal structure with latent variables based on higher order cumulants. AAAI 2024.\
[4] Kernel Single Proxy Control for Deterministic Confounding. Arxiv 2023.\
[5] Causal Discovery of Linear Non-Gaussian Causal Models with Unobserved Confounding. Arxiv 2024.

We sincerely appreciate your dedication and extra effort in navigating these challenges for the ML community. We believe that the full context of the discussion will be helpful for your final assessment.

Best regards,\
Authors

---

### Meta-Review · Area_Chair_a1Vk · 2026-01-06

**Summary:**

The paper address the causal identification problem with latent coufounder under linear non-Gaussian acyclic model framework. The proposed method relies on a single proxy while allowing unknown arbitrary agnostic proxies. With the relationship are not fully known, the method first generate causal effect as "candidate" via cumulants and select valid candidate via independence testing procedures. The asymptotic consistency for the procedure has been provided and basic agnostic proxy as well as the generalised setting are discussed. The identifiability and unidentificability families are reported. The work provides an interesting novel scope for applicability on realistic proxy observations in causal inference.

**Reviewer Concerns:**

Small sample performance:
partially address by having additional clarifications and empirical results with smaller sample size.

Details of independence test:
additional details on HSIC based procedures has been added and clarified.


Limited application via additive model assumptions: partially mentioned yet explicitly discussed.


Additional references has also been added to support part of the clarification.

**Reviewer Scores:**

Score seems to already be adjusted and may not change further.

---

### Decision · Program_Chairs · 2026-01-26

Reject